# Unequal climate impacts on global values of natural capital

B. A. Bastien-Olvera[1,2 ✉], M. N. Conte[3], X. Dong[4], T. Briceno[5], D. Batker[6], J. Emmerling[7], M. Tavoni[7,8], F. Granella[7] & F. C. Moore[4]

Ecosystems generate a wide range of benefits for humans, including some market goods as well as other benefits that are not directly reflected in market activity[1]. Climate change will alter the distribution of ecosystems around the world and change the flow of these benefits[2,3]. However, the specific implications of ecosystem changes for human welfare remain unclear, as they depend on the nature of these changes, the value of the affected benefits and the extent to which communities rely on natural systems for their well-being[4]. Here we estimate country-level changes in economic production and the value of non-market ecosystem benefits resulting from climate-change-induced shifts in terrestrial vegetation cover, as projected by dynamic global vegetation models (DGVMs) driven by general circulation climate models. Our results show that the annual population-weighted mean global flow of non-market ecosystem benefits valued in the wealth accounts of the World Bank will be reduced by 9.2% in 2100 under the Shared Socioeconomic Pathway SSP2-6.0 with respect to the baseline no climate change scenario and that the global population-weighted average change in gross domestic product (GDP) by 2100 is −1.3% of the baseline GDP. Because lower-income countries are more reliant on natural capital, these GDP effects are regressive. Approximately 90% of these damages are borne by the poorest 50% of countries and regions, whereas the wealthiest 10% experience only 2% of these losses.

Climate change has direct, widespread and long-lasting impacts on the structure and functioning of ecosystems globally[2,3,5]. This alters both market and non-market benefits that people derive from nature[6]. However, current economic estimates of climate-change damages that inform climate policy, such as the social cost of carbon, do not fully account for these changes or include outdated assessments of these ecological impacts[7–9]. Several papers have called for improved assessments of the effects of climate change on human well-being through its impacts on ecosystems[4,10], as their consideration markedly changes climate damage estimates[11].

Human well-being can be divided into goods and services exchanged in markets (hereafter referred to as market benefits); use benefits from nature that are not usually exchanged in markets (hereafter referred to as non-market benefits); and non-use values from biodiversity and ecosystems attached only to their existence (Fig. 1). In this study, we focus on the first two components. Well-being arises from a stock of valuable assets that includes human capital, manufactured capital and natural capital[12–16]. Here we expand a regional benefit–cost climate integrated assessment model to explore the welfare effects of climate impacts on terrestrial natural capital. This disaggregated, global analysis reveals how the burden of foregone value from reduced biome spatial extent is borne differentially in countries around the globe.

To conduct this analysis, we combine several models and datasets (Extended Data Fig. 1). First, we attribute the country-level data on natural capital stock values in the World Bank Changing Wealth of Nations 2021 report[12] to the biomes contained in the borders of each country. To do so, we apply random forest algorithms to subsets of ecosystem-service-valuation studies in the values of ecosystem goods and services (VEGS) database[17]. Next, we estimate future trajectories of country-level natural capital stocks under climate-change-driven biome range shifts using outputs from DGVMs. Then we use an extended version of the open-source RICE50+ integrated assessment model[18] following initial work in ref. 11 to estimate the effect of changing natural capital on the market benefits (as measured by GDP) and non-market benefits in a fully consistent framework (Extended Data Fig. 1c), enabling future research under diverse scenarios, including emissions optimization and cross-country cooperation on climate and ecosystem conservation.

In our research, market-based natural capital (mN) represents the projected value from timber revenues and non-market natural capital (nN) is intended to estimate the value of forest-related recreational services, water resources, non-timber forest products and the inherent value of protected areas. The flow of benefits from the market natural capital stock of a country, the market environmental benefits, are included in GDP, whereas the non-market benefits that flow from the non-market natural capital stock are not. These two types of natural capital consist of a larger fraction of total national wealth in lower-income and middle-income countries[12] (Fig. 1). Consequently, these countries

[1]Geography Graduate Group, University of California, Davis, Davis, CA, USA. [2]Scripps Institution of Oceanography, University of California, San Diego, San Diego, CA, USA. [3]Department of Economics, Fordham University, Bronx, NY, USA. [4]Department of Environmental Science & Policy, University of California, Davis, Davis, CA, USA. [5]Intrinsic Exchange Group, Washington, DC, USA. [6]Batker Consulting, LLC, Tacoma, WA, USA. [7]RFF-CMCC European Institute on Economics and the Environment, Fondazione Centro Euro-Mediterraneo sui Cambiamenti Climatici, Milan, Italy. [8]Politecnico di Milano, Milan, Italy. ✉e-mail: bbastien@ucsd.edu

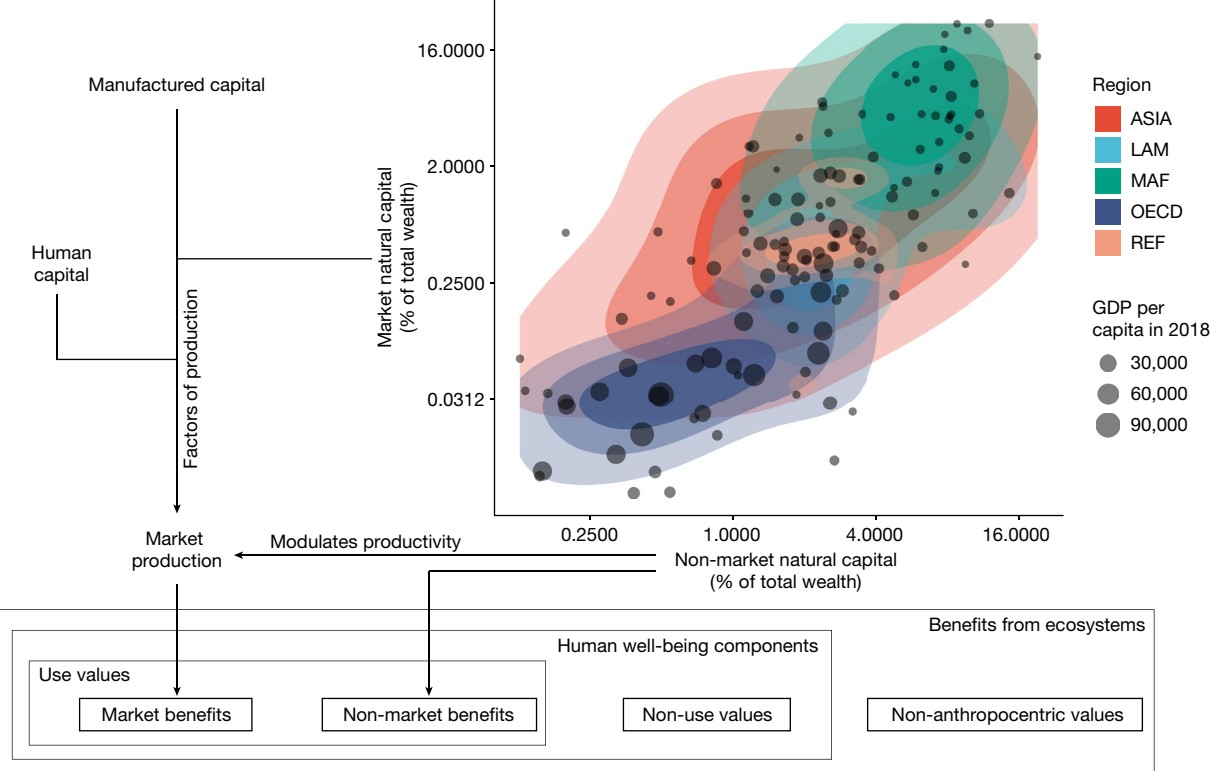

**Fig. 1 | Country-level natural capital by type and geographic region.** Natural capital contributes to market and non-market components of human well-being. Market natural capital interacts with manufactured and human capital to produce market benefits that are modulated by the regulating and maintenance services from non-market natural capital, which also generates non-market benefits (for example, cultural services). Points show country-level estimates of market and non-market natural capital as a fraction of total wealth (defined as the sum of natural capital, manufactured capital and human capital) using national accounts data from the World Bank. The five regions are: ASIA, Asian countries except the Middle East, Japan and the former Soviet Union states; LAM, Latin America and the Caribbean; MAF, the Middle East and Africa; OECD, the OECD 90 countries and the European Union member states and candidates; REF, the reforming economies of Eastern Europe and the former Soviet Union. Owing to limitations in the World Bank data, other benefits from ecosystems, such as non-use values and non-anthropocentric values, are not included in our measure of human well-being.

are especially vulnerable to any climate-change-induced loss of forest cover or changes in terrestrial vegetation patterns.

## Climate-induced biome range shifts

Changes in atmospheric carbon dioxide concentrations, temperature and water availability are already affecting forest productivity and functionality around the world[19,20]. Also, ecosystems are shifting poleward and towards higher elevations[21–24]. This paper focuses on the effect of ecosystem range shifts; the effects associated with total ecosystem area change; and changes in vegetation carbon content as a proxy for ecosystem overall health. We retrieve ecosystem cover projections under future climate-change scenarios from the LPJ-GUESS[25], ORCHIDEE-DGVM[26] and CARAIB[27] models, three process-based vegetation-terrestrial ecosystem models that simulate the establishment, growth, disturbance, competition and mortality of different types of natural vegetation, henceforth referred to as biomes.

Figure 2 shows the average present distribution of biomes across latitudes and future changes in this distribution, land cover and vegetation carbon at 2 °C of warming, using different representative concentration pathways (RCPs; RCP2.6, RCP6.0 and RCP8.5) and four Earth system model outputs, fixing land-cover and socio-economic variables at 2005 levels to isolate the sensitivity of terrestrial vegetation to climate change only[28]. Figure 2a shows the average grid-cell cover from 2016 to 2020 of each type of biome at different latitudes. The high vegetation abundance in both tropical regions and the high latitudes and the relatively low abundance in the arid midlatitudes form a trimodal distribution with a peak at the equator.

Figure 2b shows the average change of biome distributions for a world 2 °C warmer than the 2016–2020 average. Overall, biomes migrate poleward. Boreal biomes expand into high-latitude areas in which vegetation was not abundant previously and temperate biomes advance poleward. Another noteworthy pattern is the partial replacement of grasslands with forests. Grasslands show a net loss in tropical and temperate areas, with a corresponding expansion of forest biomes. In the tropics, in particular, the gain in forest areas is smaller than the loss of grasslands, leading to a net decline in vegetated areas. Figure 2c shows the percentage change in vegetation cover accounting for all biomes within a grid cell in a world 2 °C warmer. Most tropical regions are expected to see a net loss of natural vegetation-covered area, whereas higher latitudes and Central Asia will have net gains. Regardless of the changes in area cover, the whole world is projected to gain vegetation carbon content per square metre (Fig. 2d), which we include in our analysis, as it will probably affect the flow of ecosystem benefits to people (see Extended Data Figs. 4 and 5 for results under ORCHIDEE-DGVM and CARAIB, alternatives to our preferred model, LPJ-GUESS).

## Biome shift effects on natural capital

The World Bank data offer the advantage of a uniform accounting and valuation method for natural capital in all countries in the world. We use the VEGS database to apportion the natural capital values in 2018, calculated by the World Bank for each country, between biomes based on the area covered by each biome type within the country territory (Methods). Figure 3a shows the distribution of the

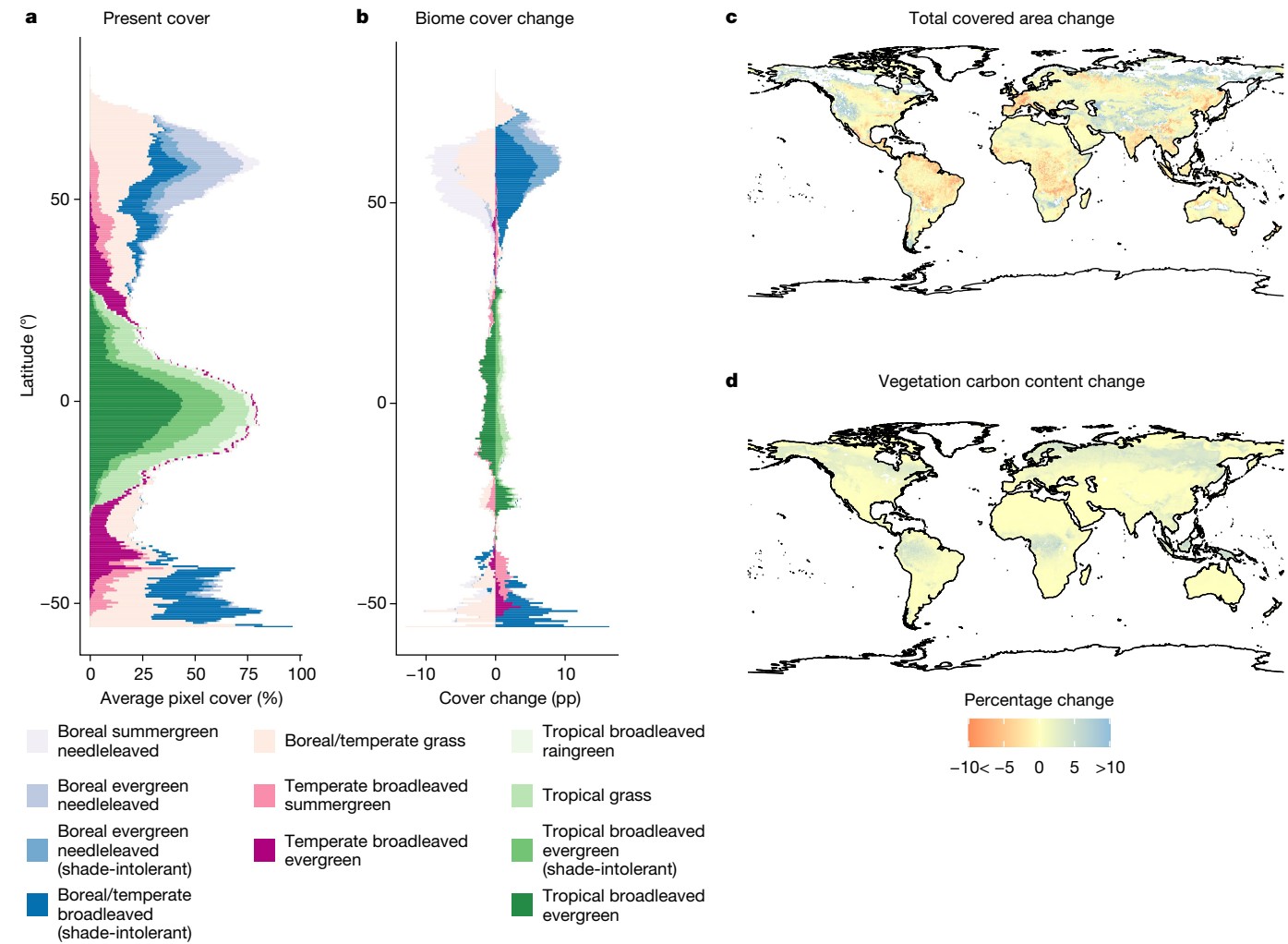

**a** Present cover

**b** Biome cover change

**c** Total covered area change

**d** Vegetation carbon content change

Percentage change

−10< −5   0   5  >10

Boreal summergreen needleleaved

Boreal evergreen needleleaved

Boreal evergreen needleleaved (shade-intolerant)

Boreal/temperate broadleaved (shade-intolerant)

Boreal/temperate grass

Temperate broadleaved summergreen

Temperate broadleaved evergreen

Tropical broadleaved raingreen

Tropical grass

Tropical broadleaved evergreen (shade-intolerant)

Tropical broadleaved evergreen

**Fig. 2 | Biome shifts and changes in area cover and vegetation carbon content. a**, Average percentage of grid cells covered by different biomes in the present (2016–2020). **b**, Change in coverage under 2 °C warming projections relative to the present day (using atmospheric forcings from different Earth system models and three RCPs). **c**, Changes in the fraction of a grid cell covered by natural vegetation. **d**, Changes in vegetation carbon content within each grid cell (kilograms per square metre). Model output from LPJ-GUESS under three warming scenarios and four climate model outputs (figures using the other two DGVMs are shown in Extended Data Figs. 4 and 5). World map from rnaturalearth package[45].

country-level estimates of market and non-market benefit values per hectare for each biome (see Extended Data Figs. 6 and 7 for results under alternative models). By multiplying the country-specific yearly benefits per hectare of each biome, which we allow to vary with biome size based on analysis of estimates from studies in the VEGS database (Extended Data Table 3), and the coverage area in each country, we get the total yearly benefits per biome. Figure 3b shows the biome-specific annual benefit flows for an average country in each geographic region. These annual flows of market and non-market benefits are used to calculate future natural capital values (Methods).

Using output from the DGVMs of projected changes in biome cover and the estimated proportional change in the per-area values in response to variations in total area and vegetation carbon, we calculate the country-level change in natural capital implied by shifts in terrestrial vegetation cover for each remaining decade over the twenty-first century. We fit a linear relationship between these decadal estimates and global temperature change during that decade (using output from four Earth system models run under three warming scenarios), giving us a damage function (equation (14) in Methods) relating changes in natural capital in each country to global temperature change (values shown in Supplementary Table 2).

## Damages to human well-being

Following previous work in ref. 11, we incorporate the country-level estimates of natural capital and their respective damage functions in the RICE50+ model, a regionally explicit integrated assessment model[18] that we use to estimate the effects of climate-driven biome range shifts on country-level market and non-market benefits (as measured by GDP). Specifically, we expand the economic production function to include natural capital as a source of raw materials and as a global environmental public good, providing regulating and maintenance services that modulate economic production, as proposed in the Dasgupta Review[16]. We call this model with regionally modified capital, production and damages 'Green' RICE50+. We run a baseline simulation with natural capital fixed at 2018 levels and economies growing following the SSP2 (SSP2 baseline, projecting temperature to increase by 3.8 °C by the end of the century). Figure 4 shows the changes in the market benefits (as measured by effects on annual GDP) and non-market benefits when climate impacts on natural capital derived from our analysis of climate-driven biome shifts are incorporated, relative to the baseline scenario.

We see varying degrees of distributional burdens concerning the climate impacts on the market and non-market benefits from natural

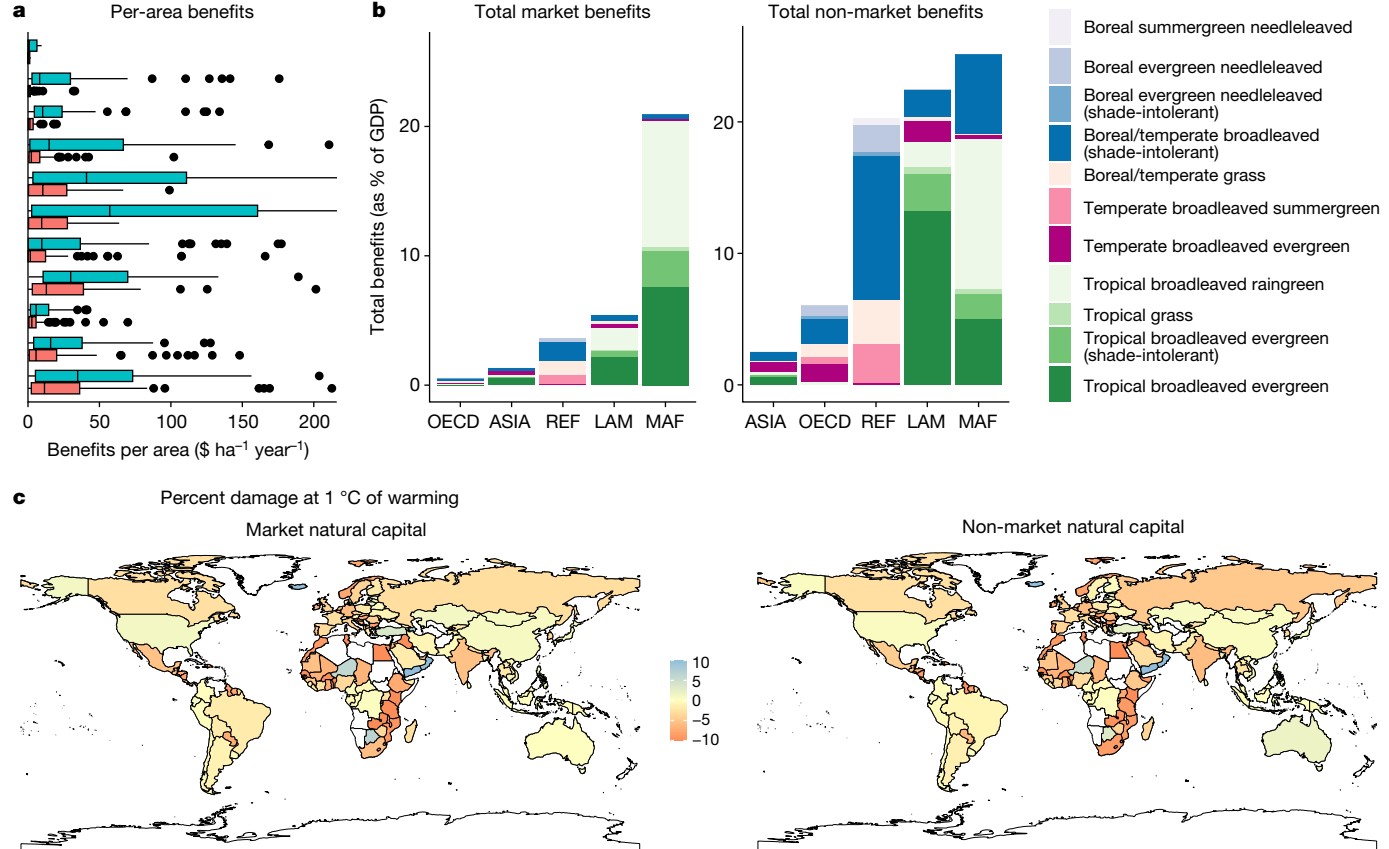

**Fig. 3 | Market and non-market benefits by country and biome. a**, Distribution of country-level benefits per hectare of biome, from boreal (top) to tropical (bottom; see colour key on the right). Red and blue boxes show market and non-market benefits, respectively. The middle line shows the median, the box covers the first to the third quartile and whiskers show the full range of the data, except for outliers shown by black points. **b**, Total yearly benefits per geographic region per biome. Note that values are reported as equivalent fractions of GDP as a comparison point only; non-market benefits are not captured in standard GDP accounting practices. The five regions are: ASIA, Asian countries except the Middle East, Japan and the former Soviet Union states; LAM, Latin America and the Caribbean; MAF, the Middle East and Africa; OECD, the OECD 90 countries and the European Union member states and candidates; REF, the reforming economies of Eastern Europe and the former Soviet Union. **c**, Changes in the market and non-market natural capital for 1 °C of warming (see Supplementary Tables 1 and 2 for coefficients and information on missing countries). Figures using the two other DGVMs are shown in Extended Data Figs. 6 and 7. World map from rnaturalearth package[45].

capital. First, 90% of the regions will experience losses in the value of their non-market benefits with respect to the undamaged baseline. The global population-weighted average change in non-market benefits value in 2100 is −9.2% relative to the baseline. Notably, when using damage functions based on CARAIB and ORCHIDEE-DGVM, the figures are 1.4% and −4.9%, respectively, highlighting the variation across different vegetation models (see discussion). Only five regions receive benefits from ecosystem shifts under a changing climate, as the ecosystems that they will gain more than offset the value of the ecosystems that they will lose. Changes in non-market benefits are not strongly correlated with the country's per-capita income or geographic region (Extended Data Fig. 9). Therefore, we find that, overall, shifting terrestrial vegetation patterns threaten both lower-income and higher-income countries.

Changes in the market benefits of terrestrial ecosystems, measured in terms of GDP loss over the baseline scenario without climate change, are smaller than the non-market impacts. The global population-weighted average change in GDP by 2100 is −1.3% of the baseline GDP (−0.7% and −1.4% using output from CARAIB and ORCHIDEE DGVMs, respectively). Compared with recent estimates of the economic impacts from global warming by 2100 of around 10% of GDP (refs. 29,30), this implies that roughly one-tenth of these total effects could be linked to the impact on shifting biome ranges. Moreover, these effects are unequally distributed across regions. The bottom 50% of the countries and regions, in terms of GDP per capita, bear approximately 90% of these damages,

whereas the top 10% only face 2% of these losses (Extended Data Fig. 11). These results indicate that ecosystem impacts of climate change could further increase economic inequality between regions, by damaging a stock of wealth that is particularly important in low-income and lower-middle-income countries[31].

## Discussion and conclusions

The effects of ecosystem damage owing to climate change on human well-being are still highly uncertain and depend on integrating insights from climate science, ecology, economics and other social sciences, which is challenging even at the regional level[6]. In this work, we approach this issue at a global scale to highlight the distributional impacts of these effects. To do so, we combine biome dynamics with the ecosystem-service-valuation literature and an integrated assessment model of climate and the economy. Integrating results from different disciplines is essential for addressing these complex questions, but it also comes with some limitations.

First, it is crucial to note that the ecosystem benefits assessed here represent only a fraction of nature's diverse contributions to people[1,32,33]. In collaboration with stakeholders, local communities and indigenous groups, an active field of research has developed a framework to better capture complex human–nature relationships[34]. This framework includes the intrinsic value of nature, the role of nature

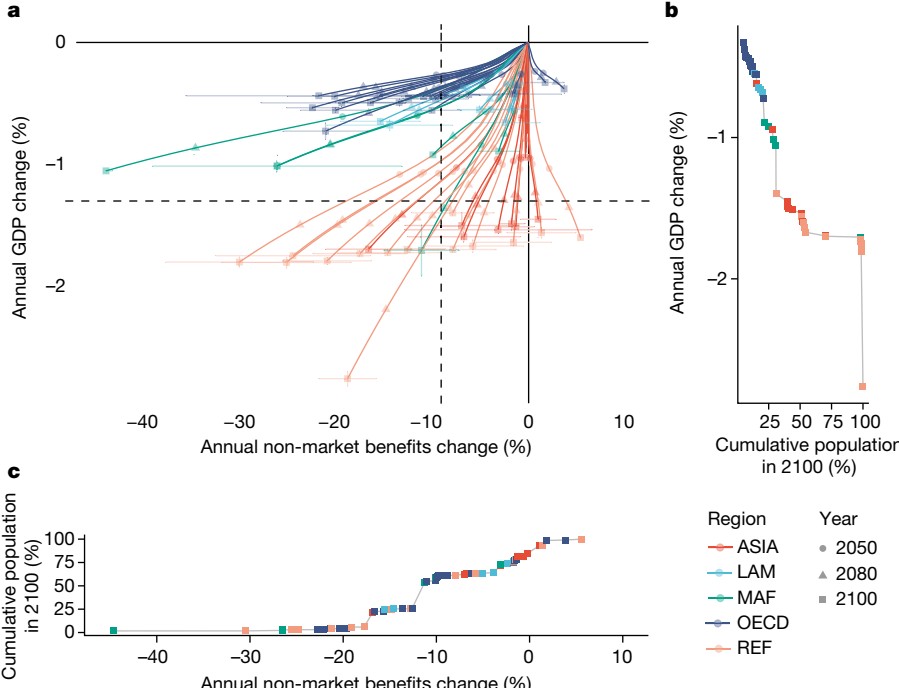

**Fig. 4 | Market and non-market impacts under the SSP2-6.0 scenario.**
**a**, Trajectories of the annual change in market benefits (shown as annual GDP change) and non-market benefits are shown for the 57 countries and regions in RICE50+ with respect to simulations without climate-change impacts, from now up to 2100. Error bars show the standard errors of the estimated damage functions under different GCM outputs. Dashed lines show the population-weighted mean values in 2100. A comparison of these results for different DGVMs with region labels is presented in Extended Data Fig. 8. **b**, Cumulative global population plotted against respective levels of annual GDP change in 2100. **c**, Cumulative global population plotted against respective levels of annual non-market benefits change in 2100. The five regions are: ASIA, Asian countries except the Middle East, Japan and the former Soviet Union states; LAM, Latin America and the Caribbean; MAF, the Middle East and Africa; OECD, the OECD 90 countries and the European Union member states and candidates; REF, the reforming economies of Eastern Europe and the former Soviet Union.

in our world views and its utility to society. This latter category is the main focus of our research and is further limited to such benefits quantified by previous work on natural capital accounting and the methods that have been used[12]. For instance, non-market values in protected areas are recorded as the value of foregone profit from intensively managed systems (such as agriculture), which may be substantially lower than the value of the suite of non-market benefits provided by these areas, including existence values for species and ecosystems, hence our loss estimates are conservative. This work would benefit from future efforts to improve the comprehensiveness and accuracy of natural capital accounts.

The ecosystems considered further limit the impact estimates. For market natural capital, the World Bank only accounts for forest rents. However, in our apportionment of natural capital values between biomes, we include grasslands, as evidence from the VEGS database shows that they provide important environmental benefits. Although this potentially allows a better representation of the benefit distribution between biomes, the market value in both forests and grasslands is probably greater than World Bank estimates. Furthermore, incorporating marine ecosystems could offer a fuller perspective on climate-change impacts on the benefits of nature to humanity.

Notably, this is the first study to implement the most recent model of economic growth proposed by the Dasgupta Review, which reflects the indirect regulating and maintenance role of global habitats in enabling economic growth, as well as providing raw materials. However, the lack of global assessments of regulation and maintenance services consistent with natural capital accounts limits our ability to accurately represent these values and their elasticities in the country-level economic production functions, causing a potential mismatch between theory and available data. Given this data gap, we use the global value

of protected areas and other provisioning and information services captured in the World Bank natural capital accounts. On one hand, this approach might overstate the actual value, as it encompasses the use values of environmental amenities that might not directly enable GDP. Conversely, this method could also underestimate the value by omitting aspects such as global benefits from climate services (see Extended Data Fig. 10 for a version of the results using a more standard economic growth model).

This analysis omits certain climate impacts on ecosystems. For instance, the simulations do not account for disturbances such as insect-driven tree mortality or wildfires, which substantially affect vegetation spatial patterns and the carbon cycle[35–38]. Although we factor in potentially offsetting effects resulting from gross primary productivity gains driven by higher atmospheric $CO_2$ concentration, studies have shown that the relationship between gross primary productivity and ecosystem functioning may not always be strong or consistent[39]. Also, climate change poses an extinction risk to a broad array of species that might play an important role in both ecosystem functioning and people's values of nature[33,40–42].

We focused on the welfare impacts of climate-driven changes in biomes, excluding the effects of habitat conversion from human activity, known to detrimentally affect ecosystem services and biodiversity[43,44]. Our estimated impacts should be seen as conservative given the expected land-use and land-cover changes over the rest of the century. This simplification could be relaxed by using the scenarios framework developed to assess climate-change policies. The SSP-RCP scenario matrix quantifies the levels of policy-led land-use change to comply with emissions, which—in some cases, such as expansion of biofuel crops in natural areas—might severely decrease biomes coverage. Also, the results based on DGVMs other than LPJ-GUESS, which generally

show smaller GDP and non-market benefits disruptions, should be interpreted as conservative, as the output used from these models provides proportional, not absolute, biome shifts, thus representing the effect of biome replacement without fully considering area change as with LPJ-GUESS.

In this research, we show that climate-driven biome range shift will generally lower terrestrial non-market benefits globally, with net gains in only a few areas. Overall, even with our conservative estimates, the estimated market losses are in the range of 10% of the total economic damages from climate change. Moreover, these losses will be borne mostly by countries from the Global South because of the relative importance of natural capital in their nation's wealth and economic production. Given the limited contribution of these regions to climate change, these market and non-market losses can further inform discussions about responsibilities and remediation. Notably, the findings from this research underscore the importance of formulating integrated climate policies that recognize and account for the unique natural capital of each country.

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

## Methods

### Study design

By bringing together complementary methodologies, this study analyses the effects of changing ecosystems under climate-change scenarios on future human well-being. We structure our research methodology into three sequential phases, as illustrated in Extended Data Fig. 1. The first phase involves estimating country-specific values per hectare of the main biomes found in each country. The biomes included in our study are based on a set of plant functional types (PFTs) relating to forest, grassland and desert ecosystems. In the second phase, we generate future projections of natural capital based on biome projections. Finally, in the third phase, we incorporate the dynamics of natural capital into an integrated assessment model. This section offers a high-level overview of each phase of our analysis to provide a comprehensive understanding of our study design; detailed explanations of each individual step are presented in subsequent sections of this paper.

In the first phase (Extended Data Fig. 1a), we extend the VEGS database[17] to incorporate estimates of biome cover and vegetation carbon content at all study locations, drawing from three DGVMs. For each country, we select a subset of the expanded VEGS database, based on studies in VEGS that exhibit similar biome distributions to the average biome coverage of the country. Next, we apply a random forest methodology to this selected subset to obtain the relative contribution of each biome in providing market and non-market benefits. Finally, we distribute country-level market and non-market natural capital estimates from the World Bank across the different biomes within the country, based on the relative importance of biomes estimated through the random forest analysis.

In the second phase (Extended Data Fig. 1b), we use country-specific values per hectare of each biome to project total market and non-market natural capital values. This projection is based on the future distributions of biomes from three DGVMs under four RCPs and four general circulation models (GCMs). The biome value per hectare, derived from the initial phase, is adjusted according to the total area of the biome cover and its vegetation carbon content. Last, we obtain the 'damage functions' by applying linear regressions to each natural capital trajectory to estimate the mean impact of global temperature rise on each country's natural capital.

In the final phase of our study, we incorporate the damage functions derived from the previous stage and the World Bank's country-level natural capital estimates into the RICE50+ model to capture endogenous economic growth with natural capital (Extended Data Fig. 1c). This expanded version of the RICE50+ model[18], now termed 'Green' RICE50+, allows us to examine potential feedback loops and interactions that might not be visible in a partial equilibrium context (such as the second phase of the study design). For instance, the accumulation of manufactured capital through the savings rate now also depends on natural capital, as it underlies economic production. This coupled approach paves the way to test and model various experiments in the future using RICE50+ in its full capacity, enabling us to model diverse scenarios and policies, including different levels of cross-country cooperation on climate, as well as on the issues of ecosystem services and biodiversity conservation, natural capital restoration investment, emissions optimization and general 'beyond GDP' policies, among others. In particular, we provide the necessary data and open-source code to integrate ecosystems in cost–benefit integrated assessment models.

### DGVMs output

We retrieve output data from the LPJ-GUESS[25], ORCHIDEE-DGVM[26] and CARAIB[27] models, the three DGVMs that participated in the Inter-Sectoral Impact Model Intercomparison Project (ISIMIP) 2b Protocol[28] under the simulation exercise 2005soc. These simulations consist of fixing land-use to 2005 conditions and simulating the response of the biomes to variables such as precipitation, daily maximum and minimum temperature, short-wave downwelling radiation, surface air pressure, near-surface relative humidity, near-surface wind speed and carbon dioxide concentration, allowing us to disentangle the climate-driven effects on natural vegetation from the direct anthropogenic disturbances.

Notably, these models use the concept of PFTs, which group plant species by their common characteristics and responses to environmental factors[46]. Specifically, LPJ-GUESS, ORCHIDEE-DGVM and CARAIB simulate the global dynamics of 11, 10 and 26 PFTs, respectively. However, of all the models, only LPJ-GUESS provides readily available output data that simulates both biome replacement and total biome cover change at the grid-cell level over time, making it the preferred choice for the main analysis. Results from other models are presented in Extended Data Figs. 4 and 5.

PFTs group together plant species sharing similar characteristics, such as comparable responses to environmental conditions, similar physiological traits and shared roles in ecosystem function[47]. By contrast, biomes are large geographic areas characterized by specific types of dominant vegetation and climate conditions. In many instances, the association between a biome and PFT category is self-evident; for instance, the 'temperate needleleaved evergreen' PFT typically dominates the 'temperate evergreen forest' biome.

However, certain biomes, such as deserts, do not straightforwardly correspond to a PFT category. In our analysis, though, these are represented indirectly, as they predominantly consist of C4 grasses, which fall within the PFT classifications of the DGVMs we use. We acknowledge that the terms PFT and biome are not perfect substitutes, but for the purpose of this study, we use them interchangeably.

Further details about the GCMs (HadGEM2-ES (ref. 48), GFDL-ESM2M (ref. 49), IPSL-CM5-LR (ref. 50) and MIROC5 (ref. 51)) and warming scenarios used in our DGVM simulations are provided in Extended Data Table 1. From each simulation and DGVM, we retrieve the annual percentage biome-covered area and carbon vegetation content at each $0.5° \times 0.5°$ pixel.

### Natural capital

Natural capital is part of society's productive base, producing flows of market and non-market benefits. Following Dasgupta[52], the total natural capital stock at time $t$ can be written as the sum of the quantity of individual natural assets (soils, forests, wetlands etc.) multiplied by their shadow price

$$N_t = \sum_i p_{t,i} n_{t,i} \qquad (1)$$

The shadow price, $p_{i,t}$, gives the present value of the stream of benefits derived from an extra unit of the natural asset $n_i$. Therefore, equation (1) can be written as

$$N_t = \sum_i \sum_{\tau=t}^{T} \frac{B_{i,\tau}}{(1+r)^{\tau-t}} \qquad (2)$$

in which $B_{i,\tau}$ is the total yearly flow of benefits from the natural asset $i$ at time $\tau$ and $r$ is the consumption discount rate. For benefits not traded in the market, measuring their values is particularly challenging, as doing so requires the application of non-market valuation approaches. Projecting estimates of this value in the future adds further complexity, as it involves assumptions over future management of natural assets and how preferences of the population might change, either exogenously or as a function of the asset stock itself[53].

As the role of natural capital becomes increasingly evident, both the World Bank and the UN Environment Programme have been developing approaches to incorporate natural capital accounting into national accounts data[12,54–56]. This literature has disaggregated natural capital valuation based on whether the stock is renewable and whether the flow

of benefits is traded in the market. We use estimates of natural capital values from the World Bank Changing Wealth of Nations 2021 report[12], excluding natural capital values related to non-renewable resources (for example, minerals), non-terrestrial ecosystems (for example, fisheries), as they cannot be modelled by DGVMs, and cropland, whose impacts from climate change have been extensively reviewed in the past[57–60]. Instead, we focus on natural capital provided by natural terrestrial ecosystems.

The World Bank uses diverse economic concepts and frameworks to estimate natural capital. On the one hand, natural capital associated with market-based benefits from forests is estimated using the present value of future timber revenues. On the other hand, natural capital embedded in forests and protected areas associated with non-market ecosystem benefits is estimated using the ecosystem services framework and the concept of opportunity cost (that is, the benefit from ecosystem services must be at least as large as the foregone value of alternative economic activities for biomes to remain intact on the landscape). Specifically, the World Bank conducts a meta-analysis of recreation services, water quality, water quantity and non-timber forest products and uses the results to estimate these services in each country. Further, the value from protected areas is estimated by the unrealized revenue had the areas been converted to agricultural fields, giving a lower-bound estimate of the value of the protected areas[12,55,56]. Hereafter, the terms market and non-market natural capital will refer only to the benefits described above, as it is the available data. However, the methodology remains valid for future additions to natural capital estimates and the results of this work should be considered as likely lower-bound estimates. In summary, we call market natural capital (mN) the estimate of natural capital provided by timber products of forests. Non-market natural capital (nN) is given by non-timber benefits from forests that offer use values (recreation, water quality and quantity, and other non-wood forest products) and the value of protected areas.

Following the World Bank assumptions of no forest area change in the future and constant per hectare value of benefits, we can rewrite equation (2) using the formula for the present value (PV) of a perpetuity (PV = FV/$r$), in which FV is the constant future value obtained each year and $r$ is the discount rate, which we set at 3% following the World Bank methodology. Therefore, the non-market natural capital value calculated in year $t$ is given by

$$nN_{t,c} = \frac{A_{t,c} ES_{t,c}}{r} \tag{3}$$

in which $ES_{t,c}$ is the per-area value of ecosystem benefits estimated in year $t$ for country $c$ and $A_{t,c}$ is the area covered by the ecosystem in year $t$ for country $c$. To calculate $nN_{t,c}$ for a given year using the perpetuity formula, we have to assume that the numerator in equation (3) remains constant over time, but we allow these terms to vary for $nN_{t,c}$ estimates across time. For example, we will use constant $ES_{2020,c}$ and $A_{2020,c}$ values to calculate $nN_{2020,c}$ that will differ from the constant values $ES_{2030,c}$ and $A_{2030,c}$ used to calculate $nN_{2030,c}$. Both of these factors can change in the future, as we allow the spatial extent of the ecosystem to change and also allow the per-area benefits to change with the total extent and the vegetation carbon (see the 'Mechanisms that affect the value per hectare' section).

Disaggregating non-market benefits and areas by types of biome ($b$), we get

$$nN_{t,c} = \sum_b \frac{a_{t,c,b} es_{t,c,b}}{r} \tag{4}$$

in which $es_{t,c,b}$ is the value of the non-market benefits flow provided by a hectare of biome $b$ in country $c$ in the year of the calculation $t$ and $a_{t,c,b}$ is the area coverage of biome $b$ in country $c$ in the year of the calculation $t$. Similarly, the market natural capital value is given by

$$mN_{t,c} = \sum_b \frac{a_{t,c,b} R_{t,c,b}}{r} \tag{5}$$

in which $R_{t,c,b}$ is the value of timber products provided by a hectare of each biome $b$ in country $c$ estimated at time $t$. To be able to use equations (4) and (5) to estimate future natural capital based on the area of the biomes, we first need to estimate $es_{t,c,b}$ and $R_{t,c,b}$ and assume that those marginal values will mostly remain constant in the future (see the 'Mechanisms that affect the value per hectare' section). To do that, we focus on the year 2018, which is the year for which the most recent country-level estimates of market and non-market natural capital are available from the World Bank. Also, we can retrieve the biome-covered area from the three DGVMs for that year.

Notably, we assume that the benefits derived from non-vegetated areas are zero in our methodology, as we apportion 100% of the benefits among the PFTs. Although non-vegetated areas could play ecological roles and potentially provide benefits such as recreation, runoff generation and soil stabilization (desert biological crusts, such as lichens and cyanobacteria), these specific values have not been factored into the estimates of the World Bank, which primarily focus on the value of benefits related to forests. We believe that this is a pragmatic assumption for our study, but we want to make it explicit here.

The World Bank data offer the advantage of a uniform accounting and valuation method for natural capital in all countries in the world. Although there are several well-established methods available to value non-market environmental amenities[61,62], these methods rely on expertise and resources that are generally unavailable in many contexts, leading to data gaps for many non-market environmental amenities and parts of the world, even in the comprehensive VEGS database. As a result, approaches for transferring value estimates from study sites to policy locations have been developed, although there are some challenges[63–65]. Our use of the World Bank data offers an alternative to the benefit-transfer approach, although our research question of interest requires us to adopt an attribution method to allocate the country-level values in this dataset to the various biomes within a country.

In the following sections, we show how we use an ecosystem-service-valuation database and a random forest algorithm to estimate $es_{2018,c,b}$ and $R_{2018,c,b}$, the attribution method.

### VEGS database

To obtain $es_{2018,c,b}$ and $R_{2018,c,b}$ in equations (4) and (5), we use the VEGS database[17]. The database curates findings and the contextual parameters that factor ecosystem service production into a standardized value per hectare per year, attributed to specific geographies. It includes 21 different ecosystem service types based on De Groot's framework. This database records annual values per hectare of 4,300 ecosystem services. However, we only use a subset that excludes meta-analyses, value-transfer studies, crop-valuation studies and observations that are pending revision. Therefore, we end up with a dataset ($V$) of 882 original estimates of ecosystem service value per hectare from 118 studies across 60 countries (Supplementary Table 3).

Also, we retrieve the georeferenced spatial boundaries of the reported locations of the studies and used them to extract the gridded GDP (ref. 66) and the mean biome-covered area percent and vegetation carbon stock for each observation from three DGVM outputs from 2016 to 2020. The variables in $V$ relevant to this study are listed in Extended Data Table 2. A version of the database with the necessary variables to replicate this study is available in the repository of the model (https://doi.org/10.5281/zenodo.8303029).

To obtain the values per hectare $es_{2018,c,b}$ and $R_{2018,c,b}$ for each country $c$, we take a subsample of the VEGS database ($V$), denoted as $V_b$ that contains observations from locations whose biome-cover percentage areas are similar to the values of the country. To do this, we obtain the Euclidean distance between observations in the VEGS database and the mean biome values of the country. We further divide the subsets

into non-market values to estimate $es_{2018,c,b}$ and $R_{2018,c,b}$, respectively. We denote these subsets $V_{i,b}$, in which $i = m, n$ for market and non-market values, respectively.

### Random forest

We use the database subsets $V_{i,b}$ to train random forests ($RF_{i,b}$) for each country. A random forest consists of a collection of decision trees that predict a dependent variable using a series of optimal subdivisions in the data. We build our random forests to predict the log value of the ecosystem benefits per hectare in $V_{i,b}$ based on the variables listed in Extended Data Table 2, such that

$$V_{i,b} = RF_{i,b}(\text{cover}_b, \text{GDPpc}, \text{PercCovered}) \tag{6}$$

We create random forests with 300 decision trees, as the decrease in the root mean squared error by each extra decision tree in the random forest reaches saturation at that point. We create one random forest for each combination of 177 countries, two types of benefit (market and non-market) and three typologies of biomes (from the three DGVMs), giving 1,062 random forests.

We use the random forests to predict a baseline value of market and non-market benefits per hectare. Next, we predict how these per-hectare values change when the extent of each biome is increased by 10 percentage points (pp), holding all other biome areas constant. We use the relative sizes of the resultant per-hectare values as indicators of the relative importance of each biome to the natural capital stocks of the country, using these values to estimate the contribution of each biome to the total natural capital estimates for a given country.

To further explain the procedure without loss of generality, we imagine a hypothetical country with only two biomes: B1 and B2. Using the random forest generated for that country, and focusing on non-market natural capital, we obtain the new values per hectare, $\widehat{es}_{B1}$ and $\widehat{es}_{B2}$, by increasing biomes B1 and B2 cover 10 pp, respectively:

$$\widehat{es}_{B1} = RF_{n,b}(\text{cover}_{B1+10}, \text{cover}_{B2}, \text{GDPpc}, \text{PercCovered}) \tag{6a}$$

$$\widehat{es}_{B2} = RF_{n,b}(\text{cover}_{B1}, \text{cover}_{B2+10}, \text{GDPpc}, \text{PercCovered}) \tag{6b}$$

From the values above, we can obtain the parameter $x_{B1}$, a scaling factor to express $\widehat{es}_{B2}$ in terms of $\widehat{es}_{B1}$, so that $\widehat{es}_{B2} = x_{B1}\widehat{es}_{B1}$. Assuming that this relationship holds for comparing two hectares fully covered by biomes B1 and B2, respectively, we can write the value of non-market benefits provided by one hectare of biome 2 ($es_{2018,c,B2}$) in terms of the value of non-market benefits provided by one hectare of biome 1 ($es_{2018,c,B1}$) as follows:

$$es_{2018,c,B2} = x_{n,c,B1}es_{2018,c,B1} \tag{7}$$

Rewriting the equation for non-market natural capital (equation (3)) for the hypothetical country in 2018,

$$nN_{2018,c} = \frac{a_{2018,c,B1}es_{2018,c,B1} + a_{2018,c,B2}es_{2018,c,B2}}{r} \tag{8}$$

$$\Rightarrow r \times nN_{2018,c} = a_{2018,c,B1}es_{2018,c,B1} + a_{2018,c,B2}es_{2018,c,B2} \tag{9}$$

Substituting $es_{2018,c,B2}$ from equation (7),

$$r \times nN_{2018,c} = a_{2018,c,B1}es_{2018,c,B1} + a_{2018,c,B2}x_{n,c,B1}es_{2018,c,B1} \tag{10}$$

$$\Rightarrow es_{2018,c,B1} = \frac{r \times nN_{2018,c}}{a_{2018,c,B1} + a_{2018,c,B2}x_{n,c,B1}} \tag{11}$$

All of the variables on the right-hand side of equation (11) are known, so we can estimate $es_{2018,c,B1}$. Similarly, we estimate $R_{2018,c,b}$. In the following section, we discuss mechanisms that can change the per-area benefits $es_{2018,c,b}$ and $R_{2018,c,b}$ when using them to calculate natural capital in future time steps.

### Mechanisms that affect the value per hectare

As well as the change in biome cover, we test two further mechanisms that could change the value of market and non-market natural capital.

Using the VEGS database, we test two hypotheses: first, we test whether the marginal value of non-market benefits $es_{t,c,b}$ varies with the area of the biome. Although this step is critical, given that we are valuing non-marginal changes in the natural capital stock and supply of related non-market benefits, this is an improvement over much of the literature in this area. Using the values from the VEGS database, we observe that the marginal value per hectare of non-market ecosystem benefits exhibits diminishing marginal utility (the elasticity is −0.103, $P < 0.001$), whereas market ecosystem benefits have a constant marginal value, so the per-area market benefits $R_{t,c,b}$ do not depend on biome size (Extended Data Table 3).

Also, we estimate the effect of an increase in the percentage of vegetation carbon content on the per-area benefits from ecosystems. Our regression results (Extended Data Table 3) show a low elasticity of non-market benefits: for a 1% increase in vegetation carbon stock per hectare, the non-market value provided by ecosystems increases by 0.282%. No effect was found for market benefits. Assuming no changes in country-level preferences, we estimate future non-market natural capital in year $t$ using the equation

$$nN_{t,c} = \sum_b \frac{a_{t,c,b}es_{c,b} \times (1 + (0.282\dot{c}_t - 0.103\dot{a}_t)/100)}{r} \tag{12}$$

in which $\dot{a}$ is the percentage change in total area and $\dot{c}$ is the percentage change in vegetation carbon content. However, for very large changes in vegetation carbon content or area, these coefficients become less reliable, as the data points begin to exceed the range of our observations and the changes are no longer marginal. To maintain the robustness of our model, we cap the vegetation carbon and area changes effect at a level corresponding to twice the range of our data, shown in Extended Data Fig. 2. This conservative approach allows ecosystems with future higher carbon content to potentially enhance non-market benefits, preventing overestimation of climate change damages.

The equation we use to calculate market natural capital only changes as a function of the biome-covered area:

$$mN_{t,c} = \sum_b \frac{a_{t,c,b}x_b\hat{R}_c}{r} \tag{13}$$

It is important to note that equations (12) and (13) assume that the agent that estimates market and non-market natural capital amounts uses the information available in year $t$ about values per hectare and areas of biomes to estimate natural capital in that year, following the World Bank assumptions of time-invariant benefits and areas. Later in the study, growing income per capita and manufactured capital in each country play an important role in obtaining the flow of market and non-market benefits.

### Damage function

We use the 28 model runs described in Extended Data Table 1 and equations (12) and (13) to generate decadal point estimates relating country-level natural capital values and global temperature changes. Our stylized damage function models the change in natural capital as a function of temperature change. We choose a linear function that captures the relationship fairly well and avoids non-convex damages and overly high damages estimated with a low level of confidence outside the observed range. Therefore, we have that

$$nN_{t,c} = nN_{0,c}(1 + \theta_{n,c}\Delta T_t) \tag{14}$$

Normalizing $nN_{t,c}$ by the initial natural capital value $nN_{c,0}$

$$\widehat{nN}_{t,c} = 1 + \theta_{n,c}\Delta T_t \tag{15}$$

Therefore, the relative change in the normalized value is

$$\Delta\widehat{nN}_{t,c} = \theta_{n,c}\Delta T_t \tag{16}$$

We use the 196 points (28 model outputs across seven decades) to fit the following equation to obtain the damage coefficient for each DGVM

$$\Delta\widehat{nN}_{t,c,\text{dgvm}} = 0 + \theta_{n,c,\text{dgvm}}\Delta T_t + \alpha_{\text{clim}} + \alpha_{\text{scen}} + \epsilon_{t,r} \tag{17}$$

in which $\alpha_{\text{clim}}$ and $\alpha_{\text{scen}}$ are fixed effects controlling for the four GCMs and the three RCPs, respectively. Similarly, we obtain $\theta_{n,c,\text{dgvm}}$ for the market natural capital. As shown in Extended Data Fig. 3, most of the countries have damage estimates with $P$-values lower than 0.01.

## Implementation in the RICE50+ model

We project losses in GDP and natural capital from 2015 to 2100 across different regions of the globe using an extended and recalibrated version of the open-source RICE50+ integrated assessment model[18]. The model is augmented to include the market and non-market value of natural capital in the production function and damages to natural capital from warming. As it is designed in a modular and integrative fashion, the natural capital module can be combined with all other parts and (future) extensions of the model. We also implemented a welfare function that incorporates natural capital, following the specification of the Green-DICE model[11], which can be combined and activated for applications to endogenous policy choices, which is beyond the scope of this paper.

GDP is computed for each region $i$ with a Cobb–Douglas production function of labour, $L_{t,c}$, manufactured capital, $K_{t,c}$, market natural capital, $mN_{t,c}$, with total factor productivity, $TFP_{t,c}$, modulated by a global value environmental good that provides maintenance and regulation services in which the economy is embedded $S_t = \sum_c nN_{t,c}$, following the Dasgupta Review[16]. Besides manufactured capital, labour and TFP, market natural capital is added as a region-specific or country-specific production factor, whereas a global level of non-market natural capital is assumed, taking into account its global public good nature:

$$GDP_{t,c} = TFP_{t,c} \times S_t^{b_r} \times L_{t,c}^{\gamma_{1c}} \times K_{t,c}^{\gamma_{2c}} \times mN_{t,c}^{\gamma_{3c}} \tag{18}$$

TFP and labour are exogenous and calibrated to match the population and economic output from SSP2. Notably, equation (18) is a non-traditional approach usually not found in neoclassical economics that we adopted from the Dasgupta Review, which adds a factor ($S^b$) that represents the extent to which market production is possible owing to all the life-maintaining mechanisms of the Earth system. However, owing to the lack of a global measure of such services consistent with the World Bank accounts, we use the global aggregate of non-market natural capital as a proxy for it. This factor, along with the standard TFP, could be understood as an adjusted TFP. Proxying for $S$ with the global aggregate of non-market natural capital may, on the one hand, overstate the true value, as it includes use values of environmental amenities that may not be a direct enabler of GDP, and, on the other hand, potentially understate $S$, as it misses, for instance, the global benefits from climate services.

The degree to which some non-market natural capital produces flows of goods and services that do or do not increase market production is reflected in the value of $b$, which varies by region. The higher is $b$, the more non-market natural capital plays a role in the production of goods and services that are priced and increase economic output. Mathematically, $b$ is the elasticity of GDP growth with respect to global

non-market natural capital $S$. For example, a stable and predictable climate reduces uncertainty over future investments in, say, agriculture, making the latter cheaper and more abundant. This, in turn, results in higher economic output, although only a fraction of the benefits stemming from it are reflected in market prices. We acknowledge that this specification is not common ground in neoclassical economics and therefore we show in Extended Data Fig. 10 a version of our results assuming $b = 0$.

The production elasticity of labour, $\gamma_1$, and the regionally calibrated elasticity of the global environmental good, $b_r$, are estimated by the following GDP-weighted panel regression model, noting that, instead of labour, in this regression model, we use human capital ($H$) as estimated by the World Bank to be consistent with the other capital estimates ($mN$ and $nN$):

$$\log(GDP_{t,c}) = R_c \times \hat{b}_r\log(S_t) + \hat{\gamma}_1\log(H_{t,c}) + \hat{\gamma}_2\log(K_{t,c})$$
$$+ \hat{\gamma}_3\log(mN_{t,c}) + \theta_t + \theta_c + \epsilon_{t,c} \tag{19}$$

in which $\mathbf{R}$ is a vector of dummy variables for each of the five high-level macro world regions: OECD (the OECD 90 countries and the European Union member states and candidates); LAM (Latin America and the Caribbean); REF (the reforming economies of Eastern Europe and the former Soviet Union); ASIA (Asian countries except the Middle East, Japan and the former Soviet Union states); and MAF (the Middle East and Africa); $\theta_t$ and $\theta_c$ are year and country fixed effects, respectively, and standard errors are clustered by region. The results are shown in Extended Data Table 4 and imply a capital share of around 27% and a labour share of around 50% in total GDP, whereas market natural capital is estimated on average with an output global elasticity of 0.004. The region-specific impact of global non-market natural capital varies between 0.15 and 0.61, in the range of values considered in the Dasgupta Review, and shows the potential importance of natural capital for economic output. The RICE50+ model implements this production function and calibrates TFP to match the SSP2 baseline trajectory, so that the baseline GDP path with and without the inclusion of natural capital is not affected. The global value $S_t$ is obtained by the regional sum of $nN_{c,t}$ and takes into account damages computed on the basis of the preceding section.

Following work in ref. 67, we use the country-level share of timber rents on GDP as the production elasticity to market natural capital $\gamma_{3,c}$. The production elasticity to manufactured capital is obtained assuming constant returns to scale of the three neoclassical factors of production, that is, $\gamma_{2,c} = 1 - \gamma_{1,c} - \gamma_{3,c}$. Market and non-market values of natural capital are sensitive to global mean temperature as described by equation (14). GDP losses are computed by comparing a model with damages to natural capital under the SSP2-6.0 scenario to a baseline without such damages.

Finally, we obtain the annual flow of non-market benefits by using the formula of the net present value of a benefit flow in perpetuity, that is, multiplying the non-market natural capital by the discount rate (3% as a chosen value) and allowing ecosystem services to increase based on the country's percent increase on GDP per capita (%ΔGDPpc) times the income elasticity obtained in Extended Data Table 3. Therefore, $ES_{t,c} = nN_{t,c} \times 0.03 \times (1 + 0.00596 \times \%\Delta GDPpc_{tc})$.

## Data availability

Code and data to replicate the analysis, results and figures are publicly available at https://doi.org/10.5281/zenodo.8303029. Source data are provided with this paper.

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

**Acknowledgements** The authors thank U. Pascual for comments on an earlier version of this manuscript. Funding: National Science Foundation grant 1924378 (F.C.M., M.N.C., X.D., B.A.B.-O.); European Union's Horizon Europe research and innovation programme (PRISMA) grant no. 101081604 (J.E., F.G., M.T.).

**Author contributions** The project roles were distributed among the team as follows: B.A.B.-O. and F.C.M. conceptualized the project. The methodology was developed by B.A.B.-O., M.N.C., X.D., M.T., J.E., F.G. and F.C.M. Investigation was carried out by B.A.B.-O., M.N.C., X.D., T.B., D.B. and F.C.M. B.A.B.-O. took charge of visualization. The funding was acquired by M.N.C., X.D., J.E. and F.C.M. F.C.M. administered the project and also provided supervision. The original draft was penned by B.A.B.-O., whereas the review and editing were collaboratively done by B.A.B.-O., M.N.C., X.D., T.B., D.B., J.E., M.T., F.G. and F.C.M. B.A.B.-O., T.B., D.B., J.E. and F.G. were in charge of data curation.

**Competing interests** The authors declare no competing interests.

**Additional information**
**Correspondence and requests for materials** should be addressed to B. A. Bastien-Olvera.

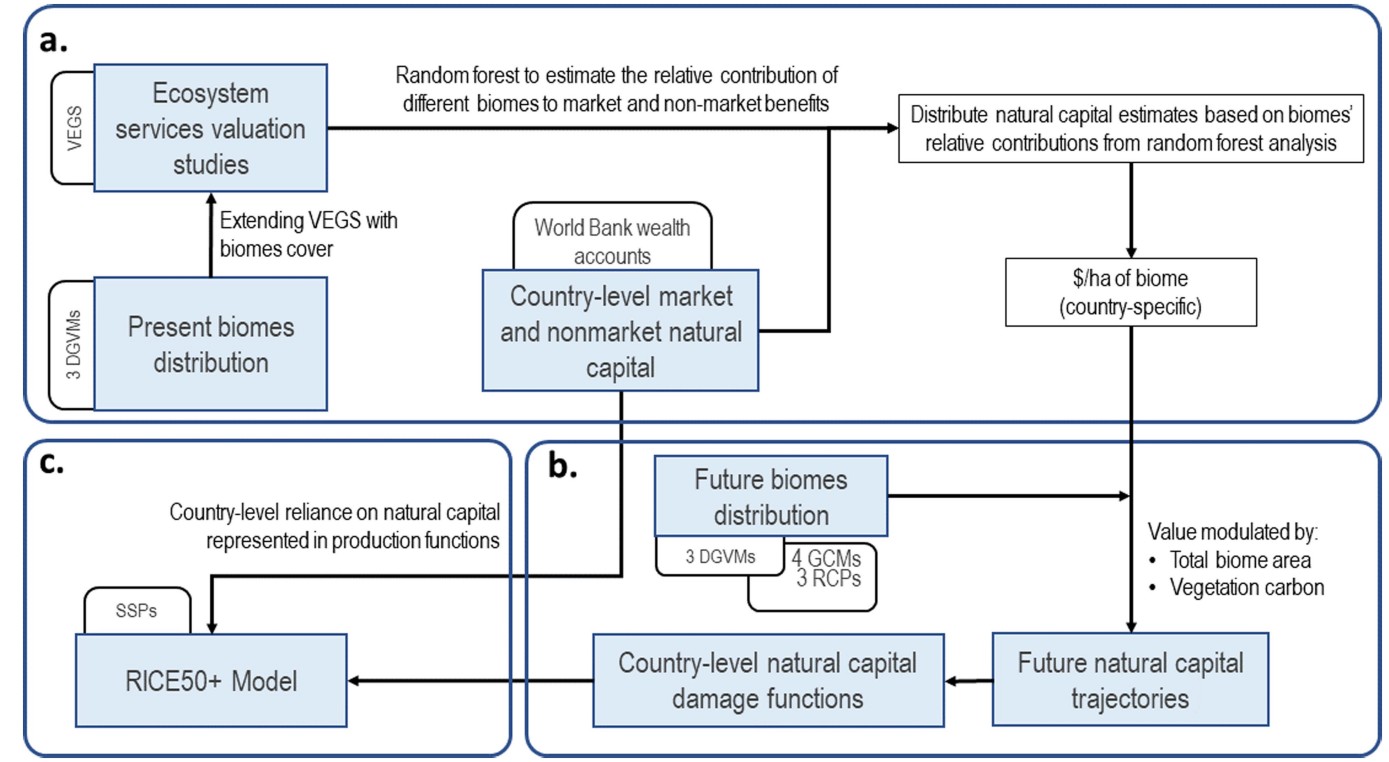

**Extended Data Fig. 1 | Study design. a**, Obtaining country-level values per hectare of biome. **b**, Estimating natural capital damage functions. **c**, Extending RICE50+ integrated assessment model.

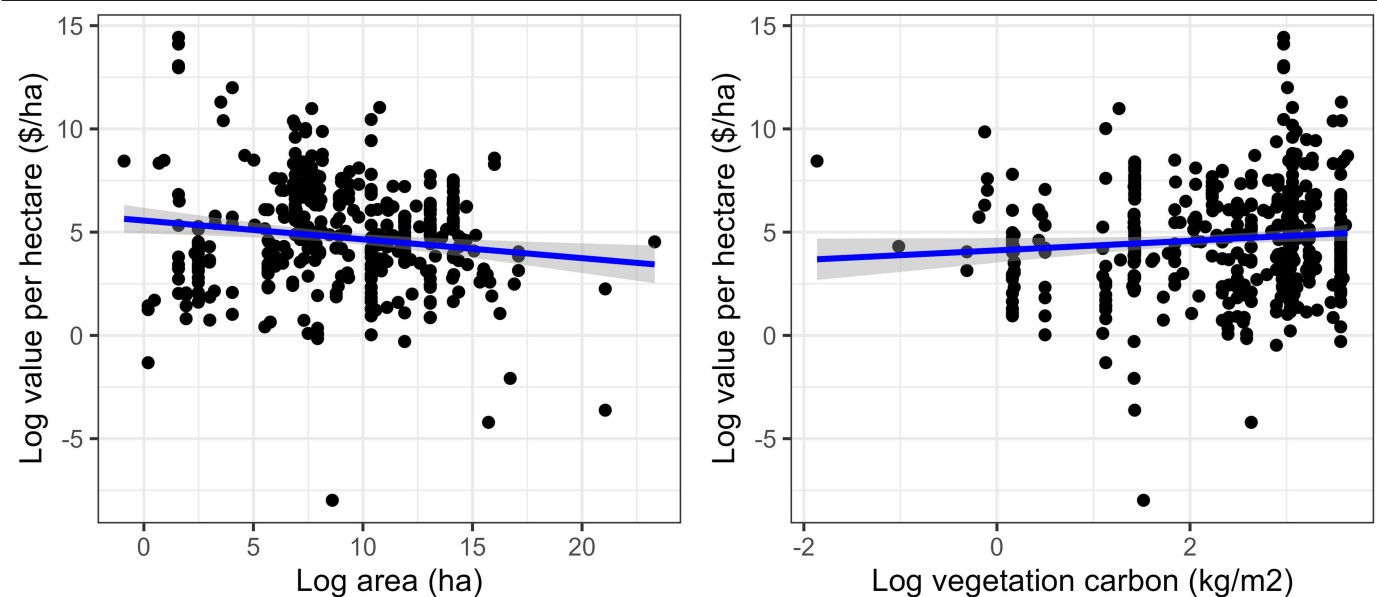

**Extended Data Fig. 2 | Change in marginal value for non-market benefits in the VEGS database.** Left, change in marginal benefits by area. Right, change in marginal benefits by vegetation carbon content. The solid line shows the fit of a linear model with a 95% confidence interval.

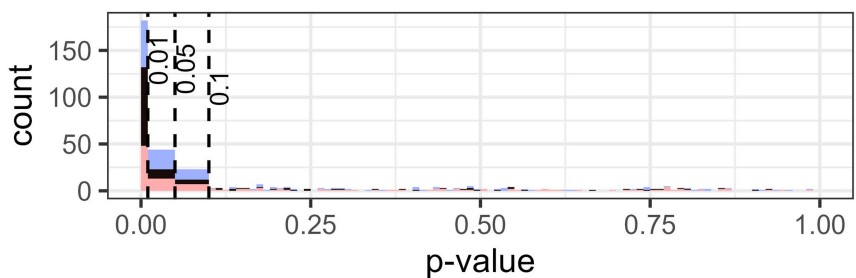

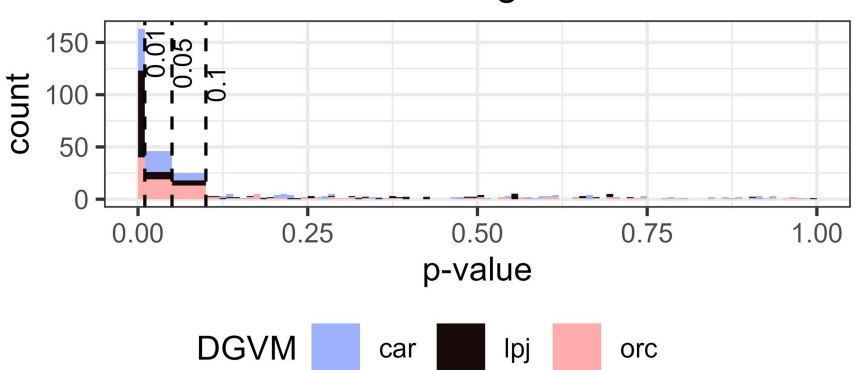

**Extended Data Fig. 3 | *P*-values of the market and non-market natural capital damage coefficients.** The theta estimates are reported in Supplementary Table 2.

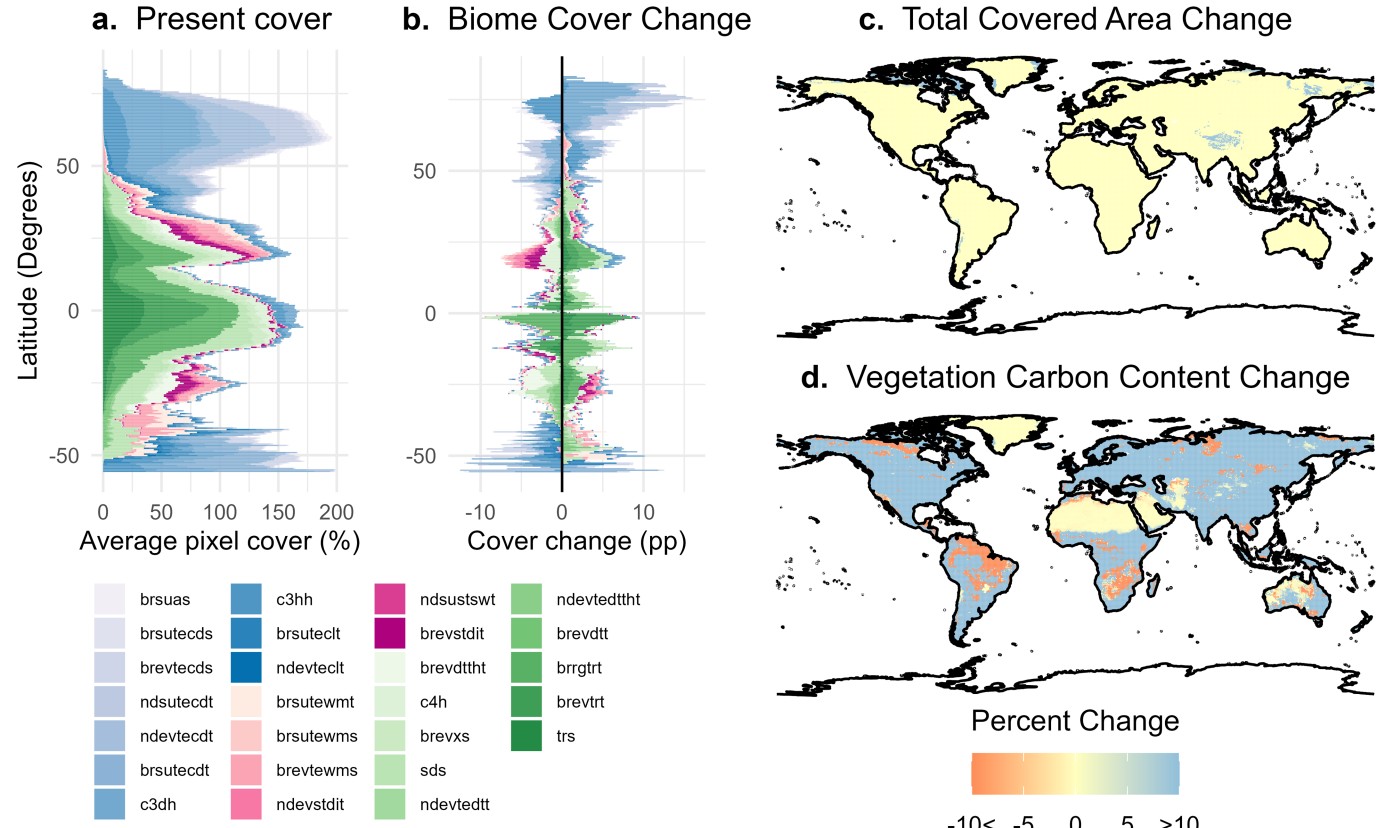

**a.** Present cover

**b.** Biome Cover Change

**c.** Total Covered Area Change

**d.** Vegetation Carbon Content Change

Latitude (Degrees)

Average pixel cover (%)

Cover change (pp)

Percent Change

-10< -5 0 5 >10

Legend:
- brsuas
- brsutecds
- brevtecds
- ndsutecdt
- ndevtecdt
- brsutecdt
- c3dh
- c3hh
- brsuteclt
- ndevteclt
- brsutewmt
- brsutewms
- brevtewms
- ndevstdit
- ndsustswt
- brevstdit
- brevdtht
- c4h
- brevxs
- sds
- ndevtedtt
- ndevtedttht
- brevdtt
- brrgtrt
- brevtrt
- trs

**Extended Data Fig. 4 | Biome shifts and changes in area cover and vegetation carbon content for the CARAIB model. a**, Average percentage of grid cell covered by different biomes in the present (2016–2020). **b**, Change in coverage under 2 °C warming projections relative to present day (using different GCMs and RCPs). **c**, Changes in fraction of grid cells covered by natural vegetation. **d**, Changes in carbon vegetation content (kilograms per square metre). Model output from CARAIB under two warming scenarios (RCP2.6 and RCP6.0) and four climate model outputs (HadGEM2-ES, GFDL-ESM2M, IPSL-CM5-LR and MIROC5).

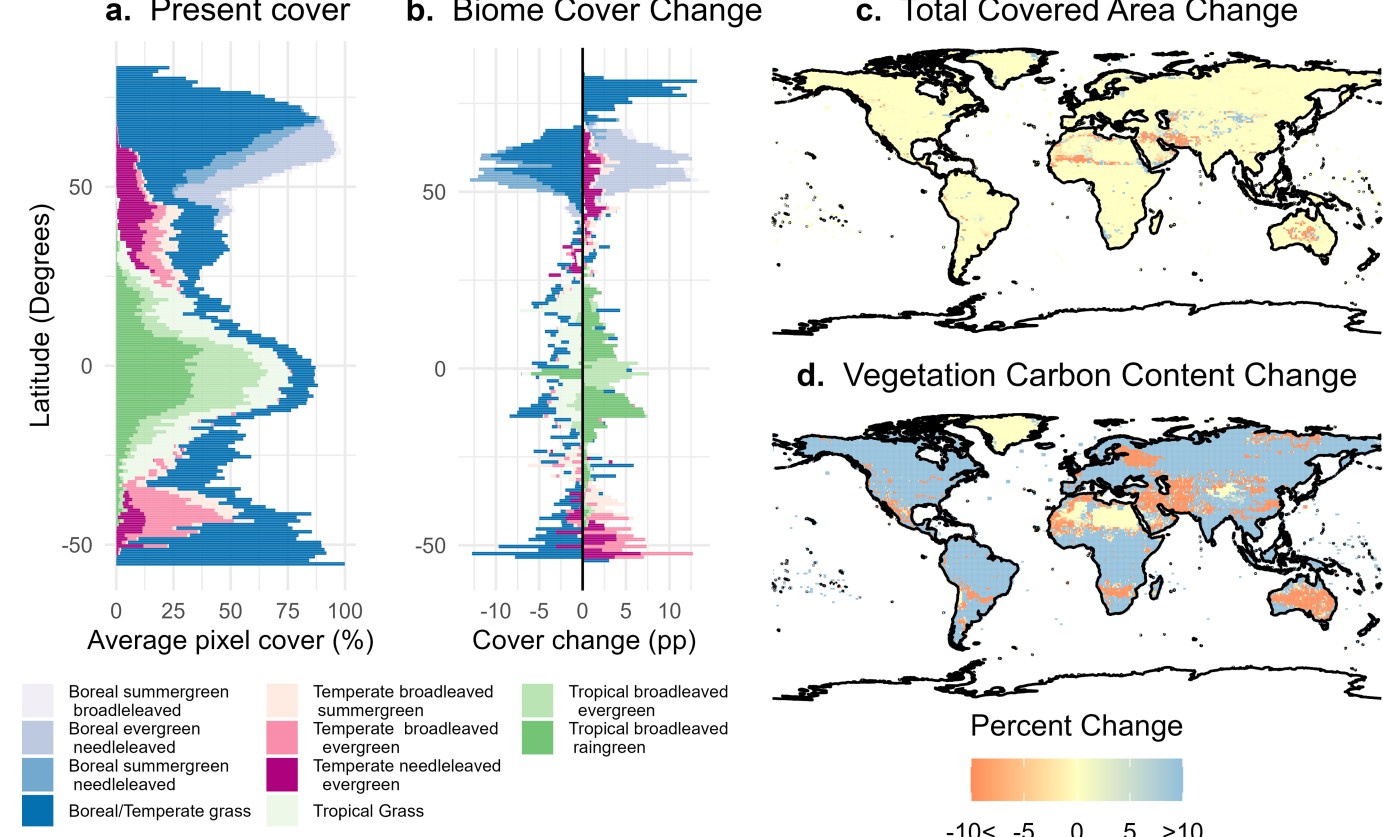

**a.** Present cover

**b.** Biome Cover Change

**c.** Total Covered Area Change

**d.** Vegetation Carbon Content Change

Percent Change

-10< -5 0 5 >10

Legend:
- Boreal summergreen broadleleaved
- Boreal evergreen needleleaved
- Boreal summergreen needleleaved
- Boreal/Temperate grass
- Temperate broadleaved summergreen
- Temperate broadleaved evergreen
- Temperate needleleaved evergreen
- Tropical Grass
- Tropical broadleaved evergreen
- Tropical broadleaved raingreen

**Extended Data Fig. 5 | Biome shifts and changes in area cover and vegetation carbon content for the ORCHIDEE-DGVM model. a**, Average percentage of grid cell covered by different biomes in the present (2016–2020). **b**, Change in coverage under 2 °C warming projections relative to present day (using different GCMs and RCPs). **c**, Changes in fraction of grid cells covered by natural vegetation. **d**, Changes in carbon vegetation content (kilograms per square metre). Model output from ORCHIDEE-DGVM under two warming scenarios (RCP2.6 and RCP6.0) and two climate model outputs (GFDL-ESM2M and IPSL-CM5-LR).

**a.** Benefits per area ($/ha/year)

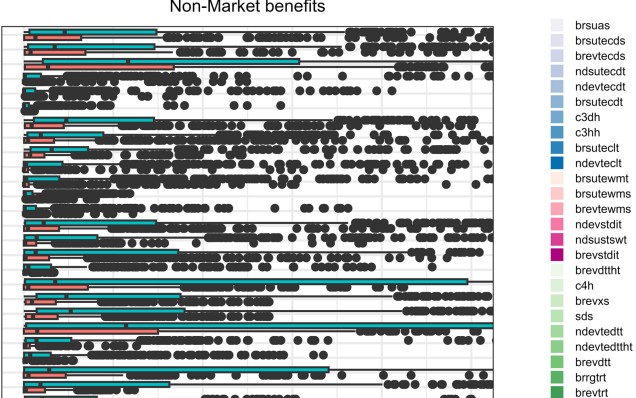

**b.** Total Benefits ($/year)

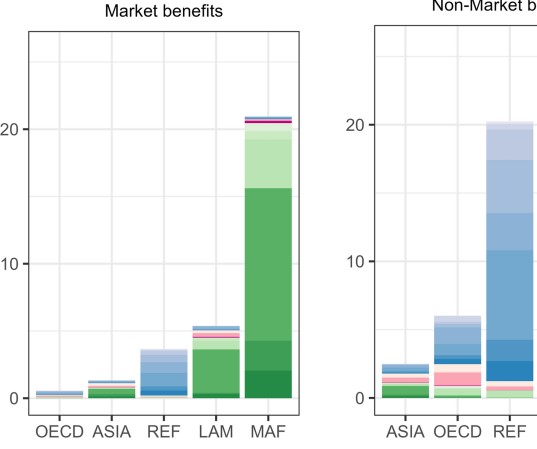

**c.** Percent Damage at 1 Degree of Warming

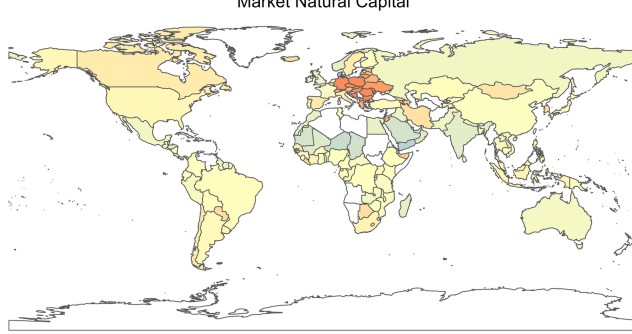

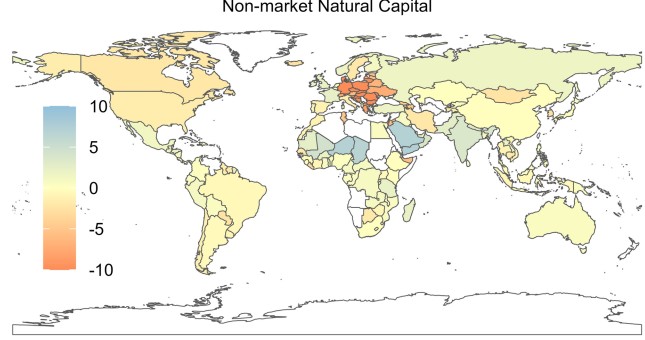

**Extended Data Fig. 6 | Market and non-market benefits by country and biome for the CARAIB model. a**, Distribution of country-level benefits per hectare of biome. Red and blue boxes show market and non-market benefits, respectively. The middle line in the box shows the median, the box covers the first to the third quartile and whiskers show the full range of the data, except for some outliers shown by black points. **b**, Total yearly benefits per geographic region per biome. Note that values are reported as equivalent fractions of GDP as a comparison point only; non-market benefits are not captured in standard GDP accounting practices. **c**, Changes in the market and non-market natural capital for 1 °C of warming.

**a.** Benefits per area ($/ha/year)

Non-Market benefits

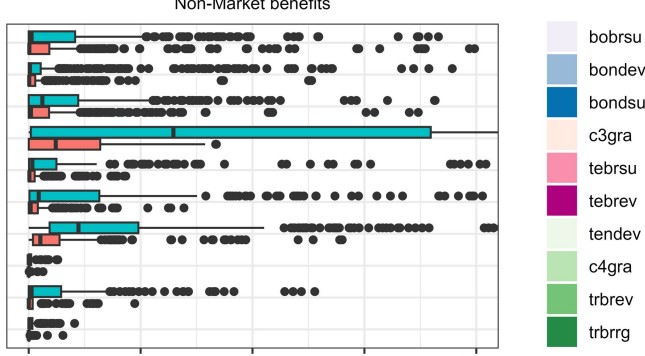

| | |
|---|---|
| bobrsu | |
| bondev | |
| bondsu | |
| c3gra | |
| tebrsu | |
| tebrev | |
| tendev | |
| c4gra | |
| trbrev | |
| trbrrg | |

**b.** Total Benefits ($/year)

Market benefits

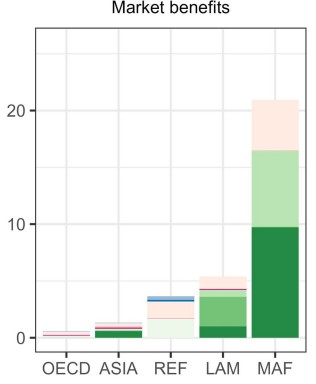

Non-Market benefits

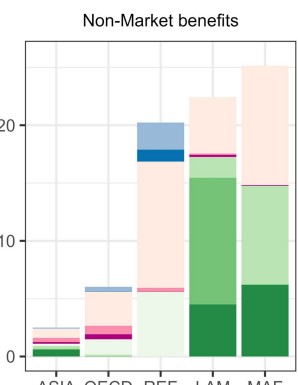

**c.** Percent Damage at 1 Degree of Warming

Market Natural Capital

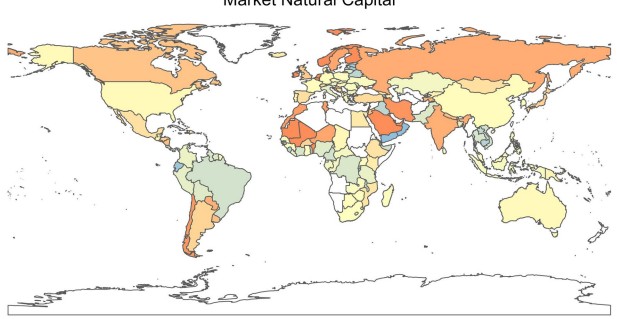

Non-market Natural Capital

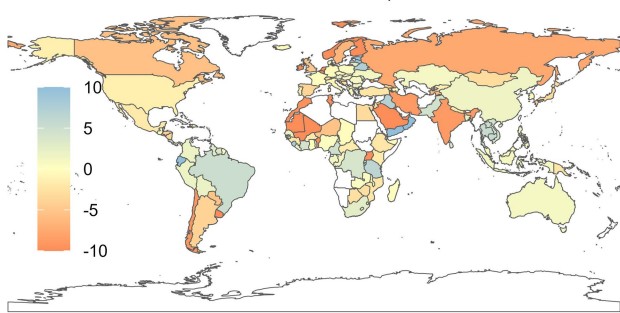

**Extended Data Fig. 7 | Market and non-market benefits by country and biome for the ORCHIDEE-DGVM model. a**, Distribution of country-level benefits per hectare of biome. Red and blue boxes show market and non-market benefits, respectively. The middle line in the box shows the median, the box covers the first to the third quartile and whiskers show the full range of the data, except for some outliers shown by black points. **b**, Total yearly benefits per geographic region per biome. Note that values are reported as equivalent fractions of GDP as a comparison point only; non-market benefits are not captured in standard GDP accounting practices. **c**, Changes in the market and non-market natural capital for 1 °C of warming.

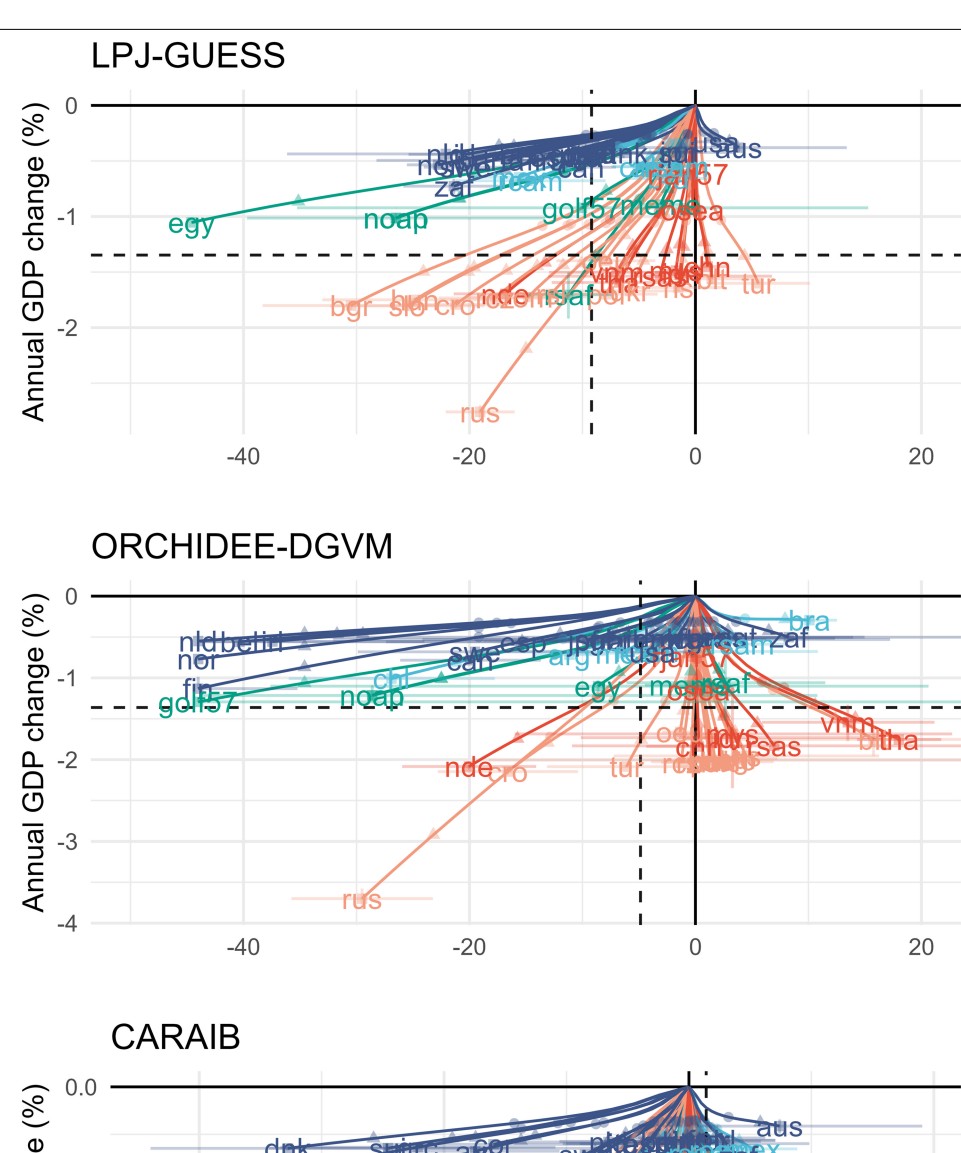

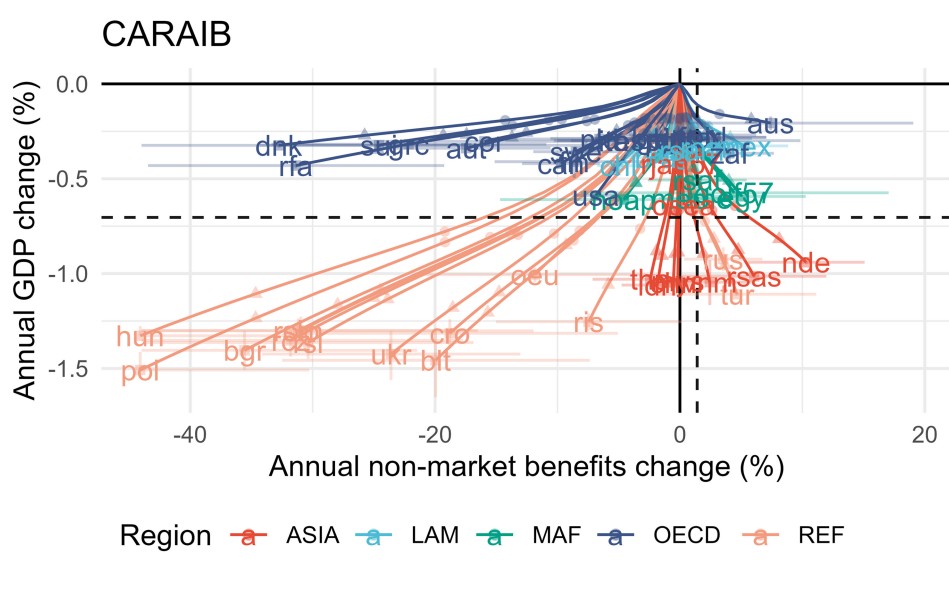

**Extended Data Fig. 8 | Trajectories of the annual change in market and non-market benefits.** Shown for 57 regions in 'Green' RICE50+ with respect to simulations without climate-change impacts. Labels in 2100 show the code of the regions as given in RICE50+. Error bars show the standard errors of the estimated damage functions under different GCM outputs. Dashed lines show the population-weighted mean values in 2100. Upper, results under LPJ-GUESS simulation output (shown in Fig. 4). Middle, results under ORCHIDEE-DGVM simulation output. Lower, results under CARAIB simulation output.

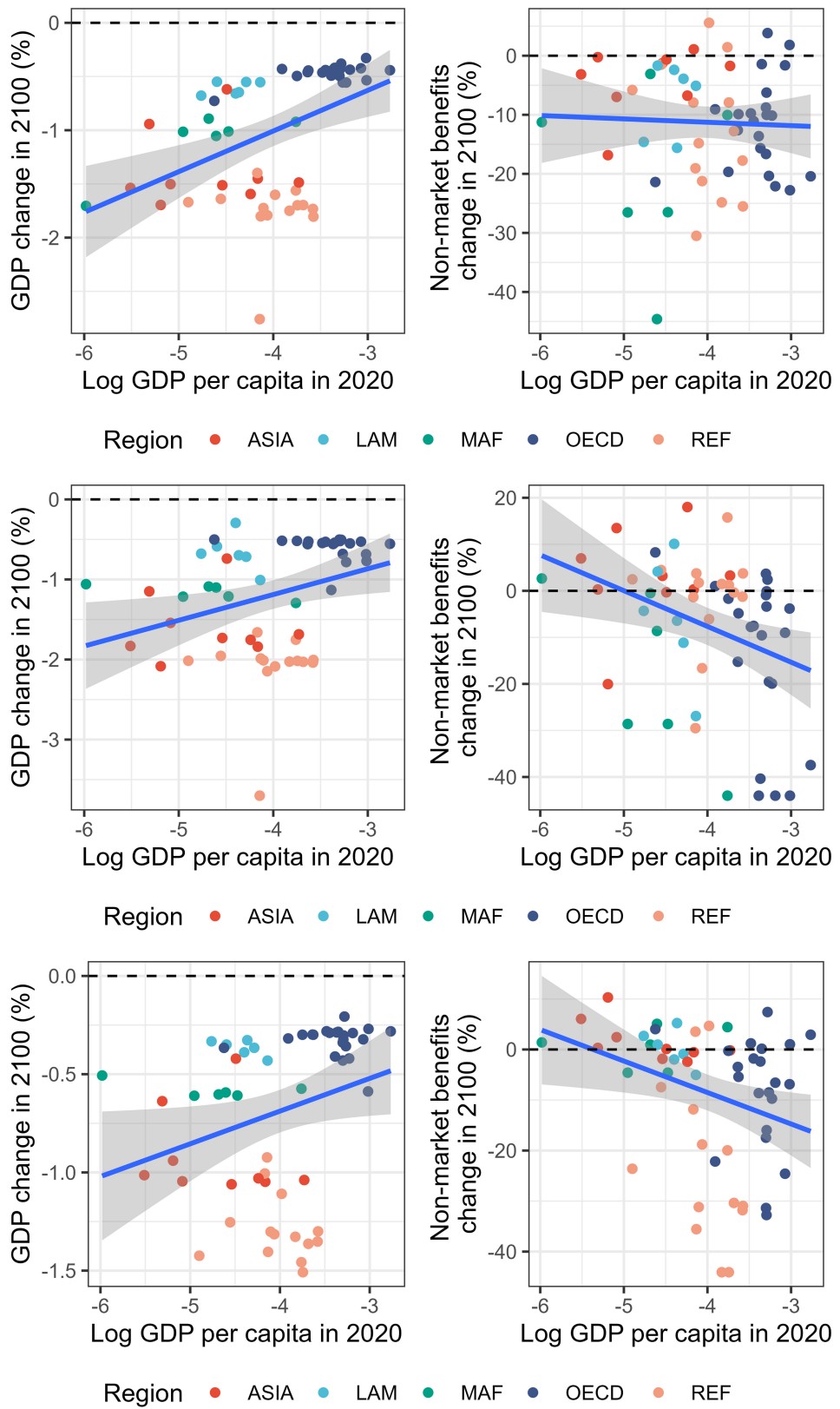

**Extended Data Fig. 9 | Impacts in 2100.** Changes in market benefits (left) and ecosystem services (right) in 2100 under SSP2-6.0 with respect to the baseline scenario in which natural capital remains constant. Upper, results under LPJ-GUESS simulation output. Middle, results under ORCHIDEE-DGVM simulation output. Lower, results under CARAIB simulation output.

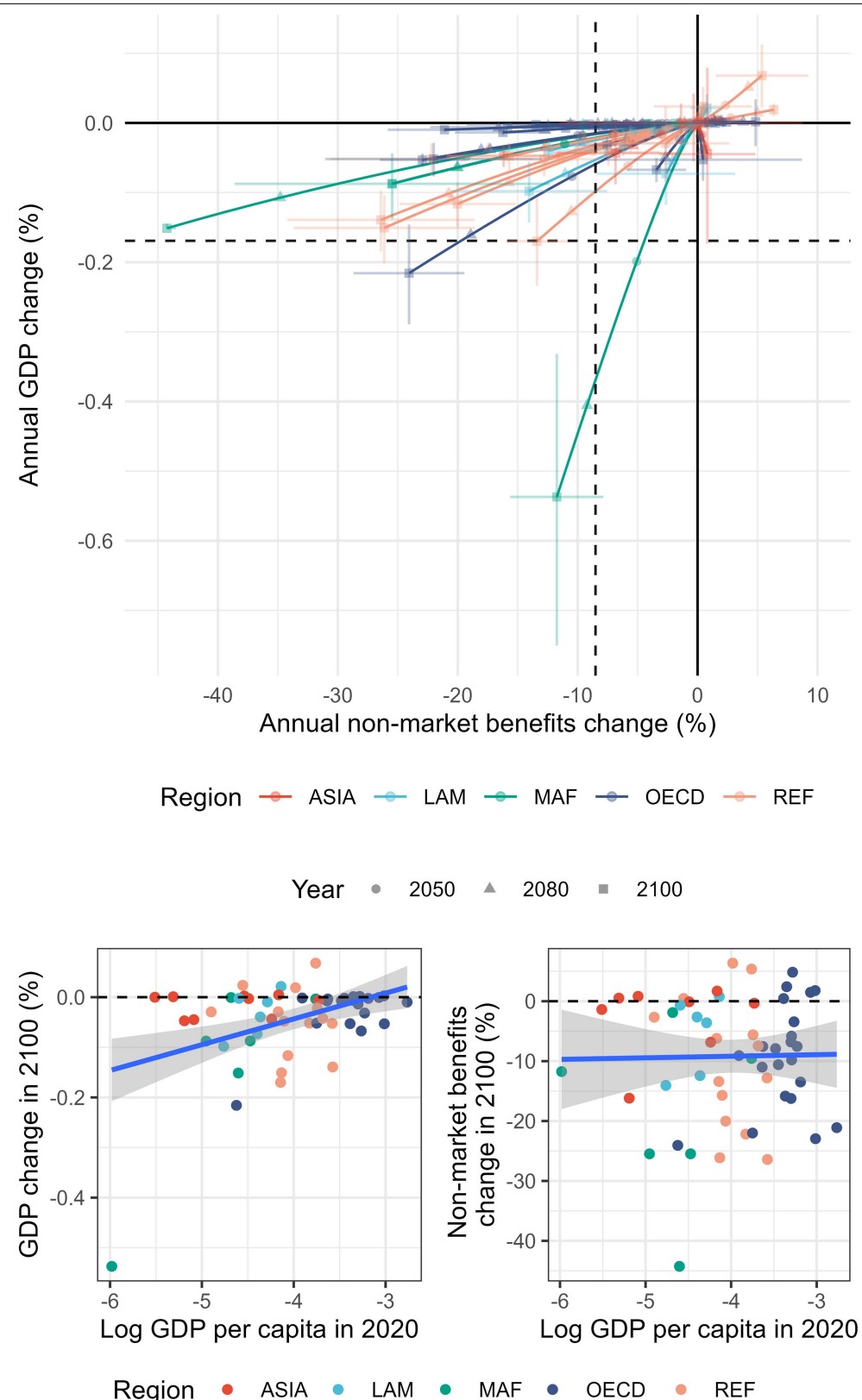

**Extended Data Fig. 10 | Main results with neoclassical production function.** Replication of the main results using a standard neoclassical production function, equation (18), assuming $b = 0$. Upper, trajectories of the annual change in market and non-market benefits are shown for 57 regions in 'Green' RICE50+ using $b = 0$ in the production function. Error bars show the standard errors of the estimated damage functions under different GCM outputs. Dashed lines show the population-weighted mean values in 2100. Population-weighted GDP change in 2100: −0.17, an order of magnitude lower than the preferred estimates for $b$. Lower, distribution of impacts in 2100; the pattern of the distributional burden of impacts remains.

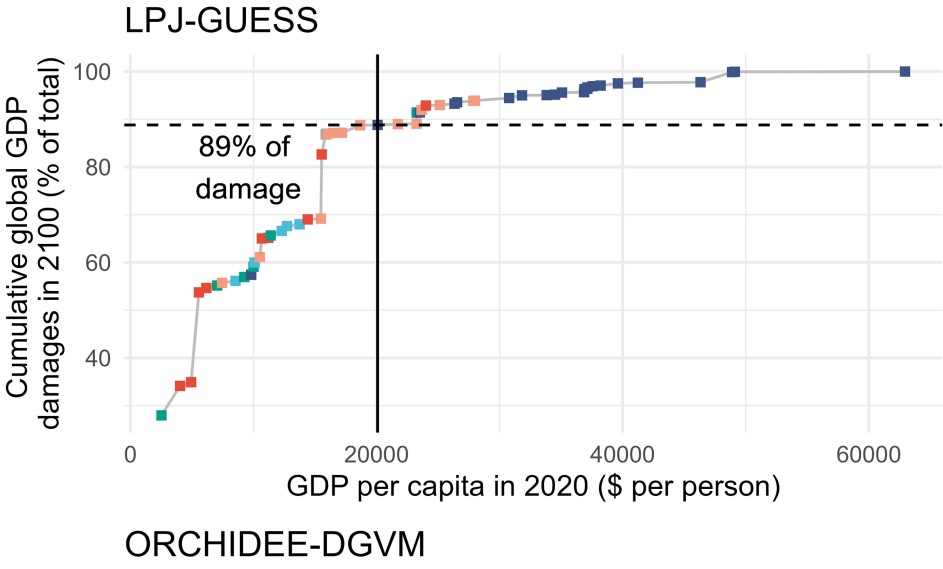

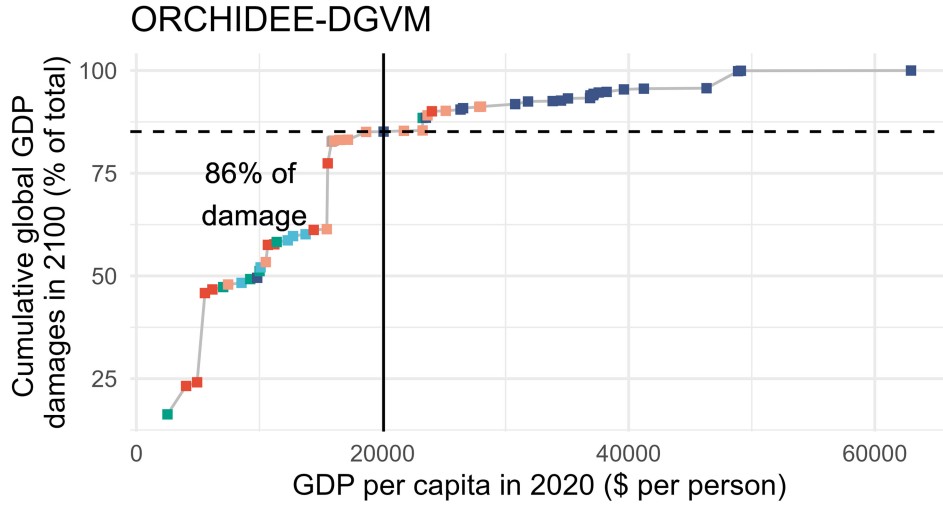

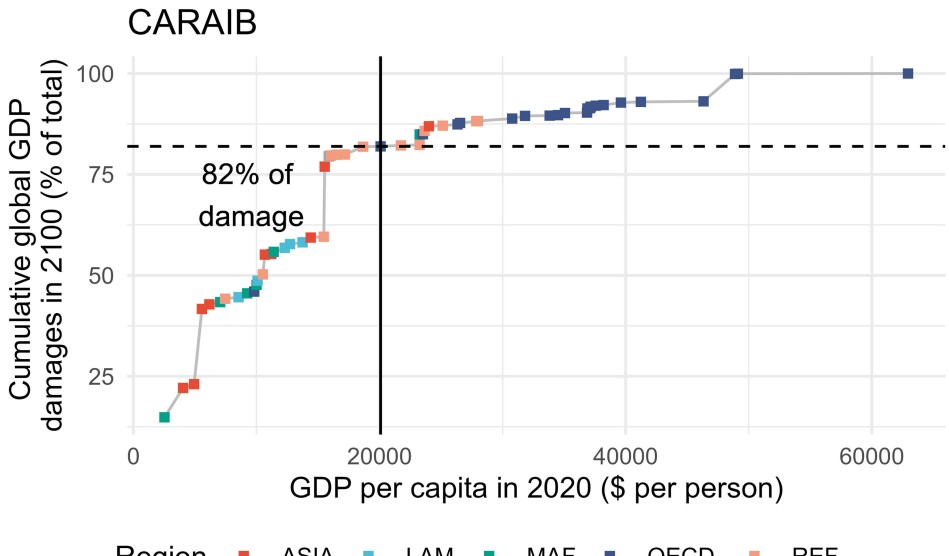

Region    ■ ASIA    ■ LAM    ■ MAF    ■ OECD    ■ REF

**Extended Data Fig. 11 | Unequal distribution of damages.** Distribution of total damages in 2100 across different countries and regions ordered by GDP per capita. Vertical solid lines show the 50% cutoff, horizontal dashed lines show the percentage of total damages borne by the bottom 50%, which is 89%, 86% and 82% for the LPJ-GUESS, ORCHIDEE-DGVM and CARAIB models, respectively. The distributional pattern remains very similar under 2100 GDP per capita estimates.

| DGVM | Biomes typology | GCMs | Warming Scenarios | number of runs |
|---|---|---|---|---|
| LPJ-GUESS (biomes=11) | Boreal needleleaved evergreen (BNE); Boreal shade intolerant needleleaved evergreen (BINE); Boreal needleleaved summergreen (BNS); Temperate broadleaved summergreen (TeBS); shade intolerant broadleaved summergreen (IBS); Temperate broadleaved evergreen (TeBE); Tropical broadleaved evergreen (TrBE); Tropical shade intolerant broadleaved evergreen (TrIBE); Tropical broadleaved raingreen (TrBR); C3 grass (C3G); C4 grass (C4G). | HadGEM2-ES, GFDL-ESM2M, IPSL-CM5-LR, MIROC5 | RCP2.6, RCP6.0, RCP8.5. | 16 |
| ORCHIDEE (biomes=10) | Tropical broadleaved evergreen (trbrev); tropical broadleaved raingreen (trbrrg); temperate needleleaf evergreen (tendev); temperate broadleaved evergreen (tebrev); temperate broadleaved summergreen (tebrsu); boreal needleleaf evergreen (bondev); boreal broadleaved summergreen (bobrsu); boreal needleleaf summergreen (bondsu); C3 natural grass (c3gra); C4 natural grass (c4gra). | GFDL-ESM2M, IPSL-CM5-LR | RCP2.6, RCP6.0 | 4 |
| CARAIB (biomes=26) | C3 herbs (humid) (c3hh); C3 herbs (dry) (c3dh); C4 herbs (c4h); Broadleaved summergreen arctic shrubs (brsuas); Broadleaved summergreen boreal or temperate cold shrubs (brsutecds); Broadleaved summergreen temperate warm shrubs (brsutewms); Broadleaved evergreen boreal or temperate cold shrubs (brevtecds); Broadleaved evergreen temperate warm shrubs (brevtewms); Broadleaved evergreen xeric shrubs (brevxs); Subdesertic shrubs (sds); Tropical shrubs (trs); Needleleaved evergreen boreal or temperate cold trees (ndevtecdt); Needleleaved evergreen temperate cool trees (ndevteclt); Needleleaved evergreen trees, drought-tolerant (ndevtedtt); Needleleaved evergreen trees, drought-tolerant, thermophilous (ndevtedttht); Needleleaved evergreen subtropical trees, drought-intolerant (ndevstdit); Needleleaved summergreen boreal or temperate cold trees (ndsutecdt); Needleleaved summergreen subtropical swamp trees (ndsustswt); Broadleaved evergreen trees, drought tolerant (brevdtt); Broadleaved evergreen trees, drought-tolerant, thermophilous (brevdttht); Broadleaved evergreen subtropical trees, drought-intolerant (brevstdit); Broadleaved summergreen boreal or temperate cold trees (brsutecdt); Broadleaved summergreen temperate cool trees (brsuteclt); Broadleaved summergreen temperate warm trees (brsutewmt); Broadleaved raingreen tropical trees (brrgtrt); Broadleaved evergreen tropical trees (brevtrt). | HadGEM2-ES, GFDL-ESM2M, IPSL-CM5-LR, MIROC5 | RCP2.6, RCP6.0 | 8 |

**Extended Data Table 2 | Variables in the extended VEGS database**

| Variable name | Description |
|---|---|
| Ecosystem benefit Value | Value of the ecosystem benefits to people as measured in the study. Values in VEGS original database.<br>Units: $/ha/year |
| Ecosystem Service Type | Categorical variable describing what type of ecosystem service is being measured. Categories: Market (Provisioning), Non-Market (Information, Regulating, Supporting services). Values in VEGS original database.<br><br>Units: NA |
| GDP per capita | Gross domestic product per capita. Using the gridded GDP per capita database from Kummu et al. Values were retrieved from the database using the year closest to the year of study in VEGS.<br><br>Units: 2018USD/person |
| $cover_b$ | Percent of the study area covered by biome b. Mean value from 2016 to 2020. Categories of b retrieved from 3 dynamic global vegetation models: LPJ-GUESS (11 biomes), ORCHIDEE (10 biomes), and CARAIB (26 biomes). 48 different columns in total for each observation.<br><br>Units: % |
| cveg | Carbon content in vegetation. Mean values from 2016 to 2020. Retrieved from 3 dynamic global vegetation models: LPJ-GUESS, ORCHIDEE, and CARAIB.<br><br>Units: $kg/m^2$ |
| perc_covered | Total percentage of the area of study covered by biomes $b$<br><br>Units: % |

**Extended Data Table 3 | Regression table of the ecosystem benefits elasticities, using the VEGS database**

| | Dependent variable: |
|---:|:---|
| | log(Value per hectare) |
| | |
| log(cveg)<br>Reference level: Non-market | 0.282** |
| | (0.143) |
| Market | 8.046 |
| | (5.004) |
| log(Area)<br>Reference level: Non-market | -0.1032* |
| | (0.049) |
| log(gdp_pc) | 0.593** |
| | (0.299) |
| log(Percentage covered) | 0.764*** |
| | (0.286) |
| log(cveg)*Market | 0.307 |
| | (0.316) |
| log(area)*Market | 0.002 |
| | (0.075) |
| log(gdp_pc) * Market | -0.969* |
| | (0.498) |
| Fixed EffectsConstant<br>Reference Level: Non-market | Country<br>Valuation Methodology |
| Observations | 483 |
| $R^2$ | 0.261 |
| Adjusted $R^2$ | 0.152 |
| Residual Std. Error | 2.476 (df = 420) |
| F Statistic | 6.07*** (df = 8; 420) |
| Note: | *$p<0.1$;**$p<0.05$;***$p<0.01$ |

**Extended Data Table 4 | Regression table of the production function, using World Bank data**

| | Dependent variable: |
|---:|:---|
| | log(GDP) |
| | |
| *log(K)* | 0.266434*** |
| | (0.09844) |
| *log(H)* | 0.502770*** |
| | (009811) |
| *log(mN)* | 0.004467 |
| | (0.004710) |
| *ASIA:log(S)* | 0.559234*** |
| | (0.0.050625) |
| *LAM:log(S)* | 0.193776*** |
| | (0.048567) |
| *MAF:log(S)* | 0.322886*** |
| | (0.046696) |
| *OECD:log(S)* | 0.153859** |
| | (0.050282) |
| *REF:log(S)* | 0.612428*** |
| | (0.048990) |
| *year* | 0.003151*** |
| | (0.001122) |
| | |
| | |
| | |
| $R^2$ | 0.9987 |
| Adjusted $R^2$ | 0.9986 |
| Residual Std. Error | 0.07643 (df = 2895) |
| F Statistic | 1589*** (df = 2895) |
| Note: | *$p<0.5$;**$p<0.01$;***$p<0.001$ |