## [Peer Review File · Nature]

Manuscript Title: Unequal Climate Impacts on Global Values of Natural Capital

Editorial Notes:

Reviewer Comments & Author Rebuttals

Reviewer Reports on the Initial Version:

Reviewers' comments:

Referee #1:

Remarks to the Author:

In this manuscript, the authors tried to attach economic values (i.e., market and non-market benefits in terms of natural capital) to the global vegetation cover changes caused by future climate changes. Based on their country-level analysis, they showed that the mean global flow of ecosystem services (i.e., non-market benefits) will be reduced by 20% in 2100 under the "middle of the road" of the shared socioeconomic pathways and radiative forcing target (SSP2-4.5) for the future. It is particularly emphasized that poor countries which are more reliant on natural capital, will be more negatively impacted and thus exacerbate economic inequality among countries.

The study is novel and the results (if robust) will have a big impact on our climate change policies. But I am not convinced by the data and method presented in the current manuscript. More detailed explanations on the hypotheses made when deriving the natural capital and damage function, and more clarification on the choice of data and methods are needed. A better quantification and discussion of the uncertainties and potential biases of the results are also required.

Major issues:

(1) The assumption that the area/coverage of a biome is the only factor affecting its market and non-market values (eq. 5 and 6 in the Supplementary materials) are questionable and can be misleading. Besides the area of a biome, other quantities (such as total biomass per unit area or wood density) might also affect its natural capital. These variables are not necessarily correlated with the coverage of a biome. It is possible that the biomass per unit area of a biome might increase in the future (increasing the natural capital), while its coverage decreases (reducing the natural capital).

The coverage of a biome is one of the most uncertain variables simulated in terrestrial ecosystem models, e.g., the tree cover in African Savannah (e.g. Forkel et al. 2019). Given such large uncertainties, I doubt the robustness of the high loss of natural capital of African countries (dominated by Savannah) due to biome shift in the future scenarios.

(2) The uncertainty is large when using only one terrestrial ecosystem model (i.e., LPJ-GUESS) from the Inter-Sectoral Impact Model Intercomparison Project (ISIMIP) to derive future biome change. Several models' output are actually available from ISIMIP, and it is highly recommended to use multi-model projections to estimate the natural capital loss, so that the uncertainties and potential biases of the results can be better evaluated.

In addition, the change of biome can have significant feedback to climate which can either enhance

or dampen the biome changes originally induced by climate change. The future biome changes from ISIMIP simulations only include the impact of climate on biome changes, but not the potential feedbacks between climate and biome changes. It is thus recommended to check available CMIP6 simulations with dynamic vegetation cover (see Arora et al. 2020), and use the biome changes from at least one of those simulations to estimate full potential changes in biomes and its natural capital.

(4) The fitting of the quadratic damage function (eq. 7) for different countries uses only 3 points (if I understand correctly), and there is no statistical significance reported in Supplementary Table 3.

(5) There is no detailed explanation on how the random forest is applied to predict the ecosystem service values, and how accurate the random forest approach is. This is a critical step for estimating country level market and non-market natural capital. A clear explanation is needed.

(3) Please explain why use SSP2-4.5 scenario for economic projections, while using RCP6.0 for biome changes? Are they consistent with each other?

Specific comments:

Abstract: need to explain “market and non-market benefits” and “SSP2-4.5”

Page 2: Please explain “non-use values” and “inclusive wealth”.

Figure 2A: Please explain why “most of the globe loses natural vegetation cover by the end of the century”? Do “biomass” or “LAI” have the same trend?

Page 4: Please add a reference for the VEGS database?

Page 5: When talking about “damage function”, please refer to “eq. 7 in Supplement”

Page 6: “equation 1” should be “equation 1 in Supplement”

Figure 4: Please refer to supplementary materials when mentioning “four damage functions estimated with different general circulation model outputs”.

Page 7: citation number 24 is wrongly used. Please check!

Page 8: What does “SCC” mean?

Eq.2: What is the meaning of “ r ” in the equation? Please explain.

Page 2 in Supplement, bottom 2 line: “The country-level X_b values are used...” should be “The country-level A_t , c , b values are used...”

Page 3 in Supplement: “See Supplementary Figure 5”? There is no Figure 5 in the Supplementary file.

Page 3 in Supplement: “time time” should be “time”.

References:

Forkel, M., Drüke, M., Thurner, M. et al. Constraining modelled global vegetation dynamics and carbon turnover using multiple satellite observations. *Sci Rep* 9, 18757 (2019).

<https://doi.org/10.1038/s41598-019-55187-7>

Arora, V. K., Katavouta, A., Williams, R. G., Jones, C. D., Brovkin, V., Friedlingstein, P., Schwinger, J., Bopp, L., Boucher, O., Cadule, P., Chamberlain, M. A., Christian, J. R., Delire, C., Fisher, R. A., Hajima, T., Ilyina, T., Joetzjer, E., Kawamiya, M., Koven, C. D., Krasting, J. P., Law, R. M., Lawrence, D. M., Lenton, A., Lindsay, K., Pongratz, J., Raddatz, T., Séférian, R., Tachiiri, K., Tjiputra, J. F., Wiltshire, A., Wu, T., and Ziehn, T.: Carbon–concentration and carbon–climate feedbacks in CMIP6 models and their comparison to CMIP5 models, *Biogeosciences*, 17, 4173–4222, <https://doi.org/10.5194/bg-17-4173-2020>, 2020.

Referee #2:

Remarks to the Author:

Summary

This paper estimates the economic value of changes in terrestrial vegetation caused by climate change over the remainder of this century. In the paper's framework, terrestrial vegetation – forests and grassland – provides two kinds of economic value. First, forests produce timber, which is a market good. Thus, forests embody "market natural capital". Second, forests and grassland produce a wider range of ecosystem services that are not priced in markets, such as recreation and watershed protection. Thus, forests also embody "non-market natural capital". A global vegetation model (LPJ-Guess) is used to simulate climate-induced changes in the spatial extent of terrestrial vegetation types (biomes). A database of ecosystem service values (VEGS) is used to estimate the relationship between natural capital values (market and non-market) and biome area, for each biome and country. Then, the 21st century simulations from the vegetation model can be translated into country-level estimates of changing natural capital and ecosystem services. The authors find that climate change causes non-market ecosystem service flows from terrestrial vegetation to fall substantially in many countries and globally on average. They find smaller losses in market ecosystem service flows (timber output), but significant heterogeneity and apparently large impacts in some countries/regions, notably Africa.

Major comments

Overall assessment

I think this paper makes a significant contribution to the literature that tries to understand the economic impacts of climate change. This literature is better developed when it comes to 'market impacts' and physical/human capital than it is when it comes to natural capital. Previous studies trying to get a handle on economic impacts of climate change via natural capital have arguably suffered from being too aggregated and stylized to provide numbers that one could have any confidence in. Thus, in my view more detailed modeling is the way forward here and this is exactly what the authors do in this paper by coupling a global vegetation model with high spatial resolution with detailed ecosystem service value data.

However, I have concerns about the methods – really one major concern – and whether the scope of the contribution is large enough to warrant publication in Nature. I also found the paper confusing in places and have some editorial suggestions.

Growth and development in the 21st century

Here is my major concern about the methods. I found the methods section difficult to follow (more on that below), but my understanding is that long-term climate-induced changes in biome area are superimposed on a world economy that is more like today's than the one we would expect to see in

2100. The extent to which it is more like today's than that of 2100 is difficult to figure out (unless I overlooked it, we are never shown what happens to GDP), but the reason I say this is that the two key engines of long-term growth, human capital and TFP, are fixed at their 2018 levels. Physical capital is not fixed, although the paper reports a savings rate of 3% (is this a typo? Should it be 30%?), which means that with a depreciation rate of 10% the physical capital stock could even be shrinking over time. In any case, assuming the savings rate is not as low as 3%, physical capital accumulation is not a major driver of growth in the long term, except perhaps in low-income countries. This matters in several respects. If the economy grows, then a given value of lost ecosystem services in the future is a smaller share of future GDP. The authors place a lot of emphasis on their results reported in terms of percentages of GDP. These may appear larger than they would really be in 2100, because they are compared with the economy of today. Moreover, with a growing economy the value of lost ecosystem services is not given, rather it is a function of incomes and growing capital stocks. Therefore, I find the analysis fundamentally incomplete. An article appearing in a top journal needs to explicitly model the interdependence of capital stocks, incomes and ecosystem service valuations.

Clarity of exposition and detail of methods section

As mentioned above, I had a hard time following the detail of the methods section and came away with the sense that I did not fully understand the procedures that the authors followed. This comment is broader though – I think that the clarity of the exposition could be improved throughout. In the main text, I initially had a hard time understanding which ecosystem services were included in the analysis and which were not. I venture that one of the reasons for my confusion is that this area of scholarship contains many partially overlapping frameworks, including capital theory, typologies of ecosystem services (provisioning, regulating, cultural and their sub-components), and the theory of economic value (market versus non-market). The main text initially mixes these concepts in a way that is not fully clear. I urge the authors to attempt some redrafting of the opening couple of pages that makes it clearer how all these frameworks relate to each other so that the reader can quickly zero in on what the analysis includes and what it leaves out.

Turning to the methods section, I do feel this needs improving. Some of the notation is never spelled out (e.g., r , the consumption discount rate). All variable and parameter symbols should be clarified in prose. Some steps require further intuition, for example it will help readers to explain that equations 3 and 4 use the formula for the PV of a perpetuity to value the initial capital stock. Some notation is confusing. For example, a , e_s and R are indexed with time in equations 3 and 4, yet they are assumed to be time-invariant so that the simple formula for the PV of a perpetuity can be used. In equations 5 and 6, the vector of fractions of each biome is given the same notation, $x_{\{b\}}$, even though I assume the elements of the vector are different for the two capital stocks. The explanation of the important stage of regressing ecosystem service values on the area covered by each biome using a random forest is not explained in enough detail. It is for example unclear whether additional explanatory variables such as the income of the sample population are controlled for. There is next to no description of how LPJ-Guess works – we shouldn't have to cross-refer to find out everything. It is not clear whether human capital in equation 8 is simply the labor force (as in DICE, referred to in the main text), or a broader notion of the totality of people's skills and knowledge. It is not then clear what TFP is measuring. And so on.

Scope of the analysis

Eventually I felt that I understood what was in and what was out. The discussion is very clear on this, but some passages earlier in the paper are arguably guilty of overselling how much further this analysis takes us compared to the existing literature. As described above, it essentially estimates the economic value of lost timber output plus presumably a subset of non-market benefits that are also provided by forests and grasslands (what subset is never clear, but I assume the VGES dataset is incomplete). This is a useful step forward but still leaves a lot of ground to cover, and sometimes the paper gives the impression that it is undertaking a much bigger analysis of comprehensive ecosystem services. Relatedly, the boundary with the existing literature is fuzzy because some models and analyses (e.g., the FUND model) already explicitly include impacts in the forestry sector, while macro-economic studies should implicitly include historical impacts on commercial forestry.

Minor comments

Why use RCP6.0 in Figure 2 but SSP2-4.5 in Figure 4? Should you not use the same scenario throughout?

Referee #3:

Remarks to the Author:

A. Summary of the key results

- Impacts at country-level of market and non-market ecosystem services value under climate change scenarios.
- Changes in GDP and ecosystem service value trajectories under the SSP2-4.5 warming scenario with respect to the baseline scenario in which natural capital is not damaged by climate change.
- Many countries will lose a large proportion of the value of their ecosystem services with respect to the baseline. The global average change in ecosystem service value by 2100 is -20.4% relative to the baseline (no damaged natural capital).
- Changes in non-market ecosystem services are not strongly related to either the country's per-capita income or geographic region that is they are equally damaged by climate change.
- Changes in the market benefits of terrestrial ecosystems, measured in terms of GDP losses are smaller in magnitude but more strongly differentiated by region. The global average change in GDP by 2100 is -0.8% with respect to the 2100 levels of the baseline simulation the equivalent to -1.5% of GDP in 2018. Africa losses 14.4% of Africa's GDP in 2018 while Europe losses by 2100 the 1.2% of Europe's GDP in 2018.
- These results suggest that climate change effects on ecosystems could further increase economic inequality between regions by damaging a stock of wealth particularly important in low and lower-middle-income countries.
- A further simulation shows that limiting warming below 2°C would cut the impacts of GDP by a third (global mean change by 2100: -0.3%) and the impacts on ecosystem services by half (global mean change by 2100: -9.4%) compared to the SSP2-4.5 that puts the planet at 2.6°C of warming by 2100 with respect to preindustrial levels.

B. Originality and significance: if not novel, please include reference

I think that the conclusions are not original. This is one more paper that support that developing countries would be more affected in terms of ecosystem services loss (market ecosystems-value) due to climate change. However, the authors rely on a linear relationship between GDP and ecosystem services.

The paper is an example of the fashion- science- research which uses global models to get global obvious results. The manuscript relies on very general assumptions that cannot be fully supported. If people agree that these approaches are useful, then the paper could be easily published.

Nevertheless, I would not go for oversimplifying the ecosystem services and ecosystem loss only as function based on the GDP.

The authors establish a linear relationship between GDP and ecosystem services. However, such relationship it is not real. GDP has doubled between 1990 and 2015 in constant dollar terms (World Bank, 2019) but not the ecosystem services which have been decreasing (beyond the nonmarket services) i.e., augmenting deforestation, pollution, etc.. The use of GDP has been highly criticized mainly for not expressing the social and well fare and so far, to show ecosystem wellbeing or ecosystem services. As Ouyan et al., (2020) pointed out GDP summarizes a vast amount of economic information in a single monetary metric but it fails to capture fully the contributions of nature to economic activity and human well-being. Other indicators have been proposed as the gross ecosystem product (GEP) or the gross economic-ecological product (GEEP) (Wang et al., 2021) which have tried to overcome GDP's issues as a social or environmental indicator. I suggest to critically analyze the GDP and instead us another economic indicator that has more relationship with ecosystems that use market prices and surrogates for market prices to calculate the accounting value of ecosystem services (Ouyang et al., 2020).

For instance, GEEP aggregates two accounting metrics (GGDP and GEP). GGDP is the adjusted GDP by removing the cost of environmental degradation caused by pollutants discharged into the environment through various pathways (air, water, land) and the loss of ecosystem services caused by the overuse of ecosystems. GEP includes ecosystem regulating value, ecosystem provisioning value, and ecosystem cultural value (UN, EC, OCDE, & World Bank Group, see (Wang et al., 2021)). Since Ecosystem Provisioning Value and Ecosystem Cultural Value are already accounted in GDP), GEEP is thus the sum of GGDP and Ecosystem Regulating Value.

C. Data & methodology: validity of approach, quality of data, quality of presentation

I notice two main points.

a) The approach is not well supported. The base of the paper is to believe that there is a strong relationship between ecosystem services (mainly market valued) with GDP and welfare (Summers et al., 2012).

b) There is a lack of critical argumentation about their approach also in considering the variety of ecosystem services and ecosystems, because they use mean values that do not reflect the diversity intra ecosystems and inter ecosystem (Carrasco et al., 2014).

c) The explanation about what ecosystem services they are assessing is complete dark, the refer to the World Bank report (2021) but it is a 504-page document, and it is not clear neither what commodity are quantifying and there is not transparency (in the report) about the links between valuation of ecosystem services and the biomes. In summary, there is not transparency in the World

Bank report in the method to calculate the mean values per biome per country.

D. Appropriate use of statistics and treatment of uncertainties

There is a lack of attention to show the uncertainties among the ecosystem services by biomes. It seems that authors are only using means, but they are not showing the uncertainties. A global study cannot oversimplify the impacts of climate change in ecosystem services by biome and country just as a single value. I would encourage the authors to go through the variation and uncertainty of their estimates.

E. Conclusions: robustness, validity, reliability

From my point of view, the main problem of the paper is the acritical oversimplification of the relationships among climate change, biomes, GDP, welfare, and inequality.

The impacts of climate change on ecosystems have widely studied for decades at different temporal and spatial scales (Batllori et al., 2013; Gonzalez et al., 2010; Hoffmann et al., 2019; Lloret and Batllori, 2021; Soteriades et al., 2017; Weiskopf et al., 2020). Consequently, in this context, I think that the paper is not original. However, the authors tried to link this biome's shift due to climate change with welfare, through the concept of ecosystem services and use the GDP as an indicator between them. I understand the purpose of that, but I think that simplifying the complexity of biodiversity loss or in this case ecosystem services loss due to climate change, and inequality among countries do not help to improve our understanding of the challenges we face, and that will be exacerbated by climate change.

F. Suggested improvements: experiments, data for possible revision

I would suggest:

a) Use instead of GDP the ecosystem product (GEP) or the gross economic-ecological product (GEEP) (Wang et al., 2021) which have tried to overcome GDP's issues as a social or environmental indicator.

b) Critically analyze the GDP in their narrative and this includes to include more disciplines in their sentence where the authors say, "The human wellbeing effects of ecosystem damages due to climate change is still highly uncertain and depends on integrating insights from climate science, ecology, and environmental economics." And I wonder what about social sciences, anthropology, I think that main driver of human welfare and ecosystem welfare nowadays is related intrinsically to social actions and human decisions. Consequently, we cannot say that only climate science, ecology and environmental economics need to be integrated to analyse climate change challenges and ecosystem damages.

c) Explain deeper what kind of ecosystem services (market and non-market) are using. It is not clear at all and the World Bank report (2021) that authors use as base it is not clear neither.

d) Try to include the uncertainty among biomes, besides the World Bank methodology I guess there are multiple papers to be used about ecosystem services per biome in different countries. However, the authors try to keep simple (it is good in one way) but they are really oversimplifying to a point where it is very difficult to notice an interesting novelty in the paper.

G. References: appropriate credit to previous work?

Batllori, E., Parisien, M.-A., Krawchuk, M.A. and Moritz, M.A., 2013. Climate change-induced shifts in fire for Mediterranean ecosystems. *Global Ecology and Biogeography*, 22(10): 1118-1129.

Carrasco, L.R., Nghiem, T.P.L., Sunderland, T. and Koh, L.P., 2014. Economic valuation of ecosystem services fails to capture biodiversity value of tropical forests. *Biological Conservation*, 178: 163-170.

Gonzalez, P., Neilson, R.P., Lenihan, J.M. and Drapek, R.J., 2010. Global patterns in the vulnerability of ecosystems to vegetation shifts due to climate change. *Global Ecology and Biogeography*, 19(6): 755-768.

Hoffmann, S., Irl, S.D.H. and Beierkuhnlein, C., 2019. Predicted climate shifts within terrestrial protected areas worldwide. *Nature Communications*, 10(1): 4787.

Lloret, F. and Batllori, E., 2021. Climate-Induced Global Forest Shifts due to Heatwave-Drought. In: J.G. Canadell and R.B. Jackson (Editors), *Ecosystem Collapse and Climate Change*. Springer International Publishing, Cham, pp. 155-186.

Ouyang, Z. et al., 2020. Using gross ecosystem product (GEP) to value nature in decision making. *Proceedings of the National Academy of Sciences*, 117(25): 14593-14601.

Pendrill, F. et al., 2022. Disentangling the numbers behind agriculture-driven tropical deforestation. *Science*, 377(6611): eabm9267.

Soteriades, A.D., Murray-Rust, D., Trabucco, A. and Metzger, M.J., 2017. Understanding global climate change scenarios through bioclimate stratification. *Environmental Research Letters*, 12(8): 084002.

Wang, J. et al., 2021. Gross economic-ecological product as an integrated measure for ecological service and economic products. *Resources, Conservation and Recycling*, 171: 105566.

Weiskopf, S.R. et al., 2020. Climate change effects on biodiversity, ecosystems, ecosystem services, and natural resource management in the United States. *Science of The Total Environment*, 733: 137782.

H. Clarity and context: lucidity of abstract/summary, appropriateness of abstract, introduction and conclusions

The abstract is good. I would recommend being more critical about the approach they use in the introduction and the conclusions are obvious. The problem is to keep doing this kind of research to find what we already know such as developing countries will be more damaged by climate change than developed countries. It is circular the argument when you are using an economic indicator to assess the impacts.

Additional comments

1. Author refers that changes in biome abundance and type will cause changes in the flow of market and non-market benefits that society receives from these ecosystems. However, they use the changes in area and ecosystem services in comparison to the baseline, but they are not, as far as I understood, integrating the shifting ecosystem services (possible future ecosystem services with different biomes) that they can gain with different conditions.
2. The authors say that limiting the temperature increase within 2°C could cut the damage on GDP and ecosystem services at least by half and reduce economic inequality among countries. There is not linear and causal relationship neither about the limits of warming (2°C) to assure that the inequality will be reduce. Inequality is more complex than 1 or 2 Celsius degrees, it is related to power and domination between groups. I would suggest not trivializing the social and environmental

inequality with this kind of sentences.

3. Figure 4. Graphs 2 and 3 are not clear at all. On the graph on the bottom, it seems that ecosystem services in Europe are in the most affected, but it is not clear when comparing with the 2nd graph.

4. Authors say that changes in non-market ecosystem services are not strongly related to either the country's per-capita, income, or geographic region, implying welfare of both rich and poor nations is equally. Perhaps at country level but that does not make sense. I think that welfare related to ecosystem services are related to the type of biome and they have different resistance, resilience, and adaptive capacity, consequently it is hardly believed that the impacts are equally. Authors are oversimplifying the ecosystems functions and ergo the ecosystem services.

5. In the discussions the authors say, "The human wellbeing effects of ecosystem damages due to climate change is still highly uncertain and depends on integrating insights from climate science, ecology, and environmental economics." And I wonder what about social sciences, anthropology, I think that main driver of human welfare and ecosystem welfare nowadays is related intrinsically to social actions and human decisions. Consequently, we cannot say that only climate science, ecology and environmental economics need to be integrated to analyse climate change challenges and ecosystem damages.

6. Authors recognize some of the limitations of their approach and they mentioned "It is important to note that many climate change impacts on ecosystems are not considered in this analysis. For instance, the vegetation model we use does not account for major ecosystem disturbance events like insect-driven tree mortality or wildfires, which substantially affect vegetation spatial patterns and the carbon cycle". But what about land use/cover change. In many tropics is the main driver of ecosystem services and biodiversity loss.

7. Authors affirm that it is possible to relate the changes in the market benefits of terrestrial ecosystems by looking at the changes in GDP under climate change conditions. I would say that it is not necessary to develop this type of analysis to see that poor countries, mainly in Africa, are struggling with impacts of changing climate like droughts. However, the study does not include droughts only what they call as "temperature change". They do not clarify if it is mean temperature (very likely), maximum temperature, min temperature, etc.

8. The paper is based on the World Bank assumptions of ecosystem services. This approach ignores the main driver of ecosystem loss in developing countries (agricultural expansion) (Pendrill et al., 2022) because they keep constant the non-forest area. Consequently, that can explain only the impacts of climate change, but they are not doing fully that because they are focusing on the relationship between ecosystem services and GDP.

9. Equation 3 and 4 are not supported from an ecological perspective. Ecosystem services value cannot be given in that simplistic form: a) the mean value by ecosystem ignores the variety of ecosystems that countries have because the LPJ model includes only a set of them, b) the agricultural land and agroforestry and agrodiversity that are one of the most important parts of ecosystem services, c) the assumption of the 4th equation that show that all the forest ecosystems can be quantified as possible timber.

10. I would suggest that authors explain more carefully what ecosystem services they consider as market and non-market because only point out that market products (timber), and non-market (recreation, watershed protection, and other non-wood forest products) it is not enough. Authors refer in the results that stocks of 5 natural, human and manufactured capital and country-specific production elasticities for manufactured and human capital but they did not explain further and it is

not clear what really ecosystem services they are estimating. The methodology is not quite clear about the ecosystem services and neither the World Bank report that they use as basis (World Bank, 2021).

Author Rebuttals to Initial Comments:

Biome range shifts under climate change: impacts on country-level macro-economic production and ecosystem services

Referee #1 (Remarks to the Author):

In this manuscript, the authors tried to attach economic values (i.e., market and non-market benefits in terms of natural capital) to the global vegetation cover changes caused by future climate changes. Based on their country-level analysis, they showed that the mean global flow of ecosystem services (i.e., non-market benefits) will be reduced by 20% in 2100 under the “middle of the road” of the shared socioeconomic pathways and radiative forcing target (SSP2-4.5) for the future. It is particularly emphasized that poor countries which are more reliant on natural capital, will be more negatively impacted and thus exacerbate economic inequality among countries.

The study is novel and the results (if robust) will have a big impact on our climate change policies. But I am not convinced by the data and method presented in the current manuscript. More detailed explanations on the hypotheses made when deriving the natural capital and damage function, and more clarification on the choice of data and methods are needed. A better quantification and discussion of the uncertainties and potential biases of the results are also required.

We thank the reviewer for his or her positive comments on the novelty and importance of the results. In response to these comments, we have made substantial changes to the manuscript described in detail below, including a much fuller discussion of methodology and the addition of two more dynamic vegetation models to better quantify uncertainty and demonstrate robustness of the findings.

Major issues:

(1) The assumption that the area/coverage of a biome is the only factor affecting its market and non-market values (eq. 5 and 6 in the Supplementary materials) are questionable and can be misleading. Besides the area of a biome, other quantities (such as total biomass per unit area or wood density) might also affect its natural capital. These variables are not necessarily correlated with the coverage of a biome. It is possible that the biomass per unit area of a biome might increase in the future (increasing the natural capital), while its coverage decreases (reducing the natural capital).

The coverage of a biome is one of the most uncertain variables simulated in terrestrial ecosystem models, e.g., the tree cover in African Savannah (e.g. Forkel et al. 2019). Given such large uncertainties, I doubt the robustness of the high loss of natural capital of African countries (dominated by Savannah) due to biome shift in the future scenarios.

We concur with the reviewer that the area of a biome being the only factor determining market and nonmarket values is a big assumption, as it is surely not the case that each hectare of biome is homogeneous. Following the reviewer’s suggestion, we adjusted our equations to calculate natural capital to include the changes in per-area vegetation carbon content as a factor affecting the market and non-market values.

To do that, we retrieved the vegetation carbon content per pixel from three DGVMs (more on new data below) and merged each observation of the ecosystem-services database VEGS with the mean vegetation carbon content in the study area. With this data, we tested the hypothesis that vegetation carbon content impacts the value of ecosystem-services provided by the biome by including this variable in the regression used to estimate the elasticity of ecosystem benefits value as a function of spatial extent. We found that for a 1 percent increase in vegetation carbon content, the value of non-market ecosystem services increases by 0.26 percent. We did not find any statistically significant effect for the market values from ecosystem function.

In the main text, we now reflect the importance of vegetation carbon content in our analysis by adding its percentage change in Figure 2.

Fig. 2. Biome shifts and changes in area cover and vegetation carbon content. A: average percentage of grid cells covered by different biomes in the present (2016-2020). B: change in coverage under 2 degree warming projections relative to present day (using different Earth system models and representative concentration pathways). C: Changes in the fraction of a grid cell covered

by natural vegetation. D: Changes in carbon vegetation content within each grid cell (kg per square meter). Model output from LPJ-GUESS under 3 warming scenarios and 4 climate model outputs (Figures using the two other DGVMs are shown in Fig S4 and Fig S5)

We further describe the implementation of this new variable in the Supplementary Material as follows:

In addition to the change in biome cover, we test two additional mechanisms that could change the market and non-market natural capital.

Using the VEGS database we test two hypotheses: first, we test whether the marginal value of ecosystem services $es_{t,c,b}$ varies with the area of the biome. While this step is critical, given that we are valuing non-marginal changes in the natural capital stock and supply of related ecosystem services, this is an improvement over much of the literature in this area. Using the values from the VEGS database, we observe that the marginal value per hectare of ecosystems associated with non-provisioning services exhibits diminishing marginal utility (the elasticity is -0.082, p-value<0.001), while provisioning services have a constant marginal value, so the per-area market benefits $R_{t,c,b}$ do not depend on biome size (See Table S3).

Supplementary Table 3. Regression table of the ecosystem benefits elasticities, using the VEGS database.

	Dependent variable:
	log(Value per hectare)
log(cveg) Reference level: Non-market	0.265*
	(0.143)
Market	-1.679
	(1.208)
log(Area) Reference level: Non-market	-0.082***
	(0.023)
log(gdp_pc)	0.367*
	(0.198)
log(Percentage covered)	-0.357*
	(0.202)
log(cveg)*Market	0.084
	(0.215)
log(area)*Market	0.026
	(0.052)
Constant Reference Level: Non-market	2.470
	(2.117)
Observations	758
R²	0.139
Adjusted R²	0.131
Residual Std. Error	2.763 (df = 750)
F Statistic	17.365*** (df = 7; 750)
Note:	*p<0.1; **p<0.05; ***p<0.01

Also, we estimate the effect of an increase in the percentage of vegetation carbon content on the per-area benefits from ecosystems. Our regression results (Table S3) show a low elasticity of ecosystem services: for a 1 percent increase in vegetation carbon stock per hectare, the value provided by ecosystems increases by 0.265 percent. No effect was found for market benefits. Assuming no changes in country-level preferences, we estimate future non-market natural capital in year t using the equation

$$nN_{t,c} = \sum_b \frac{a_{t,c,b} eS_{c,b} * (1 + (0.265c_t - 0.082a_t) / 100)}{r} \quad \text{eq. 12}$$

where a is the percentage change in total area, and c is the percentage change in vegetation carbon content. For very large percentage changes of vegetation carbon or area of biome, the coefficients from the regression lose accuracy, as changes are no longer marginal. The relationship between the percentage increase in benefits and the percentage increase in carbon content shows a breaking point at a 40% increase in vegetation carbon content, as shown in Figure S1. Therefore, we cap the vegetation carbon changes effect at 40%

Supplementary Figure 1. Elasticity of ecosystem benefits to vegetation carbon content.

The equation we use to calculate market natural capital only changes as a function of the biome-covered area

$$mN_{t,c} = \sum_b \frac{a_{t,c,b} x_b \hat{R}_c}{r} \quad \text{eq. 13}$$

(2) The uncertainty is large when using only one terrestrial ecosystem model (i.e., LPJ-GUESS) from the Inter-Sectoral Impact Model Intercomparison Project (ISIMIP) to

derive future biome change. Several models' output are actually available from ISIMIP , and it is highly recommended to use multi-model projections to estimate the natural capital loss, so that the uncertainties and potential biases of the results can be better evaluated.

In addition, the change of biome can have significant feedback to climate which can either enhance or dampen the biome changes originally induced by climate change. The future biome changes from ISIMIP simulations only include the impact of climate on biome changes, but not the potential feedbacks between climate and biome changes. It is thus recommended to check available CMIP6 simulations with dynamic vegetation cover (see Arora et al. 2020), and use the biome changes from at least one of those simulations to estimate full potential changes in biomes and its natural capital.

The Reviewer's point is well taken. In the revised version of the manuscript, we use model output from three Dynamic Global Vegetation Models: LPJ-GUESS, ORCHIDEE-DGVM, and CARAIB. These are all the available models in ISIMIP that were run under the simulations required by our study. Such simulations consist of a combination of future climate forcing with fixed land management and land-use. We use the output from these sets of experiments to retrieve damage functions that only reflect the climatic effect on the ecosystem dynamics. This is an important feature of our damage functions, given that we build them to be used by Integrated Assessment Models (IAMs), which typically represent policy decisions (such as land use and management) in other modules.

In the Supplementary Materials, we describe the data we used as follows:

We retrieve output data from LPJ-GUESS¹, ORCHIDEE-DGVM², and CARAIB³ models the three dynamic global vegetation models (DGVMs) that participated in the Inter-Sectoral Impact Model Intercomparison Project (ISIMIP) 2b Protocol⁴ under the simulation exercise 2005soc. These simulations consist of fixing land-use to 2005 conditions and simulating the biomes' response to variables such as precipitation, daily maximum and minimum temperature, short wave downwelling radiation, surface air pressure, near-surface relative humidity, near-surface wind speed and carbon dioxide concentration, allowing us to disentangle the climate-driven effects on natural vegetation from the direct anthropogenic disturbances. Each DGVM has a biome typology as shown in Table S1. This table also shows the Earth system models (ESMs) and warming scenarios used to run each of the available simulations.

Supplementary Table 1. Dynamic Global Vegetation Models output.

DGVM	Biomes typology	GCMs	Warming Scenarios	number of runs
LPJ-GUESS (biomes=11)	Boreal needleleaved evergreen (BNE); Boreal shade intolerant needleleaved evergreen (BINE); Boreal needleleaved summergreen (BNS); Temperate broadleaved summergreen (TeBS); shade intolerant broadleaved summergreen (IBS); Temperate broadleaved evergreen (TeBE); Tropical broadleaved evergreen (TrBE); Tropical shade intolerant	HadGEM2-ES ⁵ , GFDL-ESM2M ⁶ , IPSL-CM5-LR ⁷ , MIROC5 ⁸	RCP2.6, RCP6.0, RCP8.5.	12

	broadleaved evergreen (TriBE); Tropical broadleaved raingreen (TrBR); C3 grass (C3G); C4 grass (C4G).			
ORCHIDEE (biomes=10)	Tropical broadleaved evergreen (trbrev); tropical broadleaved raingreen (trbrg); temperate needleleaf evergreen (tendev); temperate broadleaved evergreen (tebrev); temperate broadleaved summergreen (tebrsu); boreal needleleaf evergreen (bondev); boreal broadleaved summergreen (bobrsu); boreal needleleaf summergreen (bondsu); C3 natural grass (c3gra); C4 natural grass (c4gra).	GFDL-ESM2M, IPSL-CM5-LR	RCP2.6, RCP6.0	4
CARAIB (biomes=26)	C3 herbs (humid) (c3hh); C3 herbs (dry) (c3dh); C4 herbs (c4h); Broadleaved summergreen arctic shrubs (brsuas); Broadleaved summergreen boreal or temperate cold shrubs (brsutecds); Broadleaved summergreen temperate warm shrubs (brsutewms); Broadleaved evergreen boreal or temperate cold shrubs (brevtecds); Broadleaved evergreen temperate warm shrubs (brevtewms); Broadleaved evergreen xeric shrubs (brevxs); Subdesertic shrubs (sds); Tropical shrubs (trs); Needleleaved evergreen boreal or temperate cold trees (ndevtecdt); Needleleaved evergreen temperate cool trees (ndevtect); Needleleaved evergreen trees, drought-tolerant (ndevtedtt); Needleleaved evergreen trees, drought-tolerant, thermophilous (ndevtedttht); Needleleaved evergreen subtropical trees, drought-intolerant (ndevstdit); Needleleaved summergreen boreal or temperate cold trees (ndsutecdt); Needleleaved summergreen subtropical swamp trees (ndsustswt); Broadleaved evergreen trees, drought tolerant (brevdtt); Broadleaved evergreen trees, drought-tolerant, thermophilous (brevdttht); Broadleaved evergreen subtropical trees, drought-intolerant (brevstdit); Broadleaved summergreen boreal or temperate cold trees (brsutecdt); Broadleaved summergreen temperate cool trees (brsutect); Broadleaved summergreen temperate warm trees (brsutewmt); Broadleaved raingreen tropical trees (brgrt);	HadGEM2-ES, GFDL-ESM2M, IPSL-CM5-LR, MIROC5	RCP2.6, RCP6.0	8

	Broadleaved evergreen tropical trees (brevtrt).			
--	---	--	--	--

From each simulation and DGVM we retrieved the annual percentage biome-covered area and carbon vegetation content at each 0.5°x0.5° pixel.

We acknowledge the Reviewer's point that for most climate science applications, using fully coupled vegetation-climate models such as the NOAA-GFDL-ESM4, MPI-ESM1.2-LR, and UKESM1-0-LL used by Arora et al. (2020) is most desirable. However, in this *particular* setting this is less appropriate:

First, a primary goal of the study is to establish a parameterized "damage function" relationship between regional temperature change and regional ecosystem service provision. This damage function is then used within a coupled climate-economy model to understand the macro-economic effects of these changes. This model contains its own representation of the carbon cycle, which accounts for the biosphere-atmosphere carbon feedback. Using coupled models to develop a damage function for this application would therefore risk double-counting this feedback, without substantial modifications to the carbon cycle component within the climate-economy model that are outside the scope of this analysis.

Second, we would require model output from experiments simulating a warming climate with fixed land use, however, in the current available CMIP6 runs, the closest experiments we found are Land no-Lu and hist no-Lu which are both historical runs with no land use change (Lawrence et al., 2016).

(4) The fitting of the quadratic damage function (eq. 7) for different countries uses only 3 points (if I understand correctly), and there is no statistical significance reported in Supplementary Table 3.

The Reviewer was correct in pointing out that we used only a few points to fit the quadratic function. In the revised version of the manuscript we use 84 (57,28) points to estimate each damage function under LPJ-GUESS (CARAIB, ORCHIDEE) models; and a total of 169 points when estimating the damage function using all models. The increase in data comes from two sources. First, we increased the number of vegetation models and climate scenarios in the analysis. And second, we computed estimations of natural capital for each decade from 2030 to 2100, instead of only using three time horizons. We now report the p-values of the coefficients of the damage functions in Table S6 and present a graphical summary in Fig S2.

Below, we show the updated description for estimating the damage function in the Supplementary Material.

Damage function

We use the 28 model runs described in Table S1 and equations 12 and 13 to generate decadal point estimates relating country-level natural capital values and global temperature changes. Our stylized damage function models the change in natural capital as a function of temperature change. We choose a linear function, which captures the relationship fairly well, avoids non-convex damages and overly high damages estimated with a low level of confidence outside the range. Therefore, we have that

$$nN_{c,t} = nN_{c,0} (1 + \theta_{n,c} \Delta T_t) \quad \text{eq 14}$$

Normalizing $nN_{c,t}$ by the initial natural capital value $nN_{c,0}$

$$\hat{nN}_{c,t} = 1 + \theta_{n,c} \Delta T_t \quad \text{eq 15}$$

Therefore, the relative change in the normalized value is

$$\Delta \hat{nN}_{c,t} = \theta_{n,c} \Delta T_t \quad \text{eq 16}$$

We use the 196 points (28 model outputs across 7 decades) to fit the following equation to obtain the damage coefficient for each DGVM

$$\Delta \hat{nN}_{c,t,dgvm} = 0 + \theta_{n,c,dgvm} \Delta T_t + \alpha_{clim} + \alpha_{scen} + \epsilon_{t,r} \quad \text{eq 17}$$

where α_{clim} and α_{scen} are fixed effects controlling for the four Earth system models, and the three warming scenarios. Similarly, we obtain $\theta_{n,c,dgvm}$ for the market natural capital. As shown in Figure S2, the majority of the countries have damage estimates with p-values lower than 0.01.

Supplementary Fig 2. P-values of the market and non-market natural capital damage coefficients. The theta estimates are reported in Table S6.

(5) There is no detailed explanation on how the random forest is applied to predict the ecosystem service values, and how accurate the random forest approach is. This is a critical step for estimating country level market and non-market natural capital. A clear explanation is needed.

The Reviewer's point is well taken. We acknowledge that we had not provided adequate description of the random forest procedure. We have added a section in the supplementary materials to describe how it is applied, how accurate it is, and how we use it to obtain the damage function coefficients. The Supplementary Material has been updated as follows:

Random Forest

We use the database subsets $V_{m,b}$ and $V_{n,b}$ to train random forests ($RF_{m,b}$ and $RF_{n,b}$) for each country. A random forest consists of a collection of decision trees that predict a dependent variable using a series of optimal subdivisions in the data. We build our random forests to predict the log value of the ecosystem benefits in $V_{m,b}$ and $V_{n,b}$ based on the variables listed in Table S2, such that

$$RF_b = f(cover_b, GDPpc, cveg, PercCovered) \quad \text{eq. 6}$$

We create random forests with 300 decision trees as the decrease in the root mean squared error by each additional decision tree in the random forest reaches saturation at that point. We create one random forest for each combination of 177 countries, 2 types of benefits (market and non-market), and 3 typologies of biomes (from the 3 DGVMs), giving 1067 random forests.

We use the random forest to predict a baseline ecosystem benefit value. Then, by increasing each biome cover by 10% one by one, we obtain the marginal effect relative to the baseline ecosystem benefit value. We can therefore compare the relative sizes of marginal effects. To illustrate the procedure without loss of generality, we imagine a hypothetical country with only two biomes: B1 and B2. Using the relative sizes of the marginal effects given by the random forest $RF_{n,b}$, we can write $es_{2018,c,B2}$ in terms of $es_{2018,c,B1}$

$$es_{2018,c,B2} = x_{n,c,B1} es_{2018,c,B1} \quad \text{eq. 7}$$

Rewriting equation 3 for the hypothetical country in 2018,

$$nN_{2018,c} = \frac{a_{2018,c,B1} es_{2018,c,B1} + a_{2018,c,B2} es_{2018,c,B2}}{r} \quad \text{eq. 8}$$

$$\Rightarrow r * nN_{2018,c} = a_{2018,c,B1} es_{2018,c,B1} + a_{2018,c,B2} es_{2018,c,B2} \quad \text{eq. 9}$$

Substituting $es_{2018,c,B2}$ from equation 7

$$r * nN_{2018,c} = a_{2018,c,B1} es_{2018,c,B1} + a_{2018,c,B2} x_{n,c,B1} es_{2018,c,B1} \quad \text{eq. 10}$$

$$\Rightarrow es_{2018,c,B1} = \frac{r * nN_{2018,c}}{a_{2018,c,B1} + a_{2018,c,B2} x_{n,c,B1}} \quad \text{eq. 11}$$

All the variables in the right-hand side of equation 11 are known so we can estimate $es_{2018,c,B1}$. Similarly, we estimate $R_{2018,c,b}$. In the following section, we discuss mechanisms that can change the per-area benefits $es_{2018,c,b}$ and $R_{2018,c,b}$ when using them to calculate natural capital in future time steps.

(3) Please explain why use SPP2-4.5 scenario for economic projections, while using RCP6.0 for biome changes? Are they consistent with each other?

We now use vegetation model output under RCP2.6, RCP6.0, and RCP8.5 scenarios, modeled by four Earth system models (HadGEM2-ES, GFDL-ESM2M, IPSL-CM5-LR, MIROC5). With this extension of our original methodology, we expect to capture a large range of temperature variability and temperature change rates. For the projections, we now use SSP2-6.0, which is the projection that uses the shared socioeconomic storyline 2 and the representative concentration pathway 6, making our results consistent with our methods.

Specific comments:

Abstract: need to explain “market and non-market benefits” and “SSP2-4.5”

We have now spelled out the meaning of SSP and explained the meaning of market and non-market ecosystem benefits in the following way in the abstract: Ecosystems bring benefits to human well-being, including both those with market value, that can be bought and sold, and those without, that are intangible and non-monetary in nature.

Page 2: Please explain “non-use values” and “inclusive wealth”.

We have now included an explanatory sentence for inclusive wealth: In the inclusive wealth literature, human well-being is understood as arising from a stock of valuable assets (or types of wealth) that produce a flow of benefits to people. This literature has focused on measuring increasingly comprehensive estimates of wealth and conducting backward-looking assessments of whether this inclusive wealth – defined as human capital, natural capital, and the more standard manufactured capital – has declined over time or not, as this is one of the criteria for inter-temporal sustainability¹⁰⁻¹⁴.

And an explanatory sentence for non-use values: Under the welfare economics framework, human well-being is divided into three components: goods and services people can buy in a market economy (hereafter known as market goods and services), benefits we get from nature that are not usually exchanged in markets (hereafter known as non-market ecosystem benefits), and non-use values from ecosystems attached only to its existence (Fig. 1).

Figure 2A: Please explain why “most of the globe loses natural vegetation cover by the end of the century” ? Do “biomass” or “LAI” have the same trend?

We have now been more explicit when referring to changes in ecosystems, in particular we note that: Figure 2C shows the percentage change in vegetation cover accounting for all plant functional types within a gridcell in a world two degrees warmer. Most tropical regions are expected to see a net loss of natural vegetation-covered area, while higher latitudes and central Asia will have net gains. Regardless of the changes in area cover, the whole world is projected to gain vegetation carbon content per square meter (Figure 2D), which we include in our analysis as it will likely affect the flow of ecosystem benefits to people.

Page 4: Please add a reference for the VEGS database?

Reference has been added.

Page 5: When talking about “damage function”, please refer to “eq. 7 in Supplement”

We have now added the reference to the equation when talking about the damage function.

Page 6: “equation 1” should be “equation 1 in Supplement”

We now specify that the equations are to be found in the supplement.

Figure 4: Please refer to supplementary materials when mentioning “four damage functions estimated with different general circulation model outputs”.

We have now pointed the reader to the Supplementary Materials when needed.

Page 7: citation number 24 is wrongly used. Please check!

Thank you for spotting this mistake, it has been corrected.

Page 8: What does “SCC” mean?

We have now spelled out “social cost of carbon”

Eq.2: What is the meaning of “r” in the equation? Please explain.

We have now explained that:

Therefore, equation 1 can be written as

$$N(t) = \sum_i \sum_t \frac{B(t)}{(1+r)^t} \quad \text{eq. 2}$$

where $B_i(t)$ is the total yearly flow of benefits from the natural asset i at time t , and r is the consumption discount rate.

Page 2 in Supplement, bottom 2 line: “The country-level Xb values are used....” should be “The country-level At, c, b values are used....”

We have now rewritten that section to be more specific and carefully following correct indexation.

Page 3 in Supplement: “See Supplementary Figure 5”? There is no Figure 5 in the Supplementary file.

We have solved that mistake.

Page 3 in Supplement: “time time” should be “time”.

We have fixed that typo.

References:

Forkel, M., Drüke, M., Thurner, M. et al. Constraining modelled global vegetation dynamics and carbon turnover using multiple satellite observations. *Sci Rep* 9, 18757 (2019). <https://doi.org/10.1038/s41598-019-55187-7>

Arora, V. K., Katavouta, A., Williams, R. G., Jones, C. D., Brovkin, V., Friedlingstein, P., Schwinger, J., Bopp, L., Boucher, O., Cadule, P., Chamberlain, M. A., Christian, J. R., Delire, C., Fisher, R. A., Hajima, T., Ilyina, T., Joetzjer, E., Kawamiya, M., Koven, C. D., Krasting, J. P., Law, R. M., Lawrence, D. M., Lenton, A., Lindsay, K., Pongratz, J., Raddatz, T., Séférian, R., Tachiiri, K., Tjiputra, J. F., Wiltshire, A., Wu, T., and Ziehn, T.: Carbon–concentration and carbon–climate feedbacks in CMIP6 models and their comparison to CMIP5 models, *Biogeosciences*, 17, 4173–4222, <https://doi.org/10.5194/bg-17-4173-2020>, 2020.

Referee #2 (Remarks to the Author):

Summary

This paper estimates the economic value of changes in terrestrial vegetation caused by climate change over the remainder of this century. In the paper’s framework, terrestrial vegetation – forests and grassland – provides two kinds of economic value. First, forests produce timber, which is a market good. Thus, forests embody “market natural capital”. Second, forests and grassland produce a wider range of ecosystem services that are not priced in markets, such as recreation and watershed protection. Thus, forests also embody “non-market natural capital”. A global vegetation model (LPJ-Guess) is used to simulate climate-induced changes in the spatial extent of terrestrial vegetation types (biomes). A database of ecosystem service values (VEGS)

is used to estimate the relationship between natural capital values (market and non-market) and biome area, for each biome and country. Then, the 21st century simulations from the vegetation model can be translated into country-level estimates of changing natural capital and ecosystem services. The authors find that climate change causes non-market ecosystem service flows from terrestrial vegetation to fall substantially in many countries and globally on average. They find smaller losses in market ecosystem service flows (timber output), but significant heterogeneity and apparently large impacts in some countries/regions, notably Africa.

Major comments

Overall assessment

I think this paper makes a significant contribution to the literature that tries to understand the economic impacts of climate change. This literature is better developed when it comes to 'market impacts' and physical/human capital than it is when it comes to natural capital. Previous studies trying to get a handle on economic impacts of climate change via natural capital have arguably suffered from being too aggregated and stylized to provide numbers that one could have any confidence in. Thus, in my view more detailed modeling is the way forward here and this is exactly what the authors do in this paper by coupling a global vegetation model with high spatial resolution with detailed ecosystem service value data.

However, I have concerns about the methods – really one major concern – and whether the scope of the contribution is large enough to warrant publication in Nature. I also found the paper confusing in places and have some editorial suggestions.

We thank the reviewer for his or her positive comments on the significance of the contribution. We have done a major improvement in the methodology that specifically addresses your principal concern regarding the evolution of the economies in our methods. In a major new analysis, the revised manuscript integrates the estimated damage functions into the RICE50+ model, a multi-regional integrated assessment model that calibrates the economic growth and greenhouse-gas abatement cost curves for more than 50 regions (Gazzotti et al., 2021).

Growth and development in the 21st century

Here is my major concern about the methods. I found the methods section difficult to follow (more on that below), but my understanding is that long-term climate-induced changes in biome area are superimposed on a world economy that is more like today's than the one we would expect to see in 2100. The extent to which it is more like today's than that of 2100 is difficult to figure out (unless I overlooked it, we are never shown what happens to GDP), but the reason I say this is that the two key engines of long-term growth, human capital and TFP, are fixed at their 2018 levels. Physical capital is not fixed, although the paper reports a savings rate of 3% (is this a typo? Should it be 30%?), which means that with a depreciation rate of 10% the

physical capital stock could even be shrinking over time. In any case, assuming the savings rate is not as low as 3%, physical capital accumulation is not a major driver of growth in the long term, except perhaps in low-income countries.

This matters in several respects. If the economy grows, then a given value of lost ecosystem services in the future is a smaller share of future GDP. The authors place a lot of emphasis on their results reported in terms of percentages of GDP. These may appear larger than they would really be in 2100, because they are compared with the economy of today. Moreover, with a growing economy the value of lost ecosystem services is not given, rather it is a function of incomes and growing capital stocks. Therefore, I find the analysis fundamentally incomplete. An article appearing in a top journal needs to explicitly model the interdependence of capital stocks, incomes and ecosystem service valuations.

The Reviewer's point is well taken. We do acknowledge that in the original manuscript the projections of the world economies were missing the dynamics of two important drivers of growth: human capital and TFP. And we thank the reviewer for pointing out the typo in the savings rate.

In the revised version of this work, we use RICE50+, a regionally explicit Integrated Assessment Model (IAM) that calibrates economic growth for more than 50 regions under distinct future socio-economic scenarios. Following the model GreenDICE (Bastien-Olvera & Moore, 2021), we modified three key parts of the IAM: the production function, the damage function, and the utility function. We call GreenRICE50+ to this new version of the model that combines the natural capital insights in GreenDICE, and the regional dynamics and economic calibrations in RICE50+. In RICE50+ TFP and labor are exogenous and calibrated to match the population and economic output from the Shared Socioeconomic Pathway 2 (SSP2).

In the methods section we fully describe this implementation.

Clarity of exposition and detail of methods section

As mentioned above, I had a hard time following the detail of the methods section and came away with the sense that I did not fully understand the procedures that the authors followed. This comment is broader though – I think that the clarity of the exposition could be improved throughout. In the main text, I initially had a hard time understanding which ecosystem services were included in the analysis and which were not. I venture that one of the reasons for my confusion is that this area of scholarship contains many partially overlapping frameworks, including capital theory, typologies of ecosystem services (provisioning, regulating, cultural and their sub-components), and the theory of economic value (market versus non-market). The main text initially mixes these concepts in a way that is not fully clear. I urge the authors to attempt some redrafting of the opening couple of pages that makes it clearer how all these frameworks relate to each other so that the reader can quickly zero in on what the analysis includes and what it leaves out.

We thank the Reviewer for this comment and acknowledge the challenges that we face in describing the multiple intellectual frameworks being drawn on in this highly integrative analysis. Following the Reviewer's advice, we have reordered and rewritten the majority of the first two pages of the main text to carefully lay out the concepts and frameworks that we use, as well as how they relate to each other. Further, we paid special attention to the paragraph that describes market and non-market natural capital to clarify what we mean when referring to those terms. That part of the main manuscript now reads:

Natural capital is characterized by the stock of physical biomass and the value of the benefits that this stock provides. The inclusive wealth accounting literature has disaggregated natural capital valuation based on whether the stock is renewable and whether the flow of benefits is traded in the market⁴. Our analysis focuses on natural capital associated with the market and non-market benefits of forests and terrestrial protected areas, as those are the estimates available in the World Bank (WB) inclusive wealth accounts, and dynamic global vegetation models can capture the changes of the underlying ecosystems. The WB uses a diverse set of economic concepts and frameworks to estimate natural capital. On the one hand, natural capital associated with market-based benefits from forests (mN) is estimated using the present value of the future timber revenues. On the other hand, natural capital embedded in forests and protected areas associated with non-market ecosystem benefits (nN) is estimated using the ecosystem services framework and the option value concept in economics. Specifically, WB conducts a meta-analysis of recreation services, water quality, water quantity and non-timber forest products and uses the results to estimate these services in each country. Further, the value from protected areas is estimated by the unrealized revenue had the areas been converted to agricultural fields, which gives a lower bound estimate of the value of the protected areas^{10,18,19}. Hereafter, the terms market and non-market natural capital will refer only to the benefits described above, as it is the available data for now, however, the methodology remains valid for future additions to natural capital estimates, and the results of this work should be regarded as likely lower bound estimates.

Turning to the methods section, I do feel this needs improving. Some of the notation is never spelled out (e.g., r , the consumption discount rate). All variable and parameter symbols should be clarified in prose. Some steps require further intuition, for example it will help readers to explain that equations 3 and 4 use the formula for the PV of a perpetuity to value the initial capital stock. Some notation is confusing. For example, a , e s and R are indexed with time in equations 3 and 4, yet they are assumed to be time-invariant so that the simple formula for the PV of a perpetuity can be used. In equations 5 and 6, the vector of fractions of each biome is given the same notation, $x_{\{b\}}$, even though I assume the elements of the vector are different for the two capital stocks.

We thank the Reviewer for pointing out these confusing or incomplete components of the methods section. We have greatly improved the Supplementary Material and included section headings to guide the readers through the relevant steps of our methods. We spell

out the consumption discount rate and explain the use of the present value formula for discounting in perpetuity:

Following World Bank assumptions of no forest area change in the future and a constant per hectare value of benefits over 100 years, we can rewrite equation 2 using the formula for the present value (PV) of a perpetuity ($PV = FV/r$), where FV is the constant future value obtained each year, and r is the discount rate, which we set at 3% following the World Bank methodology. Therefore, the non-market natural capital value calculated in year t is given by

$$nN_{t,c} = \frac{A_{t,c} ES_{t,c}}{r} \quad \text{eq. 3}$$

Also, we clarify the use of t as an index that indicates the values used in a certain year to calculate natural capital.

Where $ES_{t,c}$ is the value per-area of ecosystem benefits estimated in year t for country c , and $A_{t,c}$ is the area covered by the ecosystems at year t for country c . While both ES_t and A_t are assumed to remain constant over time for the calculation of $nN_{t,c}$ at a particular year t , we allow such values to change everytime natural capital is computed over different years. Disaggregating ecosystem services and areas by types of biome (b) we get

$$nN_{t,c} = \sum_b \frac{a_{t,c,b} es_{t,c,b}}{r} \quad \text{eq. 4}$$

Where $es_{t,c,b}$ is the value of the ecosystem services flow provided by a hectare of biome b in country c at the year of the calculation t , $a_{t,c,b}$ is the area coverage of biome b in country c at the year of the calculation t . Similarly, the market natural capital value is given by

$$mN_{t,c} = \sum_b \frac{a_{t,c,b} R_{t,c,b}}{r} \quad \text{eq. 5}$$

Where $R_{t,c,b}$ is the value of timber products provided by a hectare of each biome b in country c estimated at time t . To be able to use equations 4 and 5 to estimate future natural capital based on the area of the biomes, we first need to estimate $es_{t,c,b}$ and $R_{t,c,b}$ and assume that those marginal values will mostly remain constant in the future (See section *Mechanisms that affect value per hectare*). To do that, we focus on the year 2018, which are the most recent country-level estimates of market and non-market natural capital from the World Bank. Also, we can retrieve the biome-covered area from the three dynamic global vegetation models for that year. In the following sections, we show how we use an ecosystem service valuation database and a random forest to estimate $es_{2018,c,b}$ and $R_{2018,c,b}$:

We corrected the notation of the vectors $x_{\{b\}}$ to include the subindex m or n indicating whether they are used to calculate market or non-market benefits from ecosystems

Using the relative sizes of the marginal effects given by the random forest $RF_{n,b}$, we can write $es_{2018,c,B2}$ in terms of $es_{2018,c,B1}$

$$es_{2018,c,B2} = x_{n,c,B1} es_{2018,c,B1} \quad \text{eq. 7}$$

The explanation of the important stage of regressing ecosystem service values on the area covered by each biome using a random forest is not explained in enough detail. It is for example unclear whether additional explanatory variables such as the income of the sample population are controlled for.

The Reviewer's point is well taken. We do include income as an additional variable in the random forest. In the new manuscript, we have revised the random forest section of the methodology as follows

Random Forest

We use the database subsets $V_{m,b}$ and $V_{n,b}$ to train random forests ($RF_{m,b}$ and $RF_{n,b}$) for each country. A random forest consists of a collection of decision trees that predict a dependent variable using a series of optimal subdivisions in the data. We build our random forests to predict the log value of the ecosystem benefits in $V_{m,b}$ and $V_{n,b}$ based on the variables listed in Table S2, such that

$$RF_b = f(cover_b, GDPpc, cveg, PercCovered) \quad \text{eq. 6}$$

We create random forests with 300 decision trees as the decrease in the root mean squared error by each additional decision tree in the random forest reaches saturation at that point. We create one random forest for each combination of 177 countries, 2 types of benefits (market and non-market), and 3 typologies of biomes (from the 3 DGVMs), giving 1067 random forests.

We use the random forest to predict a baseline ecosystem benefit value. Then, by increasing each biome cover by 10% one by one, we obtain the marginal effect relative to the baseline ecosystem benefit value. We can therefore compare the relative sizes of marginal effects. To illustrate the procedure without loss of generality, we imagine a hypothetical country with only two biomes: B1 and B2. Using the relative sizes of the marginal effects given by the random forest $RF_{n,b}$, we can write $es_{2018,c,B2}$ in terms of $es_{2018,c,B1}$

$$es_{2018,c,B2} = x_{n,c,B1} es_{2018,c,B1} \quad \text{eq. 7}$$

Rewriting equation 3 for the hypothetical country in 2018,

$$nN_{2018,c} = \frac{a_{2018,c,B1} es_{2018,c,B1} + a_{2018,c,B2} es_{2018,c,B2}}{r} \quad \text{eq. 8}$$

$$\Rightarrow r * nN_{2018,c} = a_{2018,c,B1} es_{2018,c,B1} + a_{2018,c,B2} es_{2018,c,B2} \quad \text{eq. 9}$$

Substituting $es_{2018,c,B2}$ from equation 7

$$r * nN_{2018,c} = a_{2018,c,B1} es_{2018,c,B1} + a_{2018,c,B2} x_{n,c,B1} es_{2018,c,B1} \quad \text{eq. 10}$$

$$\Rightarrow es_{2018,c,B1} = \frac{r^*nN_{2018,c}}{a_{2018,c,B1} + a_{2018,c,B2}x_{n,cB1}} \quad \text{eq. 11}$$

All the variables in the right-hand side of equation 11 are known so we can estimate $es_{2018,c,B1}$. Similarly, we estimate $R_{2018,c,b}$. In the following section, we discuss mechanisms that can change the per-area benefits $es_{2018,c,b}$ and $R_{2018,c,b}$ when using them to calculate natural capital in future time steps.

There is next to no description of how LPJ-Guess works – we shouldn't have to cross-refer to find out everything.

We thank the Reviewer for pointing out this lack of detail about the LPJ-Guess model included in the original manuscript. In the revised manuscript, we now give more in-detail description of the dynamic global vegetation models:

We retrieve ecosystem cover projections under future climate change scenarios using the LPJ-GUES²⁷, ORCHIDEE-DGVM²³, and CARAIB²⁴ models, three process-based vegetation-terrestrial ecosystem models that use output from climate models, such as daily minimum and maximum temperatures, total precipitation, short wave downwelling radiation, and humidity to simulate the establishment, competition and mortality of natural vegetation. These models simulate the dynamics of 11, 10, and 26 plant functional types (PFTs)²⁵ or biomes across the world, respectively.

Further, the Supplement materials has a new section that describes the model outputs in more detail:

DGVMS model output

We retrieve output data from LPJ-GUESS¹, ORCHIDEE-DGVM², and CARAIB³ models the three dynamic global vegetation models (DGVMs) that participated in the Inter-Sectoral Impact Model Intercomparison Project (ISIMIP) 2b Protocol⁴ under the simulation exercise 2005soc. These simulations consist of fixing land-use to 2005 conditions and simulating the biomes' response to variables such as precipitation, daily maximum and minimum temperature, short wave downwelling radiation, surface air pressure, near-surface relative humidity, near-surface wind speed and carbon dioxide concentration, allowing us to disentangle the climate-driven effects on natural vegetation from the direct anthropogenic disturbances. Each DGVM has a biome typology as shown in Table S1. This table also shows the Earth system models (ESMs) and warming scenarios used to run each of the available simulations.

Supplementary Table 1. Dynamic Global Vegetation Models output.

DGVM	Biomes typology	GCMs	Warming Scenarios	number of runs
LPJ-GUESS (biomes=11)	Boreal needleleaved evergreen (BNE); Boreal shade intolerant needleleaved evergreen (BINE); Boreal needleleaved summergreen (BNS); Temperate broadleaved summergreen (TeBS); shade intolerant broadleaved summergreen (IBS); Temperate	HadGEM2-ES ⁵ , GFDL-ESM2M ⁶ , IPSL-CM5-LR ⁷ , MIROC5 ⁸	RCP2.6, RCP6.0, RCP8.5.	16

	broadleaved evergreen (TeBE); Tropical broadleaved evergreen (TrBE); Tropical shade intolerant broadleaved evergreen (TriBE); Tropical broadleaved raingreen (TrBR); C3 grass (C3G); C4 grass (C4G).			
ORCHIDEE (biomes=10)	Tropical broadleaved evergreen (trbrev); tropical broadleaved raingreen (trbrgg); temperate needleleaf evergreen (tendev); temperate broadleaved evergreen (tebrev); temperate broadleaved summergreen (tebrsu); boreal needleleaf evergreen (bondev); boreal broadleaved summergreen (bobrsu); boreal needleleaf summergreen (bonsu); C3 natural grass (c3gra); C4 natural grass (c4gra).	GFDL-ESM2M, IPSL-CM5-LR	RCP2.6, RCP6.0	4
CARAIB (biomes=26)	C3 herbs (humid) (c3hh); C3 herbs (dry) (c3dh); C4 herbs (c4h); Broadleaved summergreen arctic shrubs (brsuas); Broadleaved summergreen boreal or temperate cold shrubs (brsutecds); Broadleaved summergreen temperate warm shrubs (brsutewms); Broadleaved evergreen boreal or temperate cold shrubs (brevtecds); Broadleaved evergreen temperate warm shrubs (brevtewms); Broadleaved evergreen xeric shrubs (brevxs); Subdesertic shrubs (sds); Tropical shrubs (trs); Needleleaved evergreen boreal or temperate cold trees (ndevtecdt); Needleleaved evergreen temperate cool trees (ndevtect); Needleleaved evergreen trees, drought-tolerant (ndevtedtt); Needleleaved evergreen trees, drought-tolerant, thermophilous (ndevtedttht); Needleleaved evergreen subtropical trees, drought-intolerant (ndevstdit); Needleleaved summergreen boreal or temperate cold trees (ndsutecdt); Needleleaved summergreen subtropical swamp trees (ndsustswt); Broadleaved evergreen trees, drought tolerant (brevdtt); Broadleaved evergreen trees, drought-tolerant, thermophilous (brevdtttht); Broadleaved evergreen subtropical trees, drought-intolerant (brevstdit); Broadleaved summergreen boreal or temperate cold trees (brsutecdt); Broadleaved summergreen temperate cool trees (brsutect); Broadleaved summergreen temperate	HadGEM2-ES, GFDL-ESM2M, IPSL-CM5-LR, MIROC5	RCP2.6, RCP6.0	8

	warm trees (brsutewmt); Broadleaved raingreen tropical trees (brgtrt); Broadleaved evergreen tropical trees (brevtrt).			
--	---	--	--	--

From each simulation and DGVM we retrieved the annual percentage biome-covered area and carbon vegetation content at each 0.5°x0.5° pixel.

It is not clear whether human capital in equation 8 is simply the labor force (as in DICE, referred to in the main text), or a broader notion of the totality of people’s skills and knowledge. It is not then clear what TFP is measuring. And so on.

We agree with the Reviewer regarding the lack of clarity between labor and human capital in the original manuscript. We now clarify that the economic growth model uses labor in the production function and is calibrated to match the population and GDP in the shared socioeconomic pathways, but that the regression model that we use to estimate the elasticity of labor uses human capital as given by the World Bank to be consistent in terms of units and methodology with the other factors, such as mN and nN.

In the supplementary materials we now have included:

GDP is computed for each region i with a Cobb-Douglas production function of labor $L_{t,c}$, capital $K_{t,c}$, market natural capital $mN_{t,c}$, with total factor productivity $TFP_{t,c,\dots}$

...

$$GDP_{t,c} = TFP_{t,c} * S_t^{b_r} * L_{t,c}^{\gamma_{1c}} * K_{t,c}^{\gamma_{2c}} * mN_{t,c}^{\gamma_{3c}} \quad \text{eq. 18}$$

TFP and labor are exogenous and calibrated to match the population and economic output from the Shared Socioeconomic Pathway 2 (SSP2).

The production elasticity to labor γ_1 and the regionally-calibrated elasticity to global environmental good b_r are estimated by the following GDP-weighted panel regression model, noting that instead of labor, in this regression model we use human capital (H) as estimated by the World Bank to be consistent with the other capital estimates (mN and nN):

$$\log(GDP_{t,c}) = R * \hat{b}_r \log(S_t) + \hat{\gamma}_1 \log(H_{t,c}) + \hat{\gamma}_2 \log(K_{t,c}) + \hat{\gamma}_3 \log(mN_{t,c}) + \theta_t + \theta_c + \epsilon_{t,r} \quad \text{eq. 19}$$

Scope of the analysis

Eventually I felt that I understood what was in and what was out. The discussion is very clear on this, but some passages earlier in the paper are arguably guilty of overselling how much further this analysis takes us compared to the existing literature. As described above, it essentially estimates the economic value of lost timber output plus presumably a subset of non-market benefits that are also provided by forests and grasslands (what subset is never clear, but I assume the VGES dataset is incomplete). This is a useful step forward but still leaves a lot of ground to cover, and sometimes the paper gives the impression that it is undertaking a much bigger analysis of comprehensive ecosystem services. Relatedly, the boundary with the existing literature is fuzzy because some models and analyses (e.g., the FUND model) already explicitly include impacts in the forestry sector, while macro-economic studies should implicitly include historical impacts on commercial forestry.

We concur with the reviewer that the language used in the earlier passages was not clear enough to describe the scope of the analysis. In the revised version of the manuscript, we clarify exactly what we mean by key concepts such as natural capital, market and non-market values, ecosystem services, and more.

We slightly disagree with the Reviewer's view that our contribution essentially estimates the economic value of lost timber output and a subset of non-market benefits.

In this revised version of the manuscript we have included a global environmental public good in the economic production in addition to the value of timber as provisioning raw materials. This equation reflects how the economy is embedded in the biosphere as pointed out theoretically in the Dasgupta Review (2021). To our knowledge, this is the first global study that implements such view of economic growth. We describe this in the main manuscript and the supplementary materials:

GDP is computed for each region i with a Cobb-Douglas production function of labor $L_{t,c}$, capital $K_{t,c}$, market natural capital $mN_{t,c}$, with total factor productivity $TFP_{t,c}$, and bounded by a global value environmental good in which economy is embedded $S_t = \sum_c nN_{c,t}$, following the Dasgupta Review²¹

$$GDP_{t,c} = TFP_{t,c} * S_t^{b_r} * L_{t,c}^{\gamma_{1c}} * K_{t,c}^{\gamma_{2c}} * mN_{t,c}^{\gamma_{3c}} \quad \text{eq. 18}$$

TFP and labor are exogenous and calibrated to match the population and economic output from the Shared Socioeconomic Pathway 2 (SSP2).

The production elasticity to labor γ_1 and the regionally-calibrated elasticity to global environmental good b_r are estimated by the following GDP-weighted panel regression

model, noting that instead of labor, in this regression model we use human capital (H) as estimated by the World Bank to be consistent with the other capital estimates (mN and nN):

$$\log(GDP_{t,c}) = R * \hat{b}_r \log(S_t) + \hat{\gamma}_1 \log(H_{t,c}) + \hat{\gamma}_2 \log(K_{t,c}) + \hat{\gamma}_3 \log(mN_{t,c}) + \theta_t + \theta_c + \epsilon_{t,r}$$

eq. 19

where R is a dummy variable for each of the following five regions in RICE50+: OECD (the OECD 90 countries and the European Union member states and candidates), LAM (Latin America and the Caribbean) REF (the reforming economies of Eastern Europe and the Former Soviet Union), ASIA (Asian countries except the Middle East, Japan and the Former Soviet Union states) and MAF (the Middle East and Africa); θ_t and θ_c are year and country fixed effects and standard errors are clustered by region.

...

The global value S_t is obtained by the regional sum of $nN_{c,t}$ and take into account damages computed based on the preceding section.

Another key aspect of our model is that while market natural capital is measured with the present value of timber, our framework uses natural capital as an input variable in the economic production function. Models such as FUND that already explicitly include impacts in the forestry sector express such damages as part of the damage function of the economic output, essentially treating timber as a market good that is perfectly substitutable with any other good or service in the market.

Further, we have now clarified that non-market natural capital in the World Bank estimations includes recreation services, water quality, water quantity, non-timber forest products and the value of protected areas. In the main text we added:

The WB uses diverse economic concepts and frameworks to estimate natural capital. On the one hand, natural capital associated with market-based benefits from forests (mN) is estimated using the present value of future timber revenues. On the other hand, natural capital embedded in forests and protected areas associated with non-market ecosystem benefits (nN) is estimated using the ecosystem services framework and the option value concept in economics (i.e. the foregone value of alternative economic activities). Specifically, WB conducts a meta-analysis of recreation services, water quality, water quantity, and non-timber forest products and uses the results to estimate these services in each country. Further, the value from protected areas is estimated by the unrealized revenue had the areas been converted to agricultural fields, giving a lower bound estimate of the protected areas' value^{10,18,19}. Hereafter, the terms market and non-market natural capital will refer only to the benefits described above, as it is the available data for now. However, the methodology remains valid for future additions to natural capital estimates, and the results of this work should be regarded as likely lower-bound estimates.

Further, the VEGS database, which is used to apportion World Bank values to specific biomes, has a comprehensive coverage of ecosystem services. We now further specify the range of services in the Supplementary Information:

VEGS database

To get $es_{2018,c,b}$ and $R_{2018,c,b}$ in equations 3 and 4, we use the Valuation of Ecosystem Goods and Services (VEGS) database¹⁸. The database curates findings and the contextual parameters that factor into the ecosystem service production into a standardized value per hectare per year, attributed to specific geographies. It includes 21 different ecosystem service types based on De Groot's framework. This database records 4300 ecosystem services' annual values per hectare, however, we only use a subset that excludes meta analyses, value transfer studies, crop valuation studies, and observations that are pending revision. Therefore, we end up with a dataset (V) of 882 values (See Table S7).

Supplementary Table 7. Number of observations in the VEGS database by Ecosystem Service Subcategory under three categories: Information (n=178), Provisioning (n=152), Regulating (n=474), Supporting (n=81).

Aesthetic Information	24
Air Quality	21
Biological Control	26
Climate Stability	99
Cultural Value	33
Disaster Risk Reduction	59
Energy & Raw Materials	116
Habitat	81
Medicinal Resources	3
Ornamental Resources	4
Pollination & Seed Dispersal	30
Recreation & Tourism	82
Science & Education	39
Soil Formation	21
Soil Quality	32
Soil Retention	32
Water Capture, Conveyance, & Supply	67
Water Quality	87

Previous IAMs have not given much consideration to the non-market benefits that ecosystems provide to human well-being. This is a significant limitation of traditional IAMs, as non-market benefits make a significant impact on the quality of life for individuals and communities.

Minor comments

Why use RCP6.0 in Figure 2 but SSP2-4.5 in Figure 4? Should you not use the same scenario throughout?

We agree with the Reviewer with the lack of consistency between scenarios in the first version of the manuscript. Following Reviewer 1's suggestions, in the revised version of the manuscript we increased the DGVMs output data used to estimate the damage function. Now we retrieve biome changes under the scenarios RCP2.6, RCP6.0 and RCP8.5. The objective is to sample a wide enough range of temperature changes to try to capture some of the possible non-linear responses of ecosystems to climate change and estimate a robust damage function that could be used under any warming scenario. We now project our main set of results under SSP2-6.0 which is consistent with the central RCP6.0 used for the vegetation modeling.

Referee #3 (Remarks to the Author):

A. Summary of the key results

- Impacts at country-level of market and non-market ecosystem services value under climate change scenarios.
- Changes in GDP and ecosystem service value trajectories under the SSP2-4.5 warming scenario with respect to the baseline scenario in which natural capital is not damaged by climate change.
- Many countries will lose a large proportion of the value of their ecosystem services with respect to the baseline. The global average change in ecosystem service value by 2100 is -20.4% relative to the baseline (no damaged natural capital).
- Changes in non-market ecosystem services are not strongly related to either the country's per-capita income or geographic region that is they are equally damaged by climate change.
- Changes in the market benefits of terrestrial ecosystems, measured in terms of GDP losses are smaller in magnitude but more strongly differentiated by region. The global average change in GDP by 2100 is -0.8% with respect to the 2100 levels of the

baseline simulation the equivalent to -1.5% of GDP in 2018. Africa losses 14.4% of Africa's GDP in 2018 while Europe losses by 2100 the 1.2% of Europe's GDP in 2018.

- These results suggest that climate change effects on ecosystems could further increase economic inequality between regions by damaging a stock of wealth particularly important in low and lower-middle-income countries.

- A further simulation shows that limiting warming below 2°C would cut the impacts of GDP by a third (global mean change by 2100: -0.3%) and the impacts on ecosystem services by half (global mean change by 2100: -9.4%) compared to the SSP2-4.5 that puts the planet at 2.6°C of warming by 2100 with respect to preindustrial levels.

B. Originality and significance: if not novel, please include reference

I think that the conclusions are not original. This is one more paper that support that developing countries would be more affected in terms of ecosystem services loss (market ecosystems-value) due to climate change. However, the authors rely on a linear relationship between GDP and ecosystem services.

The paper is an example of the fashion- science- research which uses global models to get global obvious results. The manuscript relies on very general assumptions that cannot be fully supported. If people agree that these approaches are useful, then the paper could be easily published. Nevertheless, I would not go for oversimplifying the ecosystem services and ecosystem loss only as function based on the GDP.

We thank the Reviewer for her or his suggestions to our manuscript. Before addressing the Reviewer's particular concerns below, we make an important clarification. In our work we model two separate components of natural capital. On the one hand, we estimate the market value embedded in forests using timber revenues (collected by the World Bank). On the other hand, in a separate analysis, we estimate the non-market values embedded in natural ecosystems associated with recreation services, water quantity, water quality, non-timber forest products, and the overall value of protected areas (more on methods below). In other words, the "ecosystem services and ecosystem loss only as function based on the GDP" is not part of our methodology. We do acknowledge that the original manuscript could be confusing when describing this, and we have now rewritten the first couple of pages to make this clear:

Natural capital is characterized by the stock of physical biomass and the value of the benefits that this stock provides. The inclusive wealth accounting literature has disaggregated natural capital valuation based on whether the stock is renewable and whether the flow of benefits is traded in the market. Our analysis focuses on natural capital associated with the market and non-market benefits of forests and terrestrial protected areas, as those are the estimates available in the World Bank (WB) inclusive wealth accounts, and dynamic global vegetation models can capture the changes of the underlying ecosystems.

The WB uses diverse economic concepts and frameworks to estimate natural capital. On the one hand, natural capital associated with market-based benefits from forests (mN) is estimated using the present value of future timber revenues. On the other hand, natural capital embedded in forests and protected areas associated with non-market ecosystem benefits (nN) is estimated using the ecosystem services framework and the option value concept in economics (i.e. the foregone value of alternative economic activities). Specifically, WB conducts a meta-analysis of recreation services, water quality, water

quantity, and non-timber forest products and uses the results to estimate these services in each country. Further, the value from protected areas is estimated by the unrealized revenue had the areas been converted to agricultural fields, giving a lower bound estimate of the protected areas' value^{10,18,19}. Hereafter, the terms market and non-market natural capital will refer only to the benefits described above, as it is the available data for now. However, the methodology remains valid for future additions to natural capital estimates, and the results of this work should be regarded as likely lower-bound estimates.

The first type of natural capital described above (market-based natural capital) is used as a productive asset in the economic production function, a Cobb Douglas production function that estimates GDP based on manufactured capital, human capital, and market natural capital. This relationship arises directly from our definition of market natural capital: a natural stock from which flow of market goods arise. Gross domestic product is a widely implemented standardized way to measure market goods and services and therefore the relationship between market natural capital and GDP is straightforward.

Further, the production of non-market benefits from ecosystems are measured with two methods. First, the World Bank (WB) performs a meta-analysis on individual studies for recreation, water quantity, water quality, and non-wood forest products to estimate the annual flow of benefits from ecosystems to people. Second, the WB uses the option value methodology (i.e. the foregone value of alternative economic activities) to estimate a lower bound value to the natural protected areas in each country. While these estimates are usually given in dollars per hectare, it does not mean that they are measuring direct market transactions, most of the ecosystem benefits valuation studies use non-market valuation methods to estimate the value of ecosystems per hectare. Also, comparing these benefits with the GDP of a country is a way to measure the relative importance of the flow of benefits to people, but importantly, it does not mean that 1 dollar value of ecosystem benefits is fully substitutable with 1 dollar of GDP. These are fundamentally different metrics and that's why we think it is important to have different approaches and our work that make this distinction clear.

The authors establish a linear relationship between GDP and ecosystem services. However, such relationship it is not real. GDP has doubled between 1990 and 2015 in constant dollar terms (World Bank, 2019) but not the ecosystem services which have been decreasing (beyond the nonmarket services) i.e., augmenting deforestation, pollution, etc.. The use of GDP has been highly criticized mainly for not expressing the social and well fare and so far, to show ecosystem wellbeing or ecosystem services. As Ouyan et al., (2020) pointed out GDP summarizes a vast amount of economic information in a single monetary metric but it fails to capture fully the contributions of nature to economic activity and human well-being. Other indicators have been proposed as the gross ecosystem product (GEP) or the gross economic-ecological product (GEPP) (Wang et al., 2021) which have tried to overcome GDP's issues as a social or environmental indicator. I suggest to critically analyze the GDP and instead us another economic indicator that has more relationship with ecosystems that use market prices and surrogates for market prices to calculate the accounting value of ecosystem services (Ouyang et al., 2020).

For instance, GEEP aggregates two accounting metrics (GGDP and GEP). GGDP is the adjusted GDP by removing the cost of environmental degradation caused by pollutants discharged into the environment through various pathways (air, water, land) and the loss of ecosystem services caused by the overuse of ecosystems. GEP includes ecosystem regulating value, ecosystem provisioning value, and ecosystem cultural value (UN, EC, OCDE, & World Bank Group, see (Wang et al., 2021)). Since Ecosystem Provisioning Value and Ecosystem Cultural Value are already accounted in GDP), GEEP is thus the sum of GGDP and Ecosystem Regulating Value.

We agree with the Reviewer that GDP is an incomplete measure of human welfare. It is exactly for that reason that we analyze the non-market components of human well-being. We have now clarified that the approaches are separately calculated in the introductory paragraphs of our revised manuscript

Under the welfare economics framework, human well-being is divided into three components: goods and services people can buy in a market economy (hereafter known as market goods and services), benefits we get from nature that are not usually exchanged in markets (hereafter known as non-market ecosystem benefits), and non-use values from ecosystems attached only to its existence (Fig. 1). Following initial work in Bastien-Olvera and Moore⁷, in this study we estimate the impact of climate-driven biome shifts in market goods using an economic growth model that builds on the standard two-factor neoclassical growth framework¹⁶ used in the DICE integrated assessment model⁴, expanding the economic production function to include natural capital as provisioning raw materials and as a global environmental public good in which the economy is embedded, as proposed in the Dasgupta Review¹⁴ (Fig. 1). Impacts on ecosystem benefits are modeled using a production function that relies more heavily on natural capital but also requires interaction with human and manufactured capital for the benefits to arise. Other benefits from ecosystems, such as non-use or non-anthropocentric values, are not captured in current country-level estimates of natural capital and, therefore, not included in this analysis.

We emphasize the diagram shown in Figure 1 that demonstrates our conceptual framework of human wellbeing that includes not only GDP, but also non-market benefits and non-use values.

Fig 1. Country-level natural capital by type and geographic region. Natural capital is an input for market and non-market components of human well-being. Market natural capital interacts with manufactured and human capital to produce market goods and services which are bounded by the biosphere, while nonmarket natural capital generates non-market ecosystem services. Points show country-level estimates of market and non-market natural capital as a fraction of total inclusive wealth using national accounts data from the World Bank¹⁷, 5 regions are: OECD (the OECD 90 countries and the European Union member states and candidates), LatAm (Latin America and the Caribbean) REF (the reforming economies of Eastern Europe and the Former Soviet Union), Asia (Asian countries except the Middle East, Japan and the Former Soviet Union states), MAF (the Middle East and Africa; additional details in Methods).

We thank the Reviewer for the valuable suggestions regarding the use of GEP, GGDP and GEEP in our study. While we will keep in mind these metrics for future analysis, we do not agree that these are a suitable way to analyze the particular problem of this research. For instance, the proposed metrics (GEP, GGDP, GEEP) value damages on ecosystem health and the provisioning of benefits using similar valuation approaches to the ecosystem services studies used by the World Bank to estimate natural capital and those captured in the VEGS database that we use to apportion benefits to different biomes. However, metrics as GEP, GGDP and GEEP directly subtract the cost of environmental degradation to GDP. In our approach, we choose to keep these two metrics separate. On the one hand, we estimate climate-driven damages on the market economy, and on the other hand, we estimate damages on non-market benefits. Figure 4 in the main manuscript shows those changes in the two independent xy axes, which allows us to avoid making assumptions about how one dollar of GDP substitutes for one dollar of ecosystem benefits.

C. Data & methodology: validity of approach, quality of data, quality of presentation
I notice two main points.

a) The approach is not well supported. The base of the paper is to believe that there is a strong relationship between ecosystem services (mainly market valued) with GDP and welfare (Summers et al., 2012).

We must clarify that our approach does not assume that this relationship between ecosystem services and GDP. Instead, we use inclusive welfare literature to model the production of market economic output and non-market ecosystem benefits separately.

In the inclusive wealth literature, human well-being is understood as arising from a stock of valuable assets (or types of wealth) that produce a flow of benefits to people. This literature has focused on measuring increasingly comprehensive estimates of wealth and conducting backward-looking assessments of whether this inclusive wealth – defined as human capital, natural capital, and the more standard manufactured capital – has declined over time or not, as this is one of the criteria for inter-temporal sustainability¹⁰⁻¹⁴.

And we further elaborate:

Under the welfare economics framework, human well-being is divided into three components: goods and services people can buy in a market economy (hereafter known as market goods and services), benefits we get from nature that are not usually exchanged in markets (hereafter known as non-market ecosystem benefits), and non-use values from ecosystems attached only to its existence (Fig. 1).

b) There is a lack of critical argumentation about their approach also in considering the variety of ecosystem services and ecosystems, because they use mean values that do not reflect the diversity intra ecosystems and inter ecosystem (Carrasco et al., 2014).

We concur with the Reviewer regarding a lack of clarity in the original manuscript. The question of the variety of ecosystem services analyzed in these study is now clarified in one of the first paragraphs of the main text:

Specifically, WB conducts a meta-analysis of recreation services, water quality, water quantity, and non-timber forest products and uses the results to estimate these services in each country. Further, the value from protected areas is estimated by the unrealized revenue had the areas been converted to agricultural fields, giving a lower bound estimate of the protected areas' value^{10,18,19}. Hereafter, the terms market and non-market natural capital will refer only to the benefits described above, as it is the available data for now. However, the methodology remains valid for future additions to natural capital estimates, and the results of this work should be regarded as likely lower-bound estimates.

Regarding the variety of ecosystems, we now use three Dynamic Global Vegetation Models and their associated typologies of ecosystems. This is better explained in the revised version of the Supplementary Materials.

DGVMS model output

We retrieve output data from LPJ-GUESS¹, ORCHIDEE-DGVM², and CARAIB³ models the three dynamic global vegetation models (DGVMs) that participated in the Inter-Sectoral Impact Model Intercomparison Project (ISIMIP) 2b Protocol⁴ under the simulation exercise 2005soc. These simulations consist of fixing land-use to 2005 conditions and simulating the biomes' response to variables such as precipitation, daily maximum and minimum temperature, short wave downwelling radiation, surface air pressure, near-surface relative humidity, near-surface wind speed and carbon dioxide concentration, allowing us to disentangle the climate-driven effects on natural vegetation from the direct anthropogenic disturbances. Each DGVM has a biome typology as shown in Table S1. This table also shows the Earth system models (ESMs) and warming scenarios used to run each of the available simulations.

Supplementary Table 1. Dynamic Global Vegetation Models output.

DGVM	Biomes typology	GCMs	Warming Scenarios	number of runs
LPJ-GUESS (biomes=11)	Boreal needleleaved evergreen (BNE); Boreal shade intolerant needleleaved evergreen (BINE); Boreal needleleaved summergreen (BNS); Temperate broadleaved summergreen (TeBS); shade intolerant broadleaved summergreen (IBS); Temperate broadleaved evergreen (TeBE); Tropical broadleaved evergreen (TrBE); Tropical shade intolerant broadleaved evergreen (TrIBE); Tropical broadleaved raingreen (TrBR); C3 grass (C3G); C4 grass (C4G).	HadGEM2-ES ⁵ , GFDL-ESM2M ⁶ , IPSL-CM5-LR ⁷ , MIROC5 ⁸	RCP2.6, RCP6.0, RCP8.5.	16
ORCHIDEE (biomes=10)	Tropical broadleaved evergreen (trbrev); tropical broadleaved raingreen (trbrgg); temperate needleleaf evergreen (tendev); temperate broadleaved evergreen (tebrev); temperate broadleaved summergreen (tebrsu); boreal needleleaf evergreen (bondev); boreal broadleaved summergreen (bobrsu); boreal needleleaf summergreen (bondsu); C3 natural grass (c3gra); C4 natural grass (c4gra).	GFDL-ESM2M, IPSL-CM5-LR	RCP2.6, RCP6.0	4
CARAIB (biomes=26)	C3 herbs (humid) (c3hh); C3 herbs (dry) (c3dh); C4 herbs (c4h); Broadleaved summergreen arctic shrubs (brsuas); Broadleaved summergreen boreal or temperate cold shrubs (brsutecds); Broadleaved summergreen temperate warm shrubs (brsutewms); Broadleaved evergreen boreal or temperate cold shrubs	HadGEM2-ES, GFDL-ESM2M, IPSL-CM5-LR, MIROC5	RCP2.6, RCP6.0	8

	(brevtecds); Broadleaved evergreen temperate warm shrubs (brevewms); Broadleaved evergreen xeric shrubs (brevxs); Subdesertic shrubs (sds); Tropical shrubs (trs); Needleleaved evergreen boreal or temperate cold trees (ndevtecdt); Needleleaved evergreen temperate cool trees (ndevtect); Needleleaved evergreen trees, drought-tolerant (ndevtedtt); Needleleaved evergreen trees, drought-tolerant, thermophilous (ndevtedttht); Needleleaved evergreen subtropical trees, drought-intolerant (ndevstdit); Needleleaved summergreen boreal or temperate cold trees (ndsutecdt); Needleleaved summergreen subtropical swamp trees (ndsustswt); Broadleaved evergreen trees, drought tolerant (brevdtt); Broadleaved evergreen trees, drought-tolerant, thermophilous (brevdtttht); Broadleaved evergreen subtropical trees, drought-intolerant (brevstdit); Broadleaved summergreen boreal or temperate cold trees (brsutecdt); Broadleaved summergreen temperate cool trees (brsuteclt); Broadleaved summergreen temperate warm trees (brsutewmt); Broadleaved raingreen tropical trees (brrgtrt); Broadleaved evergreen tropical trees (brevtrt).			
--	--	--	--	--

From each simulation and DGVM we retrieved the annual percentage biome-covered area and carbon vegetation content at each 0.5°x0.5° pixel.

c) The explanation about what ecosystem services they are assessing is complete dark, the refer to the World Bank report (2021) but it is a 504-page document, and it is not clear neither what commodity are quantifying and there is not transparency (in the report) about the links between valuation of ecosystem services and the biomes. In summary, there is not transparency in the World Bank report in the method to calculate the mean values per biome per country.

We would like to clarify an important point here. The World Bank report does not relate the different natural capital estimates to the different types of biomes shown in our study. As mentioned in the manuscript, that was one of our jobs in this research:

A challenge in estimating the costs of climate-induced vegetation shifts is the need to apportion natural capital stocks within each country to specific vegetation types. To do this, we use the Values of Ecosystem Goods and Services (VEGS) database, a dataset of 4300 estimates of ecosystem service value per hectare from 818 studies across 123 countries. We apportion the natural capital values in 2018 calculated by the World Bank for each country c ($nN_{2018,c}$ and $mN_{2018,c}$) between biomes based on the area covered by each biome type within the country territory (see Methods).

We have further clarified the methodology section in the Supplementary Material to clarify this procedure, specifically in the Random Forest section:

Random Forest

We use the database subsets $V_{m,b}$ and $V_{n,b}$ to train random forests ($RF_{m,b}$ and $RF_{n,b}$) for each country. A random forest consists of a collection of decision trees that predict a dependent variable using a series of optimal subdivisions in the data. We build our random forests to predict the log value of the ecosystem benefits in $V_{m,b}$ and $V_{n,b}$ based on the variables listed in Table S2, such that

$$RF_{i,b} = f(\text{cover}_b, \text{GDPpc}, \text{cveg}, \text{PercCovered}) \quad \text{eq. 6}$$

We create random forests with 300 decision trees as the decrease in the root mean squared error by each additional decision tree in the random forest reaches saturation at that point. We create one random forest for each combination of 177 countries, 2 types of benefits (market and non-market), and 3 typologies of biomes (from the 3 DGVMs), giving 1067 random forests.

We use the random forest to predict a baseline ecosystem benefit value. Then, by increasing each biome cover by 10% one by one, we obtain the marginal effect relative to the baseline ecosystem benefit value. We can therefore compare the relative sizes of marginal effects. To illustrate the procedure without loss of generality, we imagine a hypothetical country with only two biomes: B1 and B2. Using the relative sizes of the marginal effects given by the random forest $RF_{n,b}$, we can write $es_{2018,c,B2}$ in terms of $es_{2018,c,B1}$

$$es_{2018,c,B2} = \chi_{n,c,B1} es_{2018,c,B1} \quad \text{eq. 7}$$

Rewriting equation 3 for the hypothetical country in 2018,

$$nN_{2018,c} = \frac{a_{2018,c,B1} es_{2018,c,B1} + a_{2018,c,B2} es_{2018,c,B2}}{r} \quad \text{eq. 8}$$

$$\Rightarrow r * nN_{2018,c} = a_{2018,c,B1} es_{2018,c,B1} + a_{2018,c,B2} es_{2018,c,B2} \quad \text{eq. 9}$$

Substituting $es_{2018,c,B2}$ from equation 7

$$r * nN_{2018,c} = a_{2018,c,B1} es_{2018,c,B1} + a_{2018,c,B2} \chi_{n,c,B1} es_{2018,c,B1} \quad \text{eq. 10}$$

$$\Rightarrow es_{2018,c,B1} = \frac{r^*nN_{2018,c}}{a_{2018,c,B1} + a_{2018,c,B2}x_{n,cB1}} \quad \text{eq. 11}$$

All the variables in the right-hand side of equation 11 are known so we can estimate $es_{2018,c,B1}$. Similarly, we estimate $R_{2018,c,b}$. In the following section, we discuss mechanisms that can change the per-area benefits $es_{2018,c,b}$ and $R_{2018,c,b}$ when using them to calculate natural capital in future time steps.

D. Appropriate use of statistics and treatment of uncertainties

There is a lack of attention to show the uncertainties among the ecosystem services by biomes. It seems that authors are only using means, but they are not showing the uncertainties. A global study cannot oversimplify the impacts of climate change in ecosystem services by biome and country just as a single value. I would encourage the authors to go through the variation and uncertainty of their estimates.

We disagree with the Reviewer on this point. Across the manuscript we are careful to show the uncertainties of each step of the methodology and carry it across until the final results (See Figure 4 in the original manuscript). We continuously show the uncertainty by doing whisker plots (Figure 3 in the original manuscript and the revised version), plotting error bars (e.g. Figure 4 in the original manuscript), reporting standard deviation of estimates (Supplementary Table 3 in the original manuscript), plotting fitted lines with 95% confidence intervals (Supplementary Figure 2 and Supplementary Figure 3 in the original manuscript), and reporting standard errors and p-values of our coefficients (Supplementary Table 1 of the original Manuscript).

In the revised version of the manuscript we have kept previous uncertainty metrics and included a graphical summary of the p-values of the damage function coefficients as suggested by Reviewer 1.

E. Conclusions: robustness, validity, reliability

From my point of view, the main problem of the paper is the acritical oversimplification of the relationships among climate change, biomes, GDP, welfare, and inequality.

The impacts of climate change on ecosystems have widely studied for decades at different temporal and spatial scales (Batllori et al., 2013; Gonzalez et al., 2010; Hoffmann et al., 2019; Lloret and Batllori, 2021; Soteriades et al., 2017; Weiskopf et al., 2020). Consequently, in this context, I think that the paper is not original. However, the authors tried to link this biome's shift due to climate change with welfare, through the concept of ecosystem services and use the GDP as an indicator between them. I understand the purpose of that, but I think that simplifying the complexity of biodiversity loss or in this case ecosystem services loss due to climate change, and inequality among countries do not help to improve our understanding of the challenges we face, and that will be exacerbated by climate change.

We disagree with the Reviewer's point of view. While we do acknowledge that there is a vast body of literature analyzing climate change impacts on ecosystem benefits from different angles, we would like to emphasize the importance of having models that integrate these vast bodies of literature into a self-consistent and self-contained analysis. These models allow us to approach overarching questions that transcend the particular contexts of the majority of the research studies. In doing so, some simplifications are necessary as in any other scientific work that aims to answer a particular question. However, our work is a major improvement over previous studies that have tried to estimate the climate-driven ecological changes in human wellbeing. In fact, this is the first global study to approach this question under the imminent biome range shifts that we expect to keep experiencing across the globe.

To further clarify, as explained above, we did not link the welfare and ecosystem services using GDP as an indicator between them. In addition, we believe it is a mistake to state that there is no useful contribution from a study that clearly shows why biome range shifts are an inequality-intensification mechanism across countries. This newly quantified information can guide countries on land-management decisions as adaptation mechanisms to face climate change. It can inform new estimates of policy-relevant metrics such as the social cost of carbon, or it can be evidence in international forums that discuss climate finance mechanisms implementation.

F. Suggested improvements: experiments, data for possible revision

I would suggest:

a) Use instead of GDP the ecosystem product (GEP) or the gross economic-ecological product (GEEP) (Wang et al., 2021) which have tried to overcome GDP's issues as a social or environmental indicator.

We thank the Reviewer for this suggestion. However, as explained above, we believe this is not a suitable approach for our study.

b) Critically analyze the GDP in their narrative and this includes to include more disciplines in their sentence where the authors say, "The human wellbeing effects of ecosystem damages due to climate change is still highly uncertain and depends on integrating insights from climate science, ecology, and environmental economics." And I wonder what about social sciences, anthropology, I think that main driver of human welfare and ecosystem welfare nowadays is related intrinsically to social actions and human decisions. Consequently, we cannot say that only climate science, ecology and environmental economics need to be integrated to analyse climate change challenges and ecosystem damages.

We concur with the Reviewer that many other disciplines play an important role in analyzing the climate change effects on human wellbeing. We have updated the main manuscript:

The human wellbeing effects of ecosystem damages due to climate change is still highly uncertain and depends on integrating insights from climate science, ecology, economics, and many other social sciences.

c) Explain deeper what kind of ecosystem services (market and non-market) are using. It is not clear at all and the World Bank report (2021) that authors use as base it is not clear neither.

The Reviewer's point is well taken. We have now dedicated a whole paragraph to clearly explain what market and non-market estimates of natural capital we use as a baseline for our study.

Natural capital is characterized by the stock of physical biomass and the value of the benefits that this stock provides. The inclusive wealth accounting literature has disaggregated natural capital valuation based on whether the stock is renewable and whether the flow of benefits is traded in the market. Our analysis focuses on natural capital associated with the market and non-market benefits of forests and terrestrial protected areas, as those are the estimates available in the World Bank (WB) inclusive wealth accounts, and dynamic global vegetation models can capture the changes of the underlying ecosystems.

The WB uses diverse economic concepts and frameworks to estimate natural capital. On the one hand, natural capital associated with market-based benefits from forests (mN) is estimated using the present value of future timber revenues. On the other hand, natural capital embedded in forests and protected areas associated with non-market ecosystem benefits (nN) is estimated using the ecosystem services framework and the option value concept in economics (i.e. the foregone value of alternative economic activities). Specifically, WB conducts a meta-analysis of recreation services, water quality, water quantity, and non-timber forest products and uses the results to estimate these services in each country. Further, the value from protected areas is estimated by the unrealized revenue had the areas been converted to agricultural fields, giving a lower bound estimate of the protected areas' value^{10,18,19}. Hereafter, the terms market and non-market natural capital will refer only to the benefits described above, as it is the available data for now. However, the methodology remains valid for future additions to natural capital estimates, and the results of this work should be regarded as likely lower-bound estimates.

d) Try to include the uncertainty among biomes, besides the World Bank methodology I guess there are multiple papers to be used about ecosystem services per biome in different countries. However, the authors try to keep simple (it is good in one way) but they are really oversimplifying to a point where it is very difficult to notice an interesting novelty in the paper.

We thank the Reviewer for the suggestion. Again, we clarify that the estimates of ecosystems benefits per biome are not part of the original World Bank study. Instead, those values are estimated by our methodology and depend not only on the natural capital estimated by the World Bank, but also on the distribution of biomes across the world. In the revised version of the manuscript we now include different biome typologies from the Dynamic Global Vegetation Models (DGVMs) LPJ-GUESS, CARAIB, and ORCHIDEE. This increases the robustness of the results as the random forest apportions different ecosystem

benefit values per hectare to the different biomes. Across the manuscript we keep the including the uncertainty measurements.

G. References: appropriate credit to previous work?

- Batllore, E., Parisien, M.-A., Krawchuk, M.A. and Moritz, M.A., 2013. Climate change-induced shifts in fire for Mediterranean ecosystems. *Global Ecology and Biogeography*, 22(10): 1118-1129.**
- Carrasco, L.R., Nghiem, T.P.L., Sunderland, T. and Koh, L.P., 2014. Economic valuation of ecosystem services fails to capture biodiversity value of tropical forests. *Biological Conservation*, 178: 163-170.**
- Gonzalez, P., Neilson, R.P., Lenihan, J.M. and Drapek, R.J., 2010. Global patterns in the vulnerability of ecosystems to vegetation shifts due to climate change. *Global Ecology and Biogeography*, 19(6): 755-768.**
- Hoffmann, S., Irl, S.D.H. and Beierkuhnlein, C., 2019. Predicted climate shifts within terrestrial protected areas worldwide. *Nature Communications*, 10(1): 4787.**
- Lloret, F. and Batllori, E., 2021. Climate-Induced Global Forest Shifts due to Heatwave-Drought. In: J.G. Canadell and R.B. Jackson (Editors), *Ecosystem Collapse and Climate Change*. Springer International Publishing, Cham, pp. 155-186.**
- Ouyang, Z. et al., 2020. Using gross ecosystem product (GEP) to value nature in decision making. *Proceedings of the National Academy of Sciences*, 117(25): 14593-14601.**
- Pendrill, F. et al., 2022. Disentangling the numbers behind agriculture-driven tropical deforestation. *Science*, 377(6611): eabm9267.**
- Soteriades, A.D., Murray-Rust, D., Trabucco, A. and Metzger, M.J., 2017. Understanding global climate change scenarios through bioclimate stratification. *Environmental Research Letters*, 12(8): 084002.**
- Wang, J. et al., 2021. Gross economic-ecological product as an integrated measure for ecological service and economic products. *Resources, Conservation and Recycling*, 171: 105566.**
- Weiskopf, S.R. et al., 2020. Climate change effects on biodiversity, ecosystems, ecosystem services, and natural resource management in the United States. *Science of The Total Environment*, 733: 137782.**

H. Clarity and context: lucidity of abstract/summary, appropriateness of abstract, introduction and conclusions

The abstract is good. I would recommend being more critical about the approach they use in the introduction and the conclusions are obvious. The problem is to keep doing this kind of research to find what we already know such as developing countries will be more damaged by climate change than developed countries. It is circular the argument when you are using an economic indicator to assess the impacts.

We thank the Reviewer for his or her positive comment on the Abstract. Regarding the following comment on the significance of the study we have extensively made our argument above.

Additional comments

1. Author refers that changes in biome abundance and type will cause changes in the flow of market and non-market benefits that society receives from these ecosystems. However, they use the changes in area and ecosystem services in comparison to the baseline, but they are not, as far as I understood, integrating the shifting ecosystem services (possible future ecosystem services with different biomes) that they can gain with different conditions.

We thank the Reviewer for pointing out this possible source of confusion. When referring to changes in abundance we also include a total disappearance of a particular biome or an appearance of a new biome. We now clarify that in the revised version of the manuscript.

These changes in biome abundance and type (including the complete disappearance or appearance of a biome in a country) will cause changes in the flow of market and non-market benefits that society receives from these ecosystems. The magnitude and direction of those changes depend on the country-specific values attached to each biome.

2. The authors say that limiting the temperature increase within 2°C could cut the damage on GDP and ecosystem services at least by half and reduce economic inequality among countries. There is not linear and causal relationship neither about the limits of warming (2°C) to assure that the inequality will be reduce. Inequality is more complex than 1 or 2 Celsius degrees, it is related to power and domination between groups. I would suggest not trivializing the social and environmental inequality with this kind of sentences.

We disagree with the Reviewer in his or her interpretation of this paragraph. This sentence is in the context of comparing an additional simulation to the initial results.

A further simulation shows that reducing emissions to comply with the recently pledged nationally determined contributions of greenhouse gasses and limiting warming below 2°C²⁸ would cut the impacts of GDP by a third (global mean change by 2100: -0.3%) and the impacts on ecosystem services by half (global mean change by 2100: -9.4%) compared to the SSP2-4.5 that puts the planet at 2.6°C of warming by 2100 with respect to preindustrial levels.

In this sense, the findings should be understood in the context of the scope of our work which is biome range shifts and not power structures. Also, the change are relative to the baseline as clearly pointed out in the paragraph. Still, in the new version of the manuscript we don't do that comparison between scenarios.

3. Figure 4. Graphs 2 and 3 are not clear at all. On the graph on the bottom, it seems that ecosystem services in Europe are in the most affected, but it is not clear when comparing with the 2nd graph.

We thank the Reviewer for pointing out this possible source of confusion. We shared extensively a Figure using the same format with many colleagues and they did not find any problems in reading it and interpreting it. We will make sure to ask the editorial designer of the journal for his or her expert advise.

4. Authors say that changes in non-market ecosystem services are not strongly related to either the country's per-capita, income, or geographic region, implying welfare of both rich and poor nations is equally. Perhaps at country level but that does not make sense. I think that welfare related to ecosystem services are related to the type of biome and they have different resistance, resilience, and adaptive capacity, consequently it is hardly believed that the impacts are equally. Authors are oversimplifying the ecosystems functions and ergo the ecosystem services.

We would like to clarify that not finding a correlation between country's per-capita income and changes in non-market ecosystem services does not mean that changes are equal across countries. The Reviewer states that he or she thinks that welfare from ecosystem services is related with type of biome and we agree with that part of the sentence (see Figure 3A.). Also, the Reviewer says that different biomes have different resistance, resilience and adaptive capacity (to climate change?), which most likely is true but I don't see how that adds to the overall argument of changes in ecosystem benefits correlated with per-capita income. The Reviewer is not making a clear point.

5. In the discussions the authors say, "The human wellbeing effects of ecosystem damages due to climate change is still highly uncertain and depends on integrating insights from climate science, ecology, and environmental economics." And I wonder what about social sciences, anthropology, I think that main driver of human welfare and ecosystem welfare nowadays is related intrinsically to social actions and human decisions. Consequently, we cannot say that only climate science, ecology and environmental economics need to be integrated to analyse climate change challenges and ecosystem damages.

We concur with the Reviewer regarding many other disciplines play an important role in analyzing the climate change effects on human wellbeing. We have updated the main manuscript:

The human wellbeing effects of ecosystem damages due to climate change is still highly uncertain and depends on integrating insights from climate science, ecology, economics, and many other social sciences.

6. Authors recognize some of the limitations of their approach and they mentioned "It is important to note that many climate change impacts on ecosystems are not considered in this analysis. For instance, the vegetation model we use does not account for major ecosystem disturbance events like insect-driven tree mortality or wildfires, which substantially affect vegetation spatial patterns and the carbon cycle". But what about land use/cover change. In many tropics is the main driver of ecosystem services and biodiversity loss.

We thank the Reviewer for the suggestion. However, we note that the Reviewer might have overlooked that we do mention land-use cover change in the main text as a limitation that could be relaxed:

Other simplifications in the simulations could be relaxed by using the scenarios framework developed to assess climate change policies. For example, the SSP-RCP scenario matrix quantifies the levels of policy-led land-use change to comply with emissions, which in some cases (e.g. expansion of biofuel crops in natural areas) might severely decrease biomes coverage.

7. Authors affirm that it is possible to relate the changes in the market benefits of terrestrial ecosystems by looking at the changes in GDP under climate change conditions. I would say that it is not necessary to develop this type of analysis to see that poor countries, mainly in Africa, are struggling with impacts of changing climate like droughts. However, the study does not include droughts only what they call as “temperature change”. They do not clarify if it is mean temperature (very likely), maximum temperature, min temperature, etc.

We would like to point out again that the contribution of our analysis is quantifying the climate damages in human welfare of a specific mechanism, which is biome range shifting. The Reviewer mentions drought as another mechanism, which could be valid but it simply is another research question. We do acknowledge though, that the Reviewer’s point regarding clarification of the climate drivers is well taken. We now mention what are such specific drivers in the main text:

This paper focuses on the effect of biome range shift, the effects associated with total area change, and changes in vegetation carbon content as a proxy for ecosystem overall health. We retrieve ecosystem cover projections under future climate change scenarios using the LPJ-GUESS²⁶, ORCHIDEE-DGVM²⁷, and CARAIB²⁸ models, three process-based vegetation-terrestrial ecosystem models that use outputs from climate models, such as daily minimum and maximum temperatures, total precipitation, short wave downwelling radiation, and humidity to simulate the establishment, competition, and mortality of natural vegetation. These models simulate the dynamics of 11, 10, and 26 plant functional types (PFTs)²⁹ or biomes across the world, respectively.

8. The paper is based on the World Bank assumptions of ecosystem services. This approach ignores the main driver of ecosystem loss in developing countries (agricultural expansion) (Pendrill et al., 2022) because they keep constant the non-forest area. Consequently, that can explain only the impacts of climate change, but they are not doing fully that because they are focusing on the relationship between ecosystem services and GDP.

We do acknowledge that we are solely focused on the effects on climate change because this is the purpose of the study. We are clear about it in several parts of the manuscript. For example, in the main text we mention:

Fig. 2 shows the average present distribution of biomes across latitudes and future changes in its distribution, land cover, and vegetation carbon at two degrees of warming using different representative concentration pathways (RCP2.6, RCP6.0, and RCP8.5) and four Earth system model outputs, fixing land-cover and socio-economic variables at 2005 levels to isolate the sensitivity of terrestrial vegetation to climate change only³⁰.

In the supplement, we specify:

We retrieve output data from LPJ-GUESS¹, ORCHIDEE-DGVM², and CARAIB³ models the three dynamic global vegetation models (DGVMs) that participated in the Inter-Sectoral Impact Model Intercomparison Project (ISIMIP) 2b Protocol⁴ under the simulation exercise 2005soc. These simulations consist of fixing land-use to 2005 conditions and simulating the biomes' response to variables such as precipitation, daily maximum and minimum temperature, short wave downwelling radiation, surface air pressure, near-surface relative humidity, near-surface wind speed and carbon dioxide concentration, allowing us to disentangle the climate-driven effects on natural vegetation from the direct anthropogenic disturbances.

9. Equation 3 and 4 are not supported from an ecological perspective. Ecosystem services value cannot be given in that simplistic form: a) the mean value by ecosystem ignores the variety of ecosystems that countries have because the LPJ model includes only a set of them, b) the agricultural land and agroforestry and agrodiversity that are one of the most important parts of ecosystem services, c) the assumption of the 4th equation that show that all the forest ecosystems can be quantified as possible timber.

We disagree with the Reviewer's points expressed here. Equations 3 and 4 of the original manuscript (now 4 and 5 of the revised version of the manuscript) are not meant to reflect an ecological perspective of ecosystem services values. Instead, those equations reflect literature on natural capital valuation, which is a fundamentally economic approach based on capital theory. Regarding point a, the mean ecosystem benefit value is precisely designed to account for the variety of ecosystems (see index b in the variables of the equation, that stands for type of biome). Naturally, there is not a single vegetation model that captures all of them and that is why we have now included three different vegetation models, including CARAIB, which has 26 natural vegetation biomes.

Regarding point b, in the supplementary material we specify that:

As the role of natural capital becomes increasingly evident, both the World Bank and the UN Environment Program (UNEP), have been developing approaches to incorporate natural capital accounting into national accounts data¹¹⁻¹⁴. We use the World Bank Changing Wealth of Nations 2021 report¹² estimates of natural capital values, excluding natural capital values related to non-renewable resources (e.g. minerals), non-terrestrial ecosystems (e.g. fisheries), and cropland whose relation with climate change has been extensively reviewed in the past¹⁵⁻¹⁷. Instead we focus on natural capital provided by natural terrestrial ecosystems – primarily forests and grasslands.

Regarding point c, we use the VEGS database and a random forest to estimate what fraction of the benefits from ecosystems corresponds to which type of biome and we do not prescribe it in the equations, as some cultures and countries might get revenues from plants that western views might see as grasses without economic value.

10. I would suggest that authors explain more carefully what ecosystem services they consider as market and non-market because only point out that market products (timber), and non-market (recreation, watershed protection, and other non-wood

forest products) it is not enough. Authors refer in the results that stocks of 5 natural, human and manufactured capital and country-specific production elasticities for manufactured and human capital but they did not explain further and it is not clear what really ecosystem services they are estimating. The methodology is not quite clear about the ecosystem services and neither the World Bank report that they use as basis (World Bank, 2021).

The Reviewer's point is well taken. We have further clarified in the main text what are the benefits that the World Bank takes into account when calculating market and non-market natural capital:

The WB uses diverse economic concepts and frameworks to estimate natural capital. On the one hand, natural capital associated with market-based benefits from forests (mN) is estimated using the present value of future timber revenues. On the other hand, natural capital embedded in forests and protected areas associated with non-market ecosystem benefits (nN) is estimated using the ecosystem services framework and the option value concept in economics (i.e. the foregone value of alternative economic activities). Specifically, WB conducts a meta-analysis of recreation services, water quality, water quantity, and non-timber forest products and uses the results to estimate these services in each country. Further, the value from protected areas is estimated by the unrealized revenue had the areas been converted to agricultural fields, giving a lower bound estimate of the protected areas' value^{10,18,19}. Hereafter, the terms market and non-market natural capital will refer only to the benefits described above, as it is the available data for now. However, the methodology remains valid for future additions to natural capital estimates, and the results of this work should be regarded as likely lower-bound estimates.

Reviewer Reports on the First Revision:

Referees' comments:

Referee #1:

Remarks to the Author:

I appreciate very much the efforts that the authors have made to address all the comments I raised. There are only a few issues in the manuscript requiring further improvements if possible.

Major comments#1

Page 4, paragraph 3: It is great that the authors used the results from three different DGVMs in the revised manuscript. The authors mentioned that "Results from other models are presented in the supplementary materials." But I only see Fig S4 and Fig S5 similar to the content of Fig 2 showing for ORCHIDEE and CARAIB.. I do not see other supplementary figures based on ORCHIDEE and CARAIB with the content similar to Fig. 3 and Fig. 4 in the manuscript. I am wondering why?

As stated in the text: "of all the models, only LPJ-GUESS can fully simulate both biome replacement and total biome cover change at the grid cell level over time, making it the preferred choice for the main analysis." I fully agree with the author's choice. But meanwhile, this argument reminds me of another issue: the authors seem to overlook the value of non-vegetated areas (i.e., bareground or desert). When the "total biome cover" changes, the non-vegetated area will also change accordingly. Do the authors assume zero value for non-vegetated areas or not, and why? Should non-vegetated areas also be included in the valuation system presented in the manuscript?

Major comments#2

The use of "market and non-market value" and "GDP and ecosystem services" seem to be mixed in some places, which make it a bit difficult to understand. For instance, in Fig. 4, X-axis uses "non-market natural capital", while Y-axis use "GDP"? Why not use "market natural capital" for Y-axis here, which is more consistent with the figure caption?

I am not sure if there are any considerations where to use "ecosystem services" and where to use "non-market value"?

Minor comments:

Page 2, paragraph 2: please add "terrestrial" before "ecosystems under climate change scenarios on future human well-being."

Page 2, last paragraph: "Impacts on ecosystem benefits are modeled using a production function that relies more heavily on natural capital but also requires interaction with human and manufactured capital for the benefits to arise." Please specify "impacts" of what (biome shift)?

Fig. 2 caption: please add "atmospheric forcings from" before "different Earth system models and representative concentration pathways"

Page 4, paragraph 3: “caomparison” should be “comparison”. Please also add “growth, disturbance” before “competition, and mortality of natural vegetation”.

Figure 3A: There is no legend about the color of the bars and the meaning of black dots. Do different color bars represent market and non-market “benefits per area”?

Figure 3B: what is the difference between the left and right plots? Please specify.

Figure 4: Please add labels (A, B, C) for each subplot and explain the content of each subplot in the figure caption. It is a bit difficult for me to read the bottom and right plots Does the “annual change in GDP ” refer to “market” impact? If so, please clearly state this in the figure caption.

Referee #2:

Remarks to the Author:

Review of “Biome range shifts under climate change: impacts on country-level macro-economic production and ecosystem services”, round 2

Major comments

Overall assessment

The authors have made some useful additions and revisions to the paper to address the reviewers’ concerns. In particular, they have added vegetation carbon content to the set of determinants of the value of ecosystem services (so that these services do not just depend on biome area), they test the robustness/sensitivity of the analysis to different DGVMs, they have improved the estimation of the damage functions, and they have tried to deal with the issues I raised about how the economy develops over the 21st century. The end-product is therefore better, but for me question marks remain.

Use of the RICE model

In response to the main comment of my previous review, the authors now use the RICE Integrated Assessment Model. But I was left unclear as to the precise purpose of this exercise. Is the model just used to make GDP projections so that the value of natural capital losses can be expressed as a proportion of GDP? I suppose not, because for this purpose it would be most straightforward to just use a pre-existing, exogenous GDP scenario like one of the SSPs.

So, I suppose that the model is used not only to generate an exogenous GDP scenario but also to factor in the interaction between natural capital prices on the one hand, and income and the prices of other input factors on the other hand? Maybe this is a dumb question, but the reason I ask is that, on the face of it, the work done on the VEGS database gives a simple means to handle the interaction between natural capital prices and income, at least. That is because income is one of the data points available in the VEGS database and is included in the random forest model (but see below – I am not totally clear on how). Thus, presumably the random forest model can yield an estimate of the income elasticity of natural capital values, which can then be combined with an exogenous GDP forecast to generate future natural capital values that go up as incomes go up?

Admittedly this is all a bit partial equilibrium, so maybe the authors have in mind that the above approach will be misleading and that it is better to do this in general equilibrium? If so, then I have a lot of doubts about the production function(18), its calibration, and whether the results will indeed be better. Eqn. (18) is based on an expansion of the standard Cobb-Douglas production function in the neoclassical growth model to include both market and non-market natural capital. One concern is the restrictive assumptions about capital substitutability imposed by the Cobb-Douglas functional form. But that is not my main concern and these are well known and mentioned in the discussion. My main concern is that some non-market natural capital produces flows of goods and services that do not increase market production, thus surely they cannot be logically included in the production function for market goods and services, which is what (18) is because it has GDP on the LHS. Some provisioning services are probably not fully priced in markets and surely this is even more of a worry for cultural services. So how is all this handled?

Derivation of ecosystem service values from VEGS

I got some useful insights on this from the Supplementary Materials, but still I didn't feel like I fully understood the procedure. Suppose VEGS has incomplete coverage, in the sense that there isn't a datapoint for every biome/ecosystem service/country tuple at time zero. How do you fill in the blanks? Is this the job of the random forest? Or is the random forest only used to make future predictions? Or is VEGS complete in the above sense? And how do you factor in income per capita? How should I understand your method and results compared to more traditional methods of benefits transfer that would use an explicit income elasticity?

Drafting of the main paper

I appreciate the efforts the authors have made to improve the clarity of the main paper, especially dealing with the overlapping frameworks for thinking about natural capital, ecosystem goods/services, etc. But still I feel like the drafting could be further improved. The first section seems to jump between basic conceptual stuff, a description of the method, and a statement of the contribution. It would be best to parse these, and I particularly missed a self-contained, clear statement of the sequence of steps contained within the method. I know that in journals like these there is not supposed to be a long methods section, but surely comprehension of the paper would be aided by a long paragraph or a couple of paragraphs that basically lay out the recipe, step by step.

Minor comments

You want to think of non-market benefits as being distinct from existence value, but to me you have the use versus non-use dichotomy, and then the market versus non-market dichotomy, and non-use values are non-market values. That's what I was brought up on. Maybe things have changed.

You add a sensitivity analysis to the DGVM to the Supplementary Materials, but you don't discuss the results anywhere, do you? This seems like an oversight given this is an important part of your response to Reviewer 1. I think it should be discussed in the main paper.

One of your key statistics is the cumulative discounted cost of natural capital losses, as a percentage of global GDP in 2020. But does it make sense to compare these two quantities? Surely a like for like comparison would be with cumulative discounted global GDP.

I think the idea of fixed land management/use should be mentioned more prominently. I understand why this is useful to focus on the pure climate effect, but some discussion of how changing land management/use could affect the results would be useful.

Referee #3:

Remarks to the Author:

The paper entitled Biome range shifts under climate change: impacts on country-level macro-economic production and ecosystem services analyses the effects of climate change on ecosystem services (market and non-market) at the country level. The results show the disparity of negative impacts in developed and developing countries.

I answered every single point in the first round of reviewing. This second round I will answer specific comments addressed by the authors.

I think that the paper improved satisfactorily. I appreciate the clarification among concepts, mainly the market value and the non-market values embedded in natural ecosystems associated with recreation services, water quantity, water quality, non-timber forest products, and the overall value of protected areas. However, I still have concerns about the World Bank's methodology, but I understand that the authors rely on it. Regarding the protected areas, I do not support the assumption of valuing them as the unrealized revenue that those areas had if they were converted to agricultural fields. I understand the approach; however, I am sure that the authors have added many details to clarify their theoretical framework.

I think it is good that authors agreed that GDP is an incomplete measure of human welfare and adding explanation on this. I hope that in the short-term, besides recognizing the limitations, science uses different approaches and elements in their frameworks. Moreover, it is true that the authors include non-market values but the World Bank methods, from my perspective, are not clear and transparent about how they quantified recreation, water quantity, water quality, and non-wood forest. We know that even at national level those data are well founded and even worse at

subnational level: that is why using global data sets (sometimes uncritically) result in not reliable outputs.

Regarding the use of different models, I think it was an excellent idea to include the diverse DVM (LPJ-GUESS 1, ORCHIDEE-DGVM2, and CARAIB) instead of using one.

The clarification of the methodology to assign different natural capital estimates to the different types of biomes now it is clearer in the Supplementary Material than the previous version.

I appreciate the authors recognize the importance of many disciplines too address environmental challenges because it was narrow their position in the previous version. However, it is evident that one sentence do not amplify the necessity of going beyond this kind of approaches of valuing ecosystems (economically or non-economically).

I still support that the paper is not original and that the impacts of climate change on ecosystems have been widely studied for decades in more interesting approaches (I added the references in the previous round).

I still support that is not original and that the impacts of climate change on ecosystems have widely studied for decades. The difference is that they are trying to link this shift with the human welfare but not in a creative way. The answer of the authors is logical. They defended their approach by supporting their simplification to answer a particular question. I understand them; however, I showed them, in the first round of reviews, a short list of many examples of similar efforts from a more ecological perspective, which in some way, from my perspective, is more useful. Logical results like showing higher impacts of climate change in developing countries do not give novel scientific developments. Besides, showing that impacts will be 232% of the GDP in 2020, makes people wrongly associate the GDP loss as the main element that we have to be worried about, instead of the socio-ecosystem risk (such as loss in ecosystem integrity or ecological functions).

I support we encourage different approaches in science by giving more realistic and plausible outputs that help us to address environmental and political possible solutions.

Author Rebuttals to First Revision:

Referee #1 (Remarks to the Author):

I appreciate very much the efforts that the authors have made to address all the comments I raised. There are only a few issues in the manuscript requiring further improvements if possible.

We appreciate your acknowledgement of the revisions made to our manuscript in response to your helpful comments during the first review. Your feedback has shaped our improvements and we have proceeded to address the few remaining issues.

Major comments#1

Page 4, paragraph 3: It is great that the authors used the results from three different DGVMs in the revised manuscript. The authors mentioned that “Results from other models are presented in the supplementary materials.” But I only see Fig S4 and Fig S5 similar to the content of Fig 2 showing for ORCHIDEE and CARAIB.. I do not see other supplementary figures based on ORCHIDEE and CARAIB with the content similar to Fig. 3 and Fig. 4 in the manuscript. I am wondering why?

As stated in the text: “of all the models, only LPJ-GUESS can fully simulate both biome replacement and total biome cover change at the grid cell level over time, making it the preferred choice for the main analysis.” I fully agree with the author’s choice.

We appreciate the reviewer's agreement with our choice of DGVM for the main analysis. Our decision to present only one set of detailed results in the main text was made to avoid diluting the importance of our preferred DGVM choice. However, we agree that it is valuable to show the sensitivity of our main results to the choice of DGVM.

You are correct in noting that we had not previously included supplementary figures mirroring the exact data in Figures 3 and 4 for the other DGVMs. We agree that the absence of such figures could cause confusion. Hence, in response to your observation, we expanded our supplementary materials to include figures mirroring Figure 3 for the ORCHIDEE-DGVM and CARAIB models, as show below (these figures are visible on pages 15 and 16 of the supplement):

Supplementary Fig 6. Market and non-market benefits by country and biome. A. Distribution of country-level benefits per hectare of biome. Red and blue boxes show market and non-market benefits, respectively. The middle line in the box shows the median, the box covers the first to the third quartile, whiskers show the full range of the data except for some outliers shown in black points. B. Total yearly benefits per geographic region per biome. Note that values are reported as equivalent fractions of GDP as a comparison point only, non-market benefits are not captured in standard GDP accounting practices. C. Changes in the market and non-market natural capital for 1 degree of warming. Using model output from CARAIB⁵.

Supplementary Fig 7. Market and non-market benefits by country and biome. A. Distribution of country-level benefits per hectare of biome. Red and blue boxes show market and non-market benefits, respectively. The middle line in the box shows the median, the box covers the first to the third quartile, whiskers show the full range of the data except for some outliers shown in black points. B. Total yearly benefits per geographic region per biome. Note that values are reported as equivalent fractions of GDP as a comparison point only, non-market benefits are not captured in standard GDP accounting practices. C. Changes in the market and non-market natural capital for 1 degree of warming. Model output from

ORCHIDEE-DGVM⁴.

Regarding Figure 4, we now provide a supplementary figure with country-level trajectories for CARAIB and ORCHIDEE-DGVM as well as labels for the data points in 2100 (visible on page 17 of the Supplement):

Supplementary Figure 8. Trajectories of the annual change in market and non-market benefits are shown for 57 regions in “Green” RICE50+ with respect to simulations without climate change impacts. Labels in 2100 show the code of the regions as given in RICE50+. Error bars show the standard errors of the estimated damage functions under different general circulation model

outputs. Dashed lines show the population-weighted mean values in 2100. Upper panel: Results under LPJ-GUESS simulation output (shown in Figure 4).

We have now rectified this omission by citing the supplemental figure in the caption for Figure 4. The revised caption (visible on page 7 of the manuscript) reads:

Fig. 4. Market and non-market impacts under SSP2-6.0 scenario. A. Trajectories of the annual change in market benefits (shown as annual GDP change) and non-market benefits are shown for the 57 countries and regions in RICE50+ with respect to simulations without climate-change impacts. Error bars show the standard errors of the estimated damage functions under different general circulation model outputs. Dashed lines show the population-weighted mean values in 2100. A comparison of these results for different DGVMs with region labels is presented in Figure S8. **B.** Cumulative global population plotted against respective levels of annual GDP change in 2100. **C.** Cumulative global population plotted against respective levels of annual non-market benefits change in 2100.

Also, when mentioning the results in the main text, we now describe the specific result from other DGVMs, for the non-market benefits section (on page 8) now it reads as follows:

The global population-weighted average change in non-market benefits value in 2100 is -9.2% relative to the baseline (1.4% and -4.9% using damage functions based on CARAIB and ORCHIDEE-DGVM).

For the market benefits section (on page 8) now it reads as follows:

The global population-weighted average change in GDP by 2100 is -1.3% of the baseline GDP (-0.7% and -1.4% using output from CARAIB and ORCHIDEE DGVMs).

Further, the discussion section (on page 9) brings back the important distinction between our preferred DGVM and the others:

[...] it is worth noting that the results based on DGVMs other than LPJ-GUESS, which generally show smaller GDP and non-market benefits disruptions, should be interpreted as conservative estimates, as the output used from these models provides proportional, not absolute, biome shifts, thus representing the effect of biome replacement without considering actual area change as with LPJ-GUESS.

We hope this addition provides clarity, and thank you for bringing this to our attention.

But meanwhile, this argument reminds me of another issue: the authors seem to overlook the value of non-vegetated areas (i.e., bareground or desert). When the “total biome cover” changes, the non-vegetated area will also change accordingly. Do the authors assume zero value for non-vegetated areas or not, and why? Should non-vegetated areas also be included in the valuation system presented in the manuscript?

We appreciate your insightful observation regarding the value of non-vegetated areas. You are correct in noting that the percentage of non-vegetated areas changes over time. In our methodology, we assume that the benefits derived from these areas are zero, since we distribute 100% of the benefits among the plant functional types represented in the models we use.

While we recognize that non-vegetated areas may play an ecological role, our analysis is limited by the specific values considered in the World Bank methodology, which do not pertain to these areas. For instance, the market natural capital primarily comprises timber value, which by definition needs vegetation. On the other hand, the non-market natural capital estimated by the World Bank includes non-timber forest products, recreation, water quality, and water quantity. Although it could be argued that non-vegetated areas could provide some of these benefits, we've chosen to make a pragmatic assumption, given our focus on plant functional types.

We agree that this assumption should be clarified in the manuscript. Therefore, we have added explicit mention of it in the revised version of the Supplementary Material (visible on page 6 of the Supplement), noting the potential for benefits from non-vegetated areas but also the reasons for their exclusion in our study:

Importantly, we assume that the benefits derived from non-vegetated areas are zero in our methodology, as we apportion 100% of the benefits among the plant functional types. While non-vegetated areas could play ecological roles and potentially provide benefits such as recreation, runoff generation, and soil stabilization (desert biological crusts, such as lichens and cyanobacteria), these specific values have not been factored into the World Bank's estimates, which primarily focus on the value of benefits related to forests. We believe this is a pragmatic assumption for our study, but we want to make it explicit here.

We appreciate your observation regarding deserts. Given that the dominant plant type in deserts, C4 grasses, is included as a plant functional type (PFT) in the models we use, we argue that deserts are implicitly included in our analyses. Nonetheless, we recognize the importance of clarifying this aspect for our readers. In response to your comment, we've expanded our definitions of PFTs and biomes in the supplementary materials and further elucidated how they are applied in our study (visible on page 2 and 3 of the Supplement):

DGVMs classify vegetation into various Plant Functional Types (PFTs). These PFTs group together plant species sharing similar characteristics, like comparable responses to environmental conditions, similar physiological traits, and shared roles in ecosystem function⁷. In contrast, biomes are large geographic areas characterized by specific types of dominant vegetation and climate conditions. In many instances, the association between a biome and PFT category is self-evident; for instance, the 'temperate needleleaved evergreen' PFT typically dominates the 'temperate evergreen forest' biome.

However, certain biomes, such as deserts, do not straightforwardly correspond to a PFT category. In our analysis, though, these are represented indirectly, as they predominantly consist of C4 grasses, which fall within the PFT classifications of the DGVMs we use. We acknowledge that the terms PFT and biome are not perfect substitutes, but for the purpose of this study, we use them interchangeably.

We appreciate your attention to this detail and hope that the further details of our analytic approach and the clarification of assumption sufficiently addresses your concern.

Major comments#2

The use of “market and non-market value” and “GDP and ecosystem services” seem to be mixed in some places, which make it a bit difficult to understand. For instance, in Fig. 4, X-axis uses “non-market natural capital” , while Y-axis use “GDP”? Why not use “market natural capital” for Y-axis here, which is more consistent with the figure caption?

I am not sure if there are any considerations where to use “ecosystem services” and where to use “non-market value”?

We appreciate your observation regarding the interchangeable use of "market and non-market value" and "GDP and ecosystem services" in our manuscript. We understand that this could potentially cause confusion and thank you for pointing it out.

In this revised version of the manuscript we are now using clearer definitions of the terms and are consistent with their use throughout the text. For instance, in the second paragraph (on page 2), we have settled on the terms “market benefits” and “non-market (ecosystem) benefits”:

Human well-being can be divided into three components: goods and services exchanged in markets (hereafter referred to as market benefits); use benefits from nature that are not usually exchanged in markets (hereafter referred to as non-market benefits); and non-use values from biodiversity and

ecosystems attached only to their existence (Fig. 1). In this study, we focus on the first two components.

In the next paragraph, we introduce the role of GDP in relation to market benefits:

to estimate the effect of changing natural capital on the market benefits (as measured by gross domestic product; GDP) and non-market benefits in a fully consistent framework

And finally, in the following paragraph, we introduce the relation between different types of natural capital (market and non-market) and different market and non-market flows of benefits:

The flow of benefits from a country's market natural capital stock, the market environmental benefits, are included in GDP, while the non-market benefits that flow from the non-market natural capital stock are not.

We use these terms with specific intent: 'Market and non-market natural capital' each refer to a stock of wealth, while 'market and non-market benefits' refer to the flow of benefits derived from each of these stocks, respectively. We have modified the shaded boxes in Figure 1 to reflect this language:

Fig 1. Country-level natural capital by type and geographic region. Natural capital contributes to market and non-market components of human well-being. Market natural capital interacts with

manufactured and human capital to produce market benefits that are modulated by the regulating and maintenance services from non-market natural capital, which also generates non-market benefits (e.g. cultural services). Points show country-level estimates of market and non-market natural capital as a fraction of total wealth (defined as the sum of natural capital, manufactured capital, and human capital) using national accounts data from the World Bank¹⁹, the 5 regions are: OECD (the OECD 90 countries and the European Union member states and candidates), LAM (Latin America and the Caribbean) REF (the reforming economies of Eastern Europe and the Former Soviet Union), ASIA (Asian countries except the Middle East, Japan, and the Former Soviet Union states), MAF (the Middle East and Africa). Due to limitations in the World Bank data, other benefits from ecosystems, such as non-use values and non-anthropocentric values, are not included in our measure of human well-being.

Ultimately, the primary variables of interest in our findings are the changes in the flow of benefits derived from the stock of natural capital. Consequently, the appropriate axis labels in Figure 4 are "Annual Changes in Market Benefits" (which we refer to as "Annual GDP change" for simplicity and straightforward interpretation) and "Annual non-market benefits change". These terms encapsulate the fluctuations in both market and non-market benefits that people derive from natural capital.

Fig. 4. Market and non-market impacts under SSP2-6.0 scenario. **A.** Trajectories of the annual change in market benefits (shown as annual GDP change) and non-market benefits are shown for the 57 countries and regions in RICE50+ with respect to simulations without climate-change impacts. Error bars show the standard errors of the estimated damage functions under different general circulation model outputs. Dashed lines show the population-weighted mean values in 2100. A comparison of these results for different DGVMs with region labels is presented in Figure S8. **B.** Cumulative global population plotted against respective levels of annual GDP change in 2100. **C.** Cumulative global population plotted against respective levels of annual non-market benefits change in 2100.

We appreciate your meticulous review as it certainly increases the clarity of our paper.

Minor comments:

Page 2, paragraph 2: please add “terrestrial” before “ecosystems under climate change scenarios on future human well-being.”

We appreciate your attention to the specific scope of our study. Although we have altered the original sentence following suggestions from Reviewer 2, we agree that it is crucial to specify our focus on "terrestrial" ecosystems. As such, we have incorporated the term "terrestrial" into the revised sentence in the equivalent section of the text. In the second paragraph on page 2, the text now reads as follows:

“In this study, we expand a regional benefit-cost climate integrated assessment model (IAM) to explore the welfare effects of climate impacts on terrestrial natural capital.”

Page 2, last paragraph: “Impacts on ecosystem benefits are modeled using a production function that relies more heavily on natural capital but also requires interaction with human and manufactured capital for the benefits to arise.” Please specify “impacts” of what (biome shift)?

We appreciate your request for clarification. In response to this and other comments, we have revised the introduction for improved clarity. As a result, the mentioned sentence is no longer included.

Fig. 2 caption: please add “atmospheric forcings from” before “different Earth system models and representative concentration pathways ”

Thank you for the suggestion, we have added the text in the figure caption:

Fig. 2. Biome shifts and changes in area cover and vegetation carbon content. A: average percentage of grid cells covered by different biomes in the present (2016-2020). B: change in coverage under 2 degree warming projections relative to present day (using atmospheric forcings from different Earth system models and representative concentration pathways). C: Changes in the fraction of a grid cell covered by natural vegetation. D: Changes in vegetation carbon content within each grid cell (kg per square meter). Model output from LPJ-GUESS under 3 warming scenarios and 4 climate model outputs (Figures using the two other DGVMs are shown in Fig S4 and Fig S5).

Page 4, paragraph 3: “caomparison” should be “comparison”. Please also add “growth, disturbance” before “competition, and mortality of natural vegetation”.

Thank you for your careful reading. We've corrected the typographical error "caomparison" to "comparison" and added "growth, disturbance" before "competition, and mortality of natural vegetation" as suggested.

Figure 3A: There is no legend about the color of the bars and the meaning of black dots. Do different color bars represent market and non-market “benefits per area”?

Thank you for your valuable suggestion. Indeed, the color bars represent market and non-market "benefits per area," and the black dots indicate outliers. We've updated the caption to reflect this information explicitly for better clarity:

Fig. 3. Market and non-market benefits by country and biome. A. Distribution of country-level benefits per hectare of biome. Red and blue boxes show market and non-market benefits, respectively. The middle line in the box shows the median, the box covers the first to the third quartile, whiskers show the full range of the data except for some outliers shown in black points. B. Total yearly benefits per geographic region per biome. Note that values are reported as equivalent fractions of GDP as a comparison point only, non-market benefits are not captured in standard GDP accounting practices. C. Changes in the market and non-market natural capital for 1 degree of warming (see Table S5 and S6 for coefficients and information on missing countries). Figures using the two other DGVMs are shown in Fig S6 and Fig S7.

Figure 3B: what is the difference between the left and right plots? Please specify.

Thank you for pointing out the lack of clarity. The left and right plots in Figure 3B differentiate between market and non-market benefits, respectively. We have now updated the figure to specify this distinction.

Fig. 3. Market and non-market benefits by country and biome. A. Distribution of country-level benefits per hectare of biome. Red and blue boxes show market and non-market benefits, respectively. The middle line in the box shows the median, the box covers the first to the third quartile, whiskers show the full range of the data except for some outliers shown in black points. B. Total yearly benefits per geographic region per biome. Note that values are reported as equivalent fractions of GDP as a comparison point only, non-market benefits are not captured in standard GDP accounting practices. C. Changes in the market and non-market natural capital for 1 degree of warming (see Table S5 and S6 for coefficients and information on missing countries). Figures using the two other DGVMs are shown in Fig S6 and Fig S7.

Figure 4: Please add labels (A, B, C) for each subplot and explain the content of each subplot in the figure caption. It is a bit difficult for me to read the bottom and right plots Does the "annual change in GDP" refer to "market"

impact? If so, please clearly state this in the figure caption.

Thank you for your feedback on Figure 4. We have now added labels (A, B, C) to each subplot for clarity, and we've expanded the figure caption to more thoroughly explain the content of each subplot. Furthermore, you're correct that "annual change in GDP" is representative of the market impact. To avoid confusion, we have explicitly stated this in the figure caption along with the revisions to the text mentioned in previous responses. These changes should make the figure easier to understand. We are grateful for your thorough and insightful feedback on this manuscript. It has been instrumental in refining our work and enhancing the clarity for our readers.

Fig. 4. Market and non-market impacts under SSP2-6.0 scenario. **A.** Trajectories of the annual change in market benefits (shown as annual GDP change) and non-market benefits are shown for the 57 countries and regions in RICE50+ with respect to simulations without climate-change impacts. Error bars show the standard errors of the estimated damage functions under different general circulation model outputs. Dashed lines show the population-weighted mean values in 2100. A comparison of these results for different DGVMs with region labels is presented in Figure S8. **B.** Cumulative global population plotted against respective levels of annual GDP change in 2100. **C.** Cumulative global population plotted against respective levels of annual non-market benefits change in 2100.

Referee #2 (Remarks to the Author):

Review of “Biome range shifts under climate change: impacts on country-level macro-economic production and ecosystem services”, round 2

Major comments

Overall assessment

The authors have made some useful additions and revisions to the paper to address the reviewers' concerns. In particular, they have added vegetation carbon content to the set of determinants of the value of ecosystem services (so that these services do not just depend on biome area), they test the robustness/sensitivity of the analysis to different DGVMs, they have improved the estimation of the damage functions, and they have tried to deal with the issues I raised about how the economy develops over the 21st century. The end-product is therefore better, but for me question marks remain.

We are grateful for your recognition of the substantial revisions and additions made to our paper in response to your first round of comments. We appreciate your acknowledgement that these changes have improved the overall quality of the work. While we note your persisting reservations, we believe that we have provided comprehensive and satisfactory responses to your remaining concerns in the revised version of the manuscript and supplement, as we detail in the following sections.

Use of the RICE model

In response to the main comment of my previous review, the authors now use the RICE. But I was left unclear as to the precise purpose of this exercise. Is the model just used to make GDP projections so that the value of natural capital losses can be expressed as a proportion of GDP? I suppose not, because for this purpose it would be most straightforward to just use a pre-existing, exogenous GDP scenario like one of the SSPs.

Your understanding is correct; we employ the RICE model for more than just retrieving GDP projections. (In particular, we use the RICE50+ Integrated Assessment Model, a model developed starting from RICE with more detailed regional resolutions and (almost) country-level calibration of mitigation costs and climate impacts). We have modified the model's main economic module and added a module for natural capital, notably to expand the production function (we call this new model “Green” RICE50+). This modification allows us to examine the impacts of changing natural capital on economic production, consumption, and welfare. Moreover, several aggregates including market and non-market ecosystem benefits are computed based on a starting point of historical

statistics and the empirical and model results of this paper. This model includes the different SSPs, but runs them in a fully integrated endogenous optimization procedure and allows varying mitigation action by countries as well as endogenous investment/savings rates. Therefore the model, in particular due to the changes in the economic system and production function, allows for a consistent set of scenarios of all key variables rather than a simulation exercise. For reference, the model is open-source <https://github.com/witch-team/RICE50xmodel> and new modules such as this one will be added and released once finalized and peer-reviewed.

So, I suppose that the model is used not only to generate an exogenous GDP scenario but also to factor in the interaction between natural capital prices on the one hand, and income and the prices of other input factors on the other hand? Maybe this is a dumb question, but the reason I ask is that, on the face of it, the work done on the VEGS database gives a simple means to handle the interaction between natural capital prices and income, at least. That is because income is one of the data points available in the VEGS database and is included in the random forest model (but see below – I am not totally clear on how). Thus, presumably the random forest model can yield an estimate of the income elasticity of natural capital values, which can then be combined with an exogenous GDP forecast to generate future natural capital values that go up as incomes go up?

Admittedly this is all a bit partial equilibrium, so maybe the authors have in mind that the above approach will be misleading and that it is better to do this in general equilibrium?

We are grateful for your insightful question, and we believe that this nuanced discussion will indeed enrich the understanding of our engaged readers. We concur that the VEGS database along with the random forest model and the incorporation of natural capital into the “Green” RICE50+ model serve different yet synergistic roles in heightening the potential impact of our research.

The VEGS database and the random forest model certainly factor in the elasticity of ecosystem benefits to income, allowing the importance of biomes to provide the total value of natural capital benefits to vary with income (more on that below). Indeed, the natural capital trajectories, which are later utilized to fit a damage function, already consider this elasticity along with the impact of climate change on biome distribution and vegetation carbon content. As you correctly noted, this approach is grounded in a partial equilibrium perspective, as it employs exogenous scenarios such as the SSPs.

This is where integrating these synthesized metrics of natural capital into the core equations of the RICE50+ model becomes valuable. The RICE50+ model allows us to consolidate all the information from scenario analysis and the VEGS database into a framework that can accommodate endogenous economic and non-economic growth. Crucially, this amalgamation paves the way for future utilization of “Green” RICE50+ in its full capacity, enabling us to model diverse scenarios, policies, including different levels of cross-country cooperation, both on climate but also ecosystem services and biodiversity conservation, natural capital restoration investment, emissions optimization, and general “beyond GDP” policies among others.

In essence, we do not view the use of the “Green” RICE50+ model as superior to employing outputs from the random forest model fitted to the VEGS database under different warming scenarios. Rather, we believe these two methodologies complement each other, adding layers of complexity and depth to our analysis, in particular also providing the necessary data and open-source model to integrate ecosystems in cost-benefit integrated assessment models.

To facilitate a deeper understanding of the numerous steps included in our methodology, we have added a new section in the supplementary materials discussing this point (on page 1 of the Supplement) and have also added a figure that shows the overall study design:

Study Design

By bringing together complementary methodologies, this study analyzes the effects of changing ecosystems under climate change scenarios on future human well-being. We structure our research methodology into three sequential phases, as illustrated in Figure S1. The first phase involves estimating country-specific values per hectare of the major biomes found in each country. The biomes included in our study are based on a set of plant functional types relating to forest, grassland, and desert ecosystems. In the second phase, we generate future projections of natural capital based on biome projections. Finally, in the third phase, we incorporate the dynamics of natural capital into an integrated assessment model. This section offers a high-level overview of each phase of our analysis to provide a comprehensive understanding of our study design; detailed explanations of each individual step are presented in subsequent sections of this document.

In the first phase (Fig S1A), we extend the Values of Ecosystem Goods and Services (VEGS) database¹ to incorporate estimates of biome cover and vegetation carbon content at all study locations, drawing from three Dynamic Global Vegetation Models (DGVMs). For each country, we select a subset of the expanded VEGS database, based on studies within VEGS that exhibit similar biome distributions to the country's average biome coverage. Next, we apply a random forest methodology to this selected subset to obtain the relative contribution of each biome in providing market and non-market benefits. Finally, we distribute country-level market and non-market natural capital estimates from the World Bank across the different biomes within the country, based on biomes' relative importance estimated via the random forest analysis.

In the second phase (Fig. S1 B), we leverage country-specific values per hectare of each biome to project total market and non-market natural capital values. This projection is based on the future distributions of biomes from three DGVMs under four representative concentration pathways (RCPs) and four general circulation models (GCMs). The biome value per hectare, derived from the initial phase, is adjusted according to the total area of the biome cover and its vegetation carbon content. Lastly, we obtain the 'damage functions' by applying linear regressions to each natural capital trajectory to estimate the mean impact of global temperature rise on each country's natural capital.

In the final phase of our study, we incorporate the 'damage functions' derived from the previous stage and the World Bank's country-level natural capital estimates into the RICE50+ model to capture endogenous economic growth with natural capital (Fig. S1 C). This expanded version of the RICE50+

model², now termed “Green” RICE50+, allows us to examine potential feedback loops and interactions that might not be visible in a partial equilibrium context (such as the second phase of the study design). For instance, the accumulation of manufactured capital through the savings rate now also depends on natural capital, as it underlies economic production. This coupled approach paves the way to test and model various experiments in the future using RICE50+ in its full capacity, enabling us to model diverse scenarios and policies, including different levels of cross-country cooperation on climate, as well as on the issues of ecosystem services and biodiversity conservation, natural capital restoration investment, emissions optimization, and general “beyond GDP” policies, among others. In particular, we provide the necessary data and open-source code to integrate ecosystems in cost-benefit integrated assessment models.

Redacted

For Legends: **See Extended Data Figure 1**

If so, then I have a lot of doubts about the production function(18), its calibration, and whether the results will indeed be better. Eqn. (18) is based on an expansion of the standard Cobb-Douglas production function in the neoclassical growth model to include both market and non-market natural capital. One concern is the restrictive assumptions about capital substitutability imposed by the Cobb-Douglas functional form. But that is not my main concern and these are well known and mentioned in the discussion. My main concern is that some non-market natural capital produces flows of goods and services that do not increase market production, thus surely they cannot be logically included in the production function for market goods and services, which is what (18) is because it has GDP on the LHS. Some provisioning services are probably not fully

priced in markets and surely this is even more of a worry for cultural services. So how is all this handled?

We acknowledge the reviewer's concerns and agree with the assertion that some ecosystem services are not fully priced or not priced at all and thus, might not logically be included in the neoclassical three-factor production function.

The production function that we adopt from the Dasgupta Review slightly diverges from the neoclassical form in the sense that market production is modulated by the maintenance and regulation services provided by the biosphere. However, due to lack of a global measure of such services consistent with the World Bank accounts, we use the global aggregate of non-market natural capital as a proxy for it, focusing on how these non-market ecosystem services can influence market production, rather than seeing them as direct priced contributors to GDP.

The equation in question, equation 18, can be divided into two parts: First, the conventional neoclassical part that depends on labor (L), manufactured capital (K), and market natural capital (mN), with elasticities that add up to 1, assuming constant returns to scale, and are calibrated using country-level data on the share of GDP corresponding to annual revenues from natural capital. For instance, if the annual revenues from market natural capital equate to 1% of the GDP, the elasticity of market natural capital (mN) in the production function is 0.01. We get these estimates from national wealth accounts and most of them are small in magnitude as the by far largest share of economic output has been based on industrial progress and human capital, in particular since the second half of the 20th century.

And the second part of equation 18 is a non-traditional approach usually not found in neoclassical economics, that we adopted from Dasgupta (2021), which adds a factor (S^b) that represents the extent to which market production is possible due to all the life-maintaining mechanisms of the Earth system. This factor, along with the standard total factor productivity (TFP) could be understood as an adjusted total factor productivity A .

$$\begin{aligned} GDP_{t,c} &= S_t^{b_r} * TFP_{t,c} * L_{t,c}^{\gamma_{1c}} * K_{t,c}^{\gamma_{2c}} * mN_{t,c}^{\gamma_{3c}} \\ &= A_{t,c,r} * L_{t,c}^{\gamma_{1c}} * K_{t,c}^{\gamma_{2c}} * mN_{t,c}^{\gamma_{3c}} \end{aligned}$$

In our model, S is represented by the global sum of non-market natural capital due to the lack of a more adequate measure that is consistent with the World Bank wealth accounts. We use these recorded values of S and the other factors to estimate the regression model shown in equation 19:

$$\log(GDP_{t,c}) = R_c * \widehat{b}_r \log(S_t) + \widehat{\gamma}_1 \log(H_{t,c}) + \widehat{\gamma}_2 \log(K_{t,c}) + \widehat{\gamma}_3 \log(mN_{t,c}) + \theta_t + \theta_c + \epsilon_{t,c}$$

eq. 19

Importantly, we do not consider S as capital that directly yields a flow of GDP, as the Reviewer correctly points out that many of the benefits of this natural capital are not fully priced. Instead, it is viewed as a factor that influences the capacity of the other factors to generate GDP. For example, a stable and predictable climate reduces uncertainty over future investments in, say, agriculture, making the latter cheaper and more abundant. This, in turn, results in higher economic output, although only a fraction of the benefits stemming from it are reflected in market prices. The degree to which some non-market natural capital produces flows of goods and services that do or do not increase market production is reflected in the value of b . The higher is b , the more non-market natural capital plays a role in the production of goods and services that are priced and increase economic output. Mathematically, 'b' is the elasticity of GDP growth with respect to non-market natural capital S .

We recognize and appreciate the reviewer's concerns, which have led us to clarify our methodology further. We clarified this point in the revised manuscript, ensuring our approach and the reasoning behind it is clearly articulated.

$$GDP_{t,c} = TFP_{t,c} * S_t^{b_r} * L_{t,c}^{\gamma_{1c}} * K_{t,c}^{\gamma_{2c}} * mN_{t,c}^{\gamma_{3c}}$$

eq. 18

TFP and labor are exogenous and calibrated to match the population and economic output from the Shared Socioeconomic Pathway 2 (SSP2). Importantly, equation 18 is a non-traditional approach usually not found in neoclassical economics, that we adopted from the Dasgupta Review, which adds a factor (S^b) that represents the extent to which market production is possible due to all the life-maintaining mechanisms of the Earth system. However, due to lack of a global measure of such services consistent with the World Bank accounts, we use the global aggregate of non-market natural capital as a proxy for it. This factor, along with the standard total factor productivity (TFP) could be understood as an adjusted total factor productivity.

The degree to which some non-market natural capital produces flows of goods and services that do or do not increase market production is reflected in the value of b , which varies by region. The higher is b , the more non-market natural capital plays a role in the production of goods and services that are priced and increase economic output. Mathematically, b is the elasticity of GDP growth with respect to global non-market natural capital S . For example, a stable and predictable climate reduces uncertainty over future investments in, say, agriculture, making the latter cheaper and more abundant. This, in turn, results in higher economic output, although only a fraction of the benefits stemming from it are reflected in market prices. We acknowledge that this specification is not common ground in neoclassical economics and therefore we show in Figure S10 a version of our results assuming $b = 0$.

In addition, recognizing that readers could be interested in the results of a standard neoclassical production function, we have now included a new supplementary Figure (on page 20) that shows the main results, but assuming $b = 0$. This figure shows that annual GDP changes are on average an order of magnitude lower than the preferred specification. These impacts, as well as the results shown in the main section, are unequally distributed with more burden towards lower-income

countries. In particular, some regions under this specification show small increases in GDP by the end of the century (13 out of 57).

Supplementary Figure 10. Replication of the main results using a standard neoclassical production function, equation 18 assuming $b = 0$. Upper panel: Trajectories of the annual change in market and non-market benefits are shown for 57 regions in "Green" RICE50+ using $b = 0$ in the production function. Error bars show the standard errors of the estimated damage functions under different general circulation model outputs. Dashed lines show the population-weighted mean values in 2100. Population-weighted change GDP change in 2100: -0.17, an order of magnitude lower than the

preferred estimates for *b*. Lower panel: distribution of impacts in 2100, the pattern of the distributional burden of impacts remains.

Derivation of ecosystem service values from VEGS

I got some useful insights on this from the Supplementary Materials, but still I didn't feel like I fully understood the procedure. Suppose VEGS has incomplete coverage, in the sense that there isn't a datapoint for every biome/ecosystem service/country tuple at time zero. How do you fill in the blanks? [...] Or is VEGS complete in the above sense?

We appreciate the reviewer's inquiry and recognize the need for further clarification regarding the use of the VEGS database and the handling of missing data. First, we want to clarify that we are not using the VEGS database to estimate the value of market and non-market benefits provided by different biomes in countries around the world. We have estimates of those values from the World Bank, though we acknowledge the limitations of the dataset. We are using the VEGS database and our random-forest model to allocate the country-level values from the World Bank across the biomes defined by our included plant-functional types in the various DGVMs.

Having made that point, we address missing data across three categories: biomes, ecosystem services, and countries.

1. **Biomes:** The dynamic global vegetation models we use don't depict biomes as discrete units with clear boundaries. Instead, multiple biomes coexist to varying degrees within a single pixel. This characteristic, combined with the geographic breadth of the VEGS database, ensures we have at least one observation where pixel cover exceeds zero for all biomes, all DGVMs and all types of market and non-market benefits (see figure below).
2. **Ecosystem services:** Our study diverges from the classic ecosystem services typology (which separates into provisioning, regulating, cultural, and supporting) and instead re-categorizes the VEGS database into market and non-market values. As mentioned in the supplementary materials, market values encompass provisioning services, while non-market values comprise regulating, cultural, and supporting services. Because we regroup four categories into two, we have much lower probability of missing data. The figure below shows the density distribution of observations divided by biomes across the different percentage cover for market and non-market values. This figure demonstrates the comprehensive representation of all biome/value combinations in the VEGS database.

3. Countries: Although the VEGS database doesn't contain observations for every country, the random forest model doesn't consider 'country' as an independent variable used to calculate the relative value contribution of each biome. Instead, we control for this by using income levels and matching observations that closely resemble a given country's vegetation distribution. The resemblance metric is given by the euclidean distance of the biome distribution in the target country to the biome distribution of the observations in VEGS. In that sense, it is not the job of the random forest to account for missing countries in VEGS.

Is this the job of the random forest? Or is the random forest only used to make future predictions?

The random forest is used to estimate the relative importance of each biome with regard to the total value of the natural capital stocks within a country. Because the World Bank data is provided at the country level, we need a credible method to attribute these values to the various biomes within a country. We do so using the VEGS database and our random forest model. Regarding the random forest methodology: we primarily employ it to deduce a correlation between the current market and non-market values and the existing vegetation distributions. By training and validating a subset of the VEGS database, which closely mirrors the biome distribution of a target country, we are able to derive a biome-specific, per-hectare value for that country.

We do this by predicting the value per hectare under the average biome distribution of the input subset of VEGS and then sequentially increase each biome present in the target country by 10% and evaluate the marginal value change per hectare prompted by this increase. The relative differences in these outcomes help us establish country-specific per-hectare values. We do that by solving a simple algebraic equation (shown below) that distributes the natural capital values given by the World Bank across the different biomes present in the country. This equation is based on the biome

areas (a) and the relative importance of biomes (x) given by the random forest. We have improved this explanation in the supplementary by better describing the meaning of the equations:

We use the random forests to predict a baseline value of market and non-market benefits per hectare. Next, we predict how these per-hectare values change when the extent of each biome is increased by 10 percentage points (pp), holding all other biome areas constant. We use the relative sizes of the resultant per-hectare values as indicators of the relative importance of each biome to the country's natural capital stocks, using these values to estimate the contribution of each biome to the total natural capital estimates for a given country.

To further explain the procedure without loss of generality, we imagine a hypothetical country with only two biomes: B1 and B2. Using the random forest generated for that country, and focusing on non-market natural capital, we obtain the new values per hectare, \widehat{es}_{B1} and \widehat{es}_{B2} , by increasing biomes B1 and B2 cover 10pp, respectively:

$$\widehat{es}_{B1} = RF_{n,b}(cover_{B1+10}, cover_{B2}, GDPpc, PercCovered) \quad \text{eq. 6a}$$

$$\widehat{es}_{B2} = RF_{n,b}(cover_{B1}, cover_{B2+10}, GDPpc, PercCovered) \quad \text{eq. 6b}$$

From the values above, we can obtain the parameter x_{B1} , a scaling factor to express \widehat{es}_{B2} in terms of \widehat{es}_{B1} , so $\widehat{es}_{B2} = x_{B1}\widehat{es}_{B1}$. Assuming that this relationship holds for comparing two hectares fully covered by biomes B1 and B2, respectively, we can write the value of non-market benefits provided by one hectare of Biome 2 ($es_{2018,c,B2}$) in terms of the value of non-market benefits provided by one hectare of Biome 1 ($es_{2018,c,B1}$) as follows:

$$es_{2018,c,B2} = x_{n,c,B1} es_{2018,c,B1} \quad \text{eq. 7}$$

Rewriting the equation for non-market natural capital (equation 3) for the hypothetical country in 2018,

$$nN_{2018,c} = \frac{a_{2018,c,B1} es_{2018,c,B1} + a_{2018,c,B2} es_{2018,c,B2}}{r} \quad \text{eq. 8}$$

$$\Rightarrow r * nN_{2018,c} = a_{2018,c,B1} es_{2018,c,B1} + a_{2018,c,B2} es_{2018,c,B2} \quad \text{eq. 9}$$

Substituting $es_{2018,c,B2}$ from equation 7

$$r * nN_{2018,c} = a_{2018,c,B1} es_{2018,c,B1} + a_{2018,c,B2} x_{n,c,B1} es_{2018,c,B1} \quad \text{eq. 10}$$

$$\Rightarrow es_{2018,c,B1} = \frac{r * nN_{2018,c}}{a_{2018,c,B1} + a_{2018,c,B2} x_{n,c,B1}} \quad \text{eq. 11}$$

All the variables in the right-hand side of equation 11 are known so we can estimate $es_{2018,c,B1}$.

So to answer your question directly of whether the random forest is used to make future prediction, the random forest provides one important component to make these future predictions (the values of benefits per hectare of biome at the country level), but in order to obtain future projections we still adjust such values by the income elasticity, total area cover elasticity, and vegetation carbon elasticity. We believe that this is much clear now with the flow chart of the study design presented above:

Redacted

For Legends: See Extended Data Figure 1

And how do you factor in income per capita? How should I understand your method and results compared to more traditional methods of benefits transfer that would use an explicit income elasticity?

Regarding income per capita, we note that we are not using the random forest to transfer values from study to policy sites; we already have country-specific values from the World Bank that should account for the impacts of income on willingness to pay for non-market benefits of natural capital. Still, there are two crucial points where this factor is included within the study design. First, income per capita is included as an independent variable in the random forest model to determine the per-hectare benefits of different biomes within a country. The inclusion of per-capita income here allows the contribution of a given biome to a country's total value of natural capital to vary with income and also allows the portion of value from a given biome associated with market or non-market benefits to vary with income. Given the flexibility in estimation associated with random-forest algorithms, this approach allows for a relatively rich interaction between income and the value of benefits from natural capital compared to a frequently-adopted value-transfer approach that might assume a constant income elasticity of WTP and then modify the study-site WTP by the ratio of the per capita income in the policy site to the study site raised to the income elasticity of WTP.

Next, we also incorporate the income elasticity of per-hectare values in our projections of future ecosystem-services values. Specifically, we use the income elasticity obtained from the regression model whose results are shown in Table S3:

Supplementary Table 3. Regression table of the ecosystem benefits elasticities, using the VEGS database.

	Dependent variable:
	log(Value per hectare)
log(cveg)	0.282**
Reference level: Non-market	
	(0.143)
Market	8.046
	(5.004)
log(Area)	-0.1032*
Reference level: Non-market	
	(0.049)
log(gdp_pc)	0.593**
	(0.299)

log(Percentage covered)	0.764***
	(0.286)
log(cveg)*Market	0.307
	(0.316)
log(area)*Market	0.002
	(0.075)
log(gdp_pc) * Market	-0.969*
	(0.498)
Fixed EffectsConstant Reference Level: Non-market	Country Valuation Methodology
Observations	483
R ²	0.261
Adjusted R ²	0.152
Residual Std. Error	2.476 (df = 420)
F Statistic	6.07*** (df = 8; 420)
Note:	*p<0.1; **p<0.05; ***p<0.01

So the future values of non-market ecosystem services when using the “Green” RICE50+ model are adjusted to increase by 0.593% for each percent increase in income. We acknowledge that the role of income elasticity was not clear in the previous version of the manuscript, and we have included the below text in the new version of the supplementary materials:

Finally, we obtain the annual flow of non-market benefits by using the formula of the net present value of a benefit flow in perpetuity, i.e. multiplying the non-market natural capital by the discount rate (3% as a chosen value) and allowing ecosystem services to increase based on the country’s percent increase on GDP per capita ($\% \Delta GDPpc$) times the income elasticity obtained in Table S3. Therefore $ES_{c,t} = nN_{c,t} * 0.03 * (1 + 0.00596 * \% \Delta GDPpc_{c,t})$.

In relation to your inquiry about the connection to benefit transfer models, we acknowledge the potential interest of readers in this subject. To address this, we’ve included a detailed clarification within the main body of the text:

The World Bank data offers the advantage of a uniform accounting and valuation method for natural capital in all countries in the world. While there are a number of well-established methods available to value non-market environmental amenities,^{31,32} these methods rely on expertise and resources that

are generally unavailable in many contexts, leading to data gaps for many non-market environmental amenities and parts of the world, even in the comprehensive VEGS database. As a result, approaches for transferring value estimates from study sites to policy locations have been developed, although there are a number of challenges.³³⁻³⁵ Our use of the World Bank data avoids the need to navigate the tradeoffs of the benefit transfer approach, though our research question of interest requires us to adopt an attribution method to allocate the country-level values in this dataset to the various biomes within a country.

Drafting of the main paper

I appreciate the efforts the authors have made to improve the clarity of the main paper, especially dealing with the overlapping frameworks for thinking about natural capital, ecosystem goods/services, etc. But still I feel like the drafting could be further improved. The first section seems to jump between basic conceptual stuff, a description of the method, and a statement of the contribution. It would be best to parse these, and I particularly missed a self-contained, clear statement of the sequence of steps contained within the method. I know that in journals like these there is not supposed to be a long methods section, but surely comprehension of the paper would be aided by a long paragraph or a couple of paragraphs that basically lay out the recipe, step by step.

We greatly appreciate your insightful feedback. We concur that the manuscript's clarity could be further improved by providing a more sequential and discernible layout of the methodology and contribution of our study.

In response to your suggestion, we have taken significant steps to revise the manuscript's structure, order of ideas, and level of detail to provide a more coherent and focused narrative. We have shortened the explanations of basic concepts, opting for a broader overview instead of a detailed breakdown. While many of these explanations are crucial, we believe that they are best suited in the supplementary materials section, so we moved them there.

Specifically, our revised introduction now consists of four concise paragraphs that provide: 1) an introductory paragraph and motivation; 2) the theoretical framework and aim of the study; 3) a straightforward, self-contained overview of the methodology; and 4) the scope of the analysis. We have also included a reference to the new supplementary materials section titled "study design," which offers a more detailed depiction of the methodology:

Climate change has direct, widespread, and long-lasting implications for the structure and functioning of ecosystems around the world¹⁻³. These changes will alter the market and non-market benefits people derive from natural systems.⁴ However, current economic estimates of climate change damages that inform climate policy do not fully account for these changes or include dated assessments of climate change effects on ecological functioning⁵⁻⁷. Previous work has shown that

accounting for climate impacts on natural systems can have large effects on estimated climate damages⁸, and several papers have called for improved assessments of the effects of climate change on human well-being via its impacts on ecosystems^{9,10}.

Human well-being can be divided into three components: goods and services exchanged in markets (hereafter referred to as market benefits); use benefits from nature that are not usually exchanged in markets (hereafter referred to as non-market benefits); and non-use values from biodiversity and ecosystems attached only to their existence (Fig. 1). In this study, we focus on the first two components. Well-being arises from a stock of valuable assets (or types of wealth) that includes human capital, manufactured capital, and natural capital^{11–15}. Here, we expand a regional benefit-cost climate integrated assessment model (IAM) to explore the welfare effects of climate impacts on terrestrial natural capital. This disaggregated, global analysis reveals how the burden of foregone value from reduced biome spatial extent is borne differentially in countries around the globe.

To conduct this analysis, we combine several models and datasets (See Fig. S1). First, we attribute the country-level data on natural capital stock values in the World Bank Changing Wealth of Nations 2021 report¹¹ to the biomes contained in each country's borders. To do so, we apply random forest algorithms to subsets of ecosystem-service valuation studies in the Values of Ecosystem Goods and Services (VEGS) database¹⁶ based on studies from locations with similar biome distributions for each country around the globe (Fig S1A). Next, we estimate future trajectories of country-level natural capital stocks under climate change-driven biome range shifts, allowing us to quantify country-level damage functions for climate-driven impacts on natural capital (Fig S1B). Then, we use an extended version of the open-source RICE50+ Integrated Assessment Model¹⁷ following initial work in Bastien-Olvera and Moore⁸, to estimate the effect of changing natural capital on the market benefits (as measured by gross domestic product; GDP) and non-market benefits in a fully consistent framework (Fig S1C), enabling future research under diverse scenarios and policies, including emissions optimization and different levels of cross-country cooperation, both on climate and ecosystems conservation.

The market and non-market benefits included in the World Bank dataset are limited, and their estimated values reflect a lower bound on the true values, meaning that our estimated welfare effects of climate-driven changes in natural capital are conservative. In our research, market-based natural capital (mN) represents the projected value from future timber revenues, and non-market natural capital (nN) is intended to estimate the value of forest-related recreational services, water resources, non-timber forest products, and the inherent value of protected areas. The flow of benefits from a country's market natural capital stock, the market environmental benefits, are included in GDP, while the non-market benefits that flow from the non-market natural capital stock are not. These two types of natural capital comprise a larger fraction of total national wealth in lower- and middle-income countries¹⁸ (Fig. 1). Consequently, these countries are most vulnerable to any climate change-induced loss of forest cover or changes in terrestrial vegetation patterns despite their limited contribution to man-made climate change.

Following the introduction, we have implemented distinct section headings and rearranged paragraphs to enhance the differentiation between: (1) "Climate-induced biome range shifts", (2) "Impacts of biome range shifts on natural capital", and (3) "The effect of damaged natural capital on human well-being".

We believe these changes will substantially enhance the clarity of our manuscript and facilitate a more fluid comprehension of our research process and findings. Thank you for your constructive feedback, which has been invaluable in improving our paper.

Minor comments

You want to think of non-market benefits as being distinct from existence value, but to me you have the use versus non-use dichotomy, and then the market versus non-market dichotomy, and non-use values are non-market values. That's what I was brought up on. Maybe things have changed.

We appreciate your insights. As context, we want to point out first that responding also to a comment from Reviewer 1 asking for further clarification of the terminology, we settled on using the market versus non-market dichotomy. We agree, existence values are a non-use subset of non-market benefits. Conversely, non-market ecosystem services like clean air pertain to the "use" segment of non-market benefits. In the new version of the manuscript though, we clarify that due to data constraints the references to "non-market values" in the text exclude non-use values. In paragraph two we have write:

Human well-being can be divided into three components: goods and services exchanged in markets (hereafter referred to as market benefits); use benefits from nature that are not usually exchanged in markets (hereafter referred to as non-market benefits); and non-use values from biodiversity and ecosystems attached only to their existence (Fig. 1). In this study, we focus on the first two components.

Further, in response to this comment, we have slightly modified Figure 1 to include a box that groups "use values" for the interested reader:

We have contextualized our work in the evolving literature on nature’s contributions to welfare in the discussion section, and hope that our efforts to be more consistent with our terminology have improved the clarity of the writing.

You add a sensitivity analysis to the DGVM to the Supplementary Materials, but you don’t discuss the results anywhere, do you? This seems like an oversight given this is an important part of your response to Reviewer 1. I think it should be discussed in the main paper.

The reviewer is right in pointing out this oversight on our side. In response to this and specifically to the comment of Reviewer 1, we expanded our supplementary materials to include figures mirroring Figure 3 for the ORCHIDEE and CARAIB models, as show below:

Supplementary Fig 6. Market and non-market benefits by country and biome. A. Distribution of country-level benefits per hectare of biome. Red and blue boxes show market and non-market benefits, respectively. The middle line in the box shows the median, the box covers the first to the third quartile, whiskers show the full range of the data except for some outliers shown in black points. B. Total yearly benefits per geographic region per biome. Note that values are reported as equivalent fractions of GDP as a comparison point only, non-market benefits are not captured in standard GDP accounting practices. C. Changes in the market and non-market natural capital for 1 degree of warming. Using model output from CARAIB⁵.

Supplementary Fig 7. Market and non-market benefits by country and biome. A. Distribution of country-level benefits per hectare of biome. Red and blue boxes show market and non-market benefits, respectively. The middle line in the box shows the median, the box covers the first to the third quartile, whiskers show the full range of the data except for some outliers shown in black points. B. Total yearly benefits per geographic region per biome. Note that values are reported as equivalent fractions of GDP as a comparison point only, non-market benefits are not captured in standard GDP accounting practices. C. Changes in the market and non-market natural capital for 1 degree of warming. Model output from ORCHIDEE-DGVM⁴.

Regarding Figure 4, we now provide a supplementary figure with country-level trajectories for CARAIB and ORCHIDEE-DGVM as well as labels for the data points in 2100 (visible on page 17 of the Supplement):

Supplementary Figure 8. Trajectories of the annual change in market and non-market benefits are shown for 57 regions in “Green” RICE50+ with respect to simulations without climate change impacts. Labels in 2100 show the code of the regions as given in RICE50+. Error bars show the standard errors of the estimated damage functions under different general circulation model outputs. Dashed lines show the population-weighted mean values in 2100. Upper panel: Results under LPJ-GUESS simulation output (shown in Figure 4).

We have now rectified this omission by citing the supplemental figure in the caption for Figure 4. The revised caption (visible on page 7 of the manuscript) reads:

Fig. 4. Market and non-market impacts under SSP2-6.0 scenario. A. Trajectories of the annual change in market benefits (shown as annual GDP change) and non-market benefits are shown for the 57 countries and regions in RICE50+ with respect to simulations without climate-change impacts. Error bars show the standard errors of the estimated damage functions under different general circulation model outputs. Dashed lines show the population-weighted mean values in 2100. A comparison of these results for different DGVMs with region labels is presented in Figure S8. **B.** Cumulative global population plotted against respective levels of annual GDP change in 2100. **C.** Cumulative global population plotted against respective levels of annual non-market benefits change in 2100.

Also, when mentioning the results in the main text, we now describe the specific result from other DGVMs, for the non-market benefits section (on page 7) now it reads as follows:

The global population-weighted average change in non-market benefits value in 2100 is -9.2% relative to the baseline (1.4% and -4.9% using damage functions based on CARAIB and ORCHIDEE-DGVM).

For the market benefits section (on page 8) now it reads as follows:

The global population-weighted average change in GDP by 2100 is -1.3% of the baseline GDP (-0.7% and -1.4% using output from CARAIB and ORCHIDEE DGVMs).

Further, the discussion section (on pages 9) brings back the important distinction between our preferred DGVM and the others:

[...] it is worth noting that the results based on DGVMs other than LPJ-GUESS, which generally show smaller GDP and non-market benefits disruptions, should be interpreted as conservative estimates, as the output used from these models provides proportional, not absolute, biome shifts, thus

representing the effect of biome replacement without considering actual area change as with LPJ-GUESS.

We hope this addition provides clarity, and thank you for bringing this to our attention.

One of your key statistics is the cumulative discounted cost of natural capital losses, as a percentage of global GDP in 2020. But does it make sense to compare these two quantities? Surely a like for like comparison would be with cumulative discounted global GDP.

We appreciate your concern about the potential ambiguity of our comparison. We wanted to communicate the impacts in a way that would resonate with a non-economic audience, but we understand that our original method could have been misleading. To rectify this, we have now focused on presenting the annual change in GDP (as shown in our main results in Figure 4) consistently across the manuscript.

In addition, we have taken steps to emphasize the distributional effects. To this end, we have created a new graph that reflects the distribution of climate impacts on GDP. We believe this new graph will be more relatable and comprehensible to the broad audience of *Nature*.

Please find the updated language and references to the graph in the abstract and the results section:

Abstract:

Our results show that the annual population-weighted mean global flow of non-market ecosystem benefits valued in the World Bank's wealth accounts will be reduced by 9.4% in 2100 under the Shared Socioeconomic Pathway SSP2-6.0 reference scenario, and that the global population-weighted average change in GDP by 2100 is -1.3% of the baseline GDP. Because lower-income countries are more reliant on natural capital, these GDP effects are regressive. Approximately 90% of these damages are borne by the poorest 50% of countries and regions, while the wealthiest 10% experience only 2% of these losses.

Results:

The bottom 50% of the countries and regions, in terms of GDP per capita, bear approximately 90% of these damages, while the top 10% only face 2% of these losses (See Figure S11).

We have also added a new supplementary figure to illustrate this point:

Supplementary Figure 11. Distribution of total damages in 2100 across different countries and regions ordered by GDP per capita. Vertical solid line shows the 50% cutoff, horizontal dashed line shows the percentage of total damages borne by the bottom 50% (88.8%). The distributional pattern remains very similar under 2100 GDP per capita estimates.

I think the idea of fixed land management/use should be mentioned more prominently. I understand why this is useful to focus on the pure climate effect, but some discussion of how changing land management/use could affect the results would be useful.

We thank the reviewer for this suggestion, we have now elaborated more on this topic in the discussion section:

Our methodology was designed to identify the welfare impacts of climate-driven changes in biome extent and quality in regions around the world. As a result, we have set aside the issue of habitat conversion due to human activity, which has been shown to have a significant negative impact on ecosystem-service provision and biodiversity^{50,51}. Our estimated impacts on natural capital and the flows of benefits that it produces should be lower than those observed in practice, due to predicted land-use and land-cover changes over the rest of the century, even with our findings that the marginal non-market benefit of biomes is decreasing in biome size.

Further, mention how this aspect could be incorporated in the future:

Other simplifications in the simulations could be relaxed by using the scenarios framework developed to assess climate change policies. For example, the SSP-RCP scenario matrix quantifies the levels of policy-led land-use change to comply with emissions, which in some cases (e.g. expansion of biofuel crops in natural areas) might severely decrease biomes coverage.

We trust that the clarifications provided above and the significant improvements made to the manuscript and supplementary materials sufficiently address your remaining concerns. Your insightful feedback has greatly contributed to refining our work, for which we are genuinely thankful.

Referee #3 (Remarks to the Author):

The paper entitled **Biome range shifts under climate change: impacts on country-level macro-economic production and ecosystem services** analyses the effects of climate change on ecosystem services (market and non-market) at the country level. The results show the disparity of negative impacts in developed and developing countries.

I answered every single point in the first round of reviewing. This second round I will answer specific comments addressed by the authors.

I think that the paper improved satisfactorily. I appreciate the clarification among concepts, mainly the market value and the non-market values embedded in natural ecosystems associated with recreation services, water quantity, water quality, non-timber forest products, and the overall value of protected areas. However, I still have concerns about the World Bank's methodology, but I understand that the authors rely on it. Regarding the protected areas, I do not support the assumption of valuing them as the unrealized revenue that those areas had if they were converted to agricultural fields. I understand the approach; however, I am sure that the authors have added many details to clarify their theoretical framework.

I think it is good that authors agreed that GDP is an incomplete measure of human welfare and adding explanation on this. I hope that in the short-term, besides recognizing the limitations, science uses different approaches and elements in their frameworks. Moreover, it is true that the authors include non-market values but the World Bank methods, from my perspective, are not clear and transparent about how they quantified recreation, water quantity, water quality, and non-wood forest. We know that even at national level those data are well founded and even worse at subnational level: that is why using global data sets (sometimes uncritically) result in not reliable outputs.

Regarding the use of different models, I think it was an excellent idea to include the diverse DVM (LPJ-GUESS 1, ORCHIDEE-DGVM2, and CARAIB) instead of using one.

The clarification of the methodology to assign different natural capital estimates to the different types of biomes now it is clearer in the Supplementary Material than the previous version.

I appreciate the authors recognize the importance of many disciplines too address environmental challenges because it was narrow their position in the previous version. However, it is evident that one sentence do not amplify the necessity of going beyond this kind of approaches of valuing ecosystems (economically or non-economically).

I still support that the paper is not original and that the impacts of climate change on ecosystems have been widely studied for decades in more interesting approaches (I added the references in the previous round).

I still support that is not original and that the impacts of climate change on ecosystems have widely studied for decades. The difference is that they are trying to link this shift with the human welfare

but not in a creative way. The answer of the authors is logical. They defended their approach by supporting their simplification to answer a particular question. I understand them; however, I showed them, in the first round of reviews, a short list of many examples of similar efforts from a more ecological perspective, which in some way, from my perspective, is more useful. Logical results like showing higher impacts of climate change in developing countries do not give novel scientific developments. Besides, showing that impacts will be 232% of the GDP in 2020, makes people wrongly associate the GDP loss as the main element that we have to be worried about, instead of the socio-ecosystem risk (such as loss in ecosystem integrity or ecological functions).

I support we encourage different approaches in science by giving more realistic and plausible outputs that help us to address environmental and political possible solutions.

We are pleased to receive your recognition of the significant efforts we have made to improve the manuscript and that you find these improvements satisfactory. Your remarks have been vital in enhancing clarity and context throughout the paper.

We understand and respect your concerns about our approach and the World Bank's methodology. It is clear that there are multiple ways to assess the impacts of climate change on ecosystems, and the debate about the most effective and reliable methods is ongoing in the scientific community.

Regarding your points on the originality of the work, we maintain that our effort to link biome range shifts and human welfare through the lens of economic impacts offers a novel perspective in this field. Although the negative impacts of climate change on developing countries have been established, we believe that quantifying these impacts in terms of GDP and loss of ecosystem services brings a new understanding to the issue that can inform policy and decision-making.

As for the concern of GDP loss overshadowing the socio-ecosystem risk, we agree that it is essential not to overlook the broader ecological implications. As a result, we have tried to balance our discussion of the economic impacts with acknowledgment of the non-market benefits and the inherent value of ecosystems.

Again, we appreciate your perspective and feedback. We believe that fostering this kind of dialogue is crucial in advancing our collective understanding of the complex challenges posed by climate change. Your comments have certainly enriched the discussion in our paper.

Reviewer Reports on the Second Revision:

Referees' comments:

Referee #1 (Remarks to the Author):

The revised manuscript has satisfyingly addressed my previous comments. Adding the results of the other DGVMs provide valuable insights to the uncertainty of the analysis. I appreciate it very much. I also like Figure S1 summarizing the workflow of the whole methodology employed in this study. This enhances the clarity of the manuscript a lot.

Nevertheless, there are a few places that can be improved further, which are listed below.

Line 50: "natural capital" can be followed by "(both market and non-market ones)" to be consistent with Figure 1.

Figure S8: The horizontal dashed line in the top plot (LPJ-GUESS) does not match the one in Figure 4a in the manuscript. Please double check if the line is correctly plotted in this figure.

Figure S11: please add similar plots for other DGVMs to confirm the pattern. I suppose that the pattern should not differ much from that of LPJ-GUESS? If so, it would provide a good support to the main argument of the manuscript.

Referee #2 (Remarks to the Author):

I think this paper has improved to the point where "enough is enough" and if the editors are otherwise happy it should be published, subject to making some minor changes as listed below. I don't need to see another draft.

Congratulations and well done for all the quite significant improvements made along the way. In particular, upon reading this third version I felt much more confident that I understood the many methodological steps and how they relate to each other. The results are intuitive and of plausible magnitudes.

Regarding my previous comments about the production function, I am still concerned that there is a fundamental mismatch between what non-market natural capital S represents in theory and what the World Bank natural capital accounts actually include. The root cause of my confusion was that several elements of the World Bank numbers are environmental amenities that conceptually enter individual utility separate from GDP. However, I do not see an obvious way out of this problem, so I

think we can live with it.

Minor comments

In paragraph three of the main paper, is it worth explicitly mentioning the vegetation models?

p5: I would probably avoid claiming your method is superior to explicit benefits transfer, as such a claim would need further investigation and substantiation. I know you don't quite say this, but you strongly imply it.

p5 lines 156-7: typo here as mathematical symbols not appearing properly

line 215-6: the difference in results between the vegetation models deserves a comment.

Referee #4 (Remarks to the Author):

Major comments

The authors used multi-source datasets and multi-model approach to assess the impact of biome transitions under future climate change scenarios on the value of natural capital on a global scale. This interdisciplinary research provides a new perspective for evaluating economic losses due to climate change.

The setting of scenarios needs to be presented more clearly. There is an error in the key conclusion in the abstract: "Our results show that the annual population-weighted mean global flow of non-market ecosystem benefits valued in the World Bank's wealth accounts will be reduced by 9.4% in 2100 under the Shared Socioeconomic Pathway SSP2-6.0 reference scenario, and that the global population-weighted average change in GDP by 2100 is -1.3% of the baseline GDP." Is it 9.4% here correct? Or should be 9.2% as stated in the main text: "The global population-weighted average change in non-market benefits value in 2100 is -9.2% relative to the baseline." Also, "under the Shared Socioeconomic Pathway SSP2-6.0 reference scenario" could be replaced with "under the Shared Socioeconomic Pathway SSP2-6.0 relative to the reference scenario" for clarity. It is advised to use a consistent terminology for "reference scenario" or "baseline scenario" throughout the paper, and provide an explanation for "reference scenario" in the text and methods section. Adding scenario narratives in the methods will help readers to better understand and follow the study.

The climate-induced shifts in biome ranges deriving from the DGVMs lay the basis for the cost-benefit analysis. I am wondering if the results have been validated or compare with other model simulation results. The Fig. 2C shows that the fractions of natural vegetation cover decrease in the tropical and subtropical biomes in Latin America, Africa, and Asia. While Fig. 2D indicates that these regions experience non-significant change in the vegetation carbon content. Such results are confusing. If climate warming exert detrimental impacts on the growth and productivity of natural

vegetation in tropical biomes with high temperature and aridity, the carbon content of the vegetation as a proxy for ecosystem overall health would probably be influenced under the warming stress.

In the discussion section, the authors could consider adding the implications and significance of their research results for formulating climate policies. In China, the assessment of natural capital for government officials upon leaving their positions is an aspect of their performance evaluation. Valuing natural assets allows for a more robust theoretical foundation to support policy management.

Minor comments:

Places that need to be cited in the abstract should have appropriate citations added.

Authors need to standardize the formatting of figure captions. “(Fig S1A)” should be “(Fig. S1A)”. There is a missing period. Both “Fig. 2” and “(Figure 2D)” are used in the main text.

I suggest making standardized and uniform modifications to the format of the reference list and Supplementary Materials. Some equations in the Supplementary Materials seem to be incomplete.

$$N(t) = \sum_i [p_i(t)n_i(t)] \quad , \text{ eq. 1}$$

$$N(t) = \sum_i \sum_t t^{\Delta t} (B(t)) / ((1+r)^t) \quad \text{eq. 2}$$

Supplementary Materials, Supplementary Fig 4 and Fig 5: “Changes in carbon vegetation content” should be “Changes in vegetation carbon content”.

Author Rebuttals to Second Revision:

Title: Unequal Climate Impacts on Global Values of Natural Capital

Response to Referees

Referee #1 (Remarks to the Author):

The revised manuscript has satisfyingly addressed my previous comments. Adding the results of the other DGVMs provide valuable insights to the uncertainty of the analysis. I appreciate it very much. I also like Figure S1 summarizing the workflow of the whole methodology employed in this study. This enhances the clarity of the manuscript a lot.

We appreciate your positive feedback and are pleased that the additions and revisions have addressed your concerns.

Nevertheless, there are a few places that can be improved further, which are listed below.

Line 50: “natural capital” can be followed by “(both market and non-market ones)” to be consistent with Figure 1.

We acknowledge your point regarding consistency between the text and Figure 1. However, we feel that gradually introducing the reader to the overarching concept of natural capital before delving into its subtypes (i.e., market and non-market natural capital) aids in comprehension. To delve into these subtypes, we have a dedicated section further in the text (previously line 69) where we detail this subdivision. In this context, our intent was to prioritize a gradual introduction to the concept for the reader, even if it means a slight deviation from the immediate consistency with Figure 1.

Figure S8: The horizontal dashed line in the top plot (LPJ-GUESS) does not match the one in Figure 4a in the manuscript. Please double check if the line is correctly plotted in this figure.

We appreciate the reviewer's attention to detail regarding Figure S8. The discrepancy has been addressed, and the figures now are consistent.

Fig. 4. Market and non-market impacts under SSP2-6.0 scenario. **A.** Trajectories of the annual change in market benefits (shown as annual GDP change) and non-market benefits are shown for the 57 countries and regions in RICE50+ with respect to simulations without climate-change impacts, from now till 2100. Error bars show the standard errors of the estimated damage functions under different general circulation model outputs. Dashed lines show the population-weighted mean values in 2100. A comparison of these results for different DGVMs with region labels is presented in Extended Data Fig. 8. **B.** Cumulative global population plotted against respective levels of annual GDP change in 2100. **C.** Cumulative global population plotted against respective levels of annual non-market benefits change in 2100.

Extended Data Fig. 8. Trajectories of the annual change in market and non-market benefits are shown for 57 regions in “Green” RICE50+ with respect to simulations without climate change impacts. Labels in 2100 show the code of the regions as given in RICE50+. Error bars show the standard errors of the estimated damage functions under different general circulation model outputs. Dashed lines show the population-weighted mean values in 2100. Upper panel: Results under LPJ-GUESS simulation output (shown in Figure 4). Middle panel: Results under ORCHIDEE-DGVM simulation output. Lower panel: Results under CARAIB simulation output.

Figure S11: please add similar plots for other DGVMs to confirm the pattern. I suppose that the pattern should not differ much from that of LPJ-GUESS? If so, it would provide a good support to the main argument of the manuscript.

We appreciate the reviewer's suggestion. Following this, we have expanded Extended Data Fig. 11 to include plots for the other DGVMs. Indeed, the patterns from these models are consistent with those from LPJ-GUESS, reinforcing the main argument of our manuscript. This addition provides a more comprehensive perspective for our readers.

Extended Data Fig. 11. Distribution of total damages in 2100 across different countries and regions ordered by GDP per capita. Vertical solid line shows the 50% cutoff, horizontal dashed line shows the percentage of total damages borne by the bottom 50%, which is 89%, 86%, and 82% for the LPJ-GUESS, ORCHIDEE-DGVM and CARAIB, respectively. The distributional pattern remains very similar under 2100 GDP per capita estimates.

Referee #2 (Remarks to the Author):

I think this paper has improved to the point where "enough is enough" and if the editors are otherwise happy it should be published, subject to making some minor changes as listed below. I don't need to see another draft.

Congratulations and well done for all the quite significant improvements made along the way. In particular, upon reading this third version I felt much more confident that I understood the many methodological steps and how they relate to each other. The results are intuitive and of plausible magnitudes.

We sincerely appreciate your feedback and the recognition of the improvements made to the manuscript. Your insights throughout the review process have been invaluable in refining the methodology and presentation of our results. We will address the minor changes you've suggested in the subsequent sections.

Regarding my previous comments about the production function, I am still concerned that there is a fundamental mismatch between what non-market natural capital S represents in theory and what the World Bank natural capital accounts actually include. The root cause of my confusion was that several elements of the World Bank numbers are environmental amenities that conceptually enter individual utility separate from GDP. However, I do not see an obvious way out of this problem, so I think we can live with it.

We appreciate your continued attention to the complexities of our approach concerning the production function. Indeed, our approach acknowledges the imperfect alignment between the theoretical representation of non-market natural capital S and the specific components within the World Bank accounts in the methods section:

“due to the lack of a global measure of such services [S] consistent with the World Bank accounts, we use the global aggregate of non-market natural capital as a proxy for it. This factor, along with the standard total factor productivity (TFP) could be understood as an adjusted total factor productivity.”

Recognizing this mismatch, we have updated our Methods sections to emphasize both its potential overestimations and underestimations.

“Proxying for S with the global aggregate of non-market natural capital may on the one hand overstate the true value, as it includes use values of environmental amenities that may not be a direct enabler of GDP, and potentially understate S, as it misses, for instance, the global benefits from climate services.”

We believe that by explicitly addressing these nuances, the readers can make informed interpretations of our results.

Minor comments

In paragraph three of the main paper, is it worth explicitly mentioning the vegetation models?

We thank you for the suggestion. Incorporating explicit mention of the dynamic global vegetation models in paragraph three certainly provides a clearer context for our methodology. The text has been revised:

Next, we estimate future trajectories of country-level natural capital stocks under climate change-driven biome range shifts using output from dynamic global vegetation models, allowing us to quantify country-level damage functions for climate-driven impacts on natural capital (Extended Data Fig. 1b).

p5: I would probably avoid claiming your method is superior to explicit benefits transfer, as such a claim would need further investigation and substantiation. I know you don't quite say this, but you strongly imply it.

We appreciate your perspective on this matter and have taken it into account. As a result, we've made adjustments to the main text to more accurately reflect our approach without drawing this comparison.

Our use of the World Bank data offers an alternative to the benefit transfer approach, though our research question of interest requires us to adopt an attribution method to allocate the country-level values in this dataset to the various biomes within a country.

p5 lines 156-7: typo here as mathematical symbols not appearing properly

Thank you for pointing that out. We have rectified the typographical error in the mathematical symbols.

line 215-6: the difference in results between the vegetation models deserves a comment.

We appreciate your observation about the variability in results across different vegetation models. We concur that these differences deserve comment in addition to the part of the discussion that reads: In relation to this, it is worth noting that the results based on DGVMs other than LPJ-GUESS, which generally show smaller GDP and non-market benefits disruptions, should be interpreted as

conservative estimates, as the output used from these models provides proportional, not absolute, biome shifts, thus representing the effect of biome replacement without considering actual area change as with LPJ-GUESS.

To address your suggestion, we've amended the manuscript to highlight this variation more explicitly:

The global population-weighted average change in non-market benefits value in 2100 is -9.2% relative to the baseline. Notably, when using damage functions based on CARAIB and ORCHIDEE-DGVM, the figures are 1.4% and -4.9% respectively, highlighting the variation across different vegetation models

This added nuance, we believe, offers readers a clearer understanding of the results within the context of different DGVMs.

Referee #4 (Remarks to the Author):

Major comments

The authors used multi-source datasets and multi-model approach to assess the impact of biome transitions under future climate change scenarios on the value of natural capital on a global scale. This interdisciplinary research provides a new perspective for evaluating economic losses due to climate change.

Thank you for recognizing the novelty of our interdisciplinary approach. We endeavored to provide a new perspective in evaluating economic losses due to climate change, and we are pleased that this was acknowledged.

The setting of scenarios needs to be presented more clearly. There is an error in the key conclusion in the abstract: "Our results show that the annual population-weighted mean global flow of non-market ecosystem benefits valued in the World Bank's wealth accounts will be reduced by 9.4% in 2100 under the Shared Socioeconomic Pathway SSP2-6.0 reference scenario, and that the global

population-weighted average change in GDP by 2100 is -1.3% of the baseline GDP." Is it 9.4% here correct? Or should be 9.2% as stated in the main text: "The global population-weighted average change in non-market benefits value in 2100 is -9.2% relative to the baseline."

Thank you for pointing out this inconsistency. The correct number is 9.2%. We have now corrected the number in the abstract:

Our results show that the annual population-weighted mean global flow of non-market ecosystem benefits valued in the World Bank's wealth accounts will be reduced by 9.2% in 2100 under the Shared Socioeconomic Pathway SSP2-6.0 with respect to the baseline, and that the global population-weighted average change in GDP by 2100 is -1.3% of the baseline GDP.

Also, "under the Shared Socioeconomic Pathway SSP2-6.0 reference scenario" could be replaced with "under the Shared Socioeconomic Pathway SSP2-6.0 relative to the reference scenario" for clarity. It is advised to use a consistent terminology for "reference scenario" or "baseline scenario" throughout the paper, and provide an explanation for "reference scenario" in the text and methods section. Adding scenario narratives in the methods will help readers to better understand and follow the study.

Thank you for pointing out the ambiguity in our terminology. To clarify, we used "reference scenario" to indicate the specific SSP2-6.0 quantification in the RICE50+ model, while "baseline scenario" denotes a simulation without climate change impacts. Though we define "baseline scenario" in the text, the one-time use of "reference scenario" inadvertently caused confusion. In line with your recommendation, we've removed the term "reference" in relation to SSP2-6.0 for clarity. We've made this adjustment in the abstract, the only section where "reference scenario" was mentioned. Thank you for your insightful feedback:

Our results show that the annual population-weighted mean global flow of non-market ecosystem benefits valued in the World Bank's wealth accounts will be reduced by 9.2% in 2100 under the Shared Socioeconomic Pathway SSP2-6.0 with respect to the baseline, and that the global population-weighted average change in GDP by 2100 is -1.3% of the baseline GDP.

The climate-induced shifts in biome ranges deriving from the DGVMs lay the basis for the cost-benefit analysis. I am wondering if the results have been validated or compare with other model simulation results. The Fig. 2C shows that the fractions of natural vegetation cover decrease in the tropical and subtropical biomes in Latin America, Africa, and Asia.

Thank you for highlighting the importance of model validation. We utilized well-established process-based dynamic vegetation models. These models have been rigorously validated and compared in prior research and diligently follow the protocols of the Inter-sectoral Intercomparison Model (ISIMIP) as documented by Frieler et al. (2017). As such validations have been already done extensively elsewhere and the rigor of these models have been well-established, we do not repeat model validations here.

LPJ-GUESS, our preferred model, has been validated by Hickler et al. (2012) and Kukla et al. (2021). Some validation work for the CARAIB model can be found in Henrot et al. (2017), Jacquemin et al. (2021), Minet et al. (2015). ORCHIDEE-DGVM validations are documented by Yue et al. (2020). Although some of these studies suggest areas for further refinement in these DGVMs, there is a consensus within the scientific community that they are robust, valuable, and among the best available tools for predicting vegetation dynamics.

In our research, in addition to providing results under the three main Dynamic Vegetation Models, we accounted for the potential sensitivity of vegetation dynamics to climate forcings as given by different General Circulation model outputs. To do that, we derived our damage functions from multiple runs that utilized diverse climate output data as forcings given by HadGEM2-ES, GFDL-E5M2M, IPSL-CM5-LR, and MIROC5, as shown in Extended Table 1.

Concerning the noted changes in tropical and subtropical biomes in Latin America, Africa, and Asia, our confidence in these results is bolstered by two lines of evidence. First, a consistent pattern of change is observed across the three dynamic global vegetation models (See Figure Below).

Response Figure 1. Average Pixel Cover Change at 2C with respect to present-day biomes. a) Tropical and Subtropical Africa. b) Tropical and Subtropical Latin America. c) Tropical and Subtropical Asia.

Second, independent research focusing on vegetation changes in these areas using different methods corroborates the DGVMs results. For example, Boonman et al. (2022) predicted that tropical forests will expand in Central Africa and north of South America at the expense of grasslands. These results are also observed in the figure above. Important to note, the total biome cover changes shown by the red dots in the figure are mostly negative. In the main manuscript, we note this in the following paragraph:

Another major pattern is the partial replacement of grasslands with forests. Grasslands show a net loss in tropical and temperate areas, with a corresponding expansion of forest biomes. In the tropics, in particular, the gain in forest areas is smaller than the loss of grasslands, leading to a net decline in vegetated area.

These results also shed light on the follow-up question from the Reviewer:

While Fig. 2D indicates that these regions experience non-significant change in the vegetation carbon content. Such results are confusing. If climate warming exert detrimental impacts on the growth and productivity of natural vegetation in tropical biomes with high temperate and aridity, the carbon content of the vegetation as a proxy for ecosystem overall health would probably be influenced under the warming stress.

The vegetation carbon content depicted in Figure 2D reflects consistent patterns of vegetation replacement. While climate change may indeed have detrimental impacts on natural vegetation in tropical biomes, two processes can enhance vegetation carbon content, which likely offset the detrimental effects of warming. First, the model results also indicate a partial replacement of grasslands by forests in some areas. In most cases, forests have more biomass per unit area compared to grasslands. Therefore, even if forests do not fully replace grasslands, the carbon content of the vegetation can remain stable due to the higher biomass content of these newly emerging forests. Second, due to the higher CO₂ content in the atmosphere, the effect of CO₂ fertilization (enhanced biomass growth) is also present. In fact, strong CO₂ fertilization effects on vegetation have already been observed, especially in warm, arid regions (Donohue et al., 2013).

We thank the reviewer for raising these observations and trust this provides clarity on the underlying results from the dynamic global vegetation models. We hope that interested readers will find this discussion valuable, as it is our understanding that the response to reviewers will be available alongside the main article.

In the discussion section, the authors could consider adding the implications and significance of their research results for formulating climate policies. In China, the assessment of natural capital for government officials upon leaving their positions is an aspect of their performance evaluation. Valuing natural assets allows for a more robust theoretical foundation to support policy management.

Thank you for your valuable suggestion regarding the inclusion of the implications of our research results for formulating climate policies. We concur that this perspective would enhance the message and relevance of our study. Accordingly, we have integrated your recommendation at the end of the Discussion and Conclusions section. We believe this addition serves as a compelling closure, emphasizing the broader policy implications of our research. The conclusion now reads:

Importantly, the findings from this research underscore the significance of formulating integrated climate policies that recognize and account for the unique natural capital of each country.

Minor comments:

Places that need to be cited in the abstract should have appropriate citations added.

We thank the reviewer for this observation and have now included the necessary citations.

Authors need to standardize the formatting of figure captions. “(Fig S1A)” should be “(Fig. S1A)”. There is a missing period. Both “Fig. 2” and “(Figure 2D)” are used in the maintext.

Thank you for pointing out the inconsistencies in the formatting of the figure captions. We have taken your feedback into account and standardized the format throughout the manuscript, ensuring consistency in referencing figures. We appreciate your meticulous attention to detail.

I suggest making standardized and uniform modifications to the format of the reference list and Supplementary Materials.

Some equations in the Supplementary Materials seem to be incomplete.

$N(t) = \sum_i [p_i(t)n_i(t)]$, eq. 1

$$N(t) = \sum_i \sum_t t^T \frac{B(t)}{(1+r)^t} \text{ eq. 2}$$

Thank you for highlighting the issue with the equations in the Supplementary Materials. The equations were indeed complete, but an unnoticed blank space caused them to appear incomplete. We have now addressed and corrected this oversight.

Supplementary Materials, Supplementary Fig 4 and Fig 5: “Changes in carbon vegetation content” should be “Changes in vegetation carbon content”.

Thank you for highlighting this oversight. We have amended the phrasing from "carbon vegetation content" to "vegetation carbon content" in Supplementary Fig 4 and Fig 5.

References

- Boonman, C. C. F., Huijbregts, M. A. J., Benítez-López, A., Schipper, A. M., Thuiller, W., & Santini, L. (2022). Trait-based projections of climate change effects on global biome distributions. *Diversity and Distributions*, 28(1), 25–37. <https://doi.org/10.1111/ddi.13431>
- Donohue, R. J., Roderick, M. L., McVicar, T. R., & Farquhar, G. D. (2013). Impact of CO₂ fertilization on maximum foliage cover across the globe’s warm, arid environments. *Geophysical Research Letters*, 40(12), 3031–3035. <https://doi.org/10.1002/grl.50563>
- Frieler, K., Lange, S., Piontek, F., Reyer, C. P. O., Schewe, J., Warszawski, L., Zhao, F., Chini, L., Denvil, S., Emanuel, K., Geiger, T., Halladay, K., Hurtt, G., Mengel, M., Murakami, D., Ostberg, S., Popp, A., Riva, R., Stevanovic, M., ... Yamagata, Y. (2017). Assessing the impacts of 1.5 °C global warming – simulation protocol of the Inter-Sectoral Impact Model Intercomparison Project (ISIMIP2b). *Geoscientific Model Development*, 10(12), 4321–4345. <https://doi.org/10.5194/gmd-10-4321-2017>
- Henrot, A.-J., Utescher, T., Erdei, B., Dury, M., Hamon, N., Ramstein, G., Krapp, M., Herold, N., Goldner, A., Favre, E., Munhoven, G., & François, L. (2017). Middle Miocene climate and

vegetation models and their validation with proxy data. *Palaeogeography, Palaeoclimatology, Palaeoecology*, 467, 95–119.

<https://doi.org/10.1016/j.palaeo.2016.05.026>

Hickler, T., Vohland, K., Feehan, J., Miller, P. A., Smith, B., Costa, L., Giesecke, T., Fronzek, S., Carter, T. R., Cramer, W., Kühn, I., & Sykes, M. T. (2012). Projecting the future distribution of European potential natural vegetation zones with a generalized, tree species-based dynamic vegetation model. *Global Ecology and Biogeography*, 21(1), 50–63.

<https://doi.org/10.1111/j.1466-8238.2010.00613.x>

Jacquemin, I., Berckmans, J., Henrot, A.-J., Dury, M., Tychon, B., Hambuckers, A., Hamdi, R., & François, L. (2021). Using the CARAIB dynamic vegetation model to simulate crop yields in Belgium: Validation and projections for the 2035 horizon. *Geo-Eco-Trop: Revue Internationale de Géologie, de Géographie et d'Écologie Tropicales*, 44(4).

<https://orbi.uliege.be/handle/2268/256139>

Kukla, T., Ahlström, A., Maezumi, S. Y., Chevalier, M., Lu, Z., Winnick, M. J., & Chamberlain, C. P. (2021). The resilience of Amazon tree cover to past and present drying. *Global and Planetary Change*, 202, 103520. <https://doi.org/10.1016/j.gloplacha.2021.103520>

Minet, J., Laloy, E., Tychon, B., & François, L. (2015). Bayesian inversions of a dynamic vegetation model at four European grassland sites. *Biogeosciences*, 12(9), 2809–2829.

<https://doi.org/10.5194/bg-12-2809-2015>

Yue, C., Ciais, P., Houghton, R. A., & Nassikas, A. A. (2020). Contribution of land use to the interannual variability of the land carbon cycle. *Nature Communications*, 11(1), Article 1.

<https://doi.org/10.1038/s41467-020-16953-8>

Reviewer Reports on the Third Revision

Referees' comments:

Referee #4:

Remarks to the Author:

I have no further comments. The revised version is satisfied.

Author Rebuttals to Third Revision:

Referees' comments:

Referee #4 (Remarks to the Author):

I have no further comments. The revised version is satisfied.

Thank you for your feedback. We appreciate your positive review and are pleased to hear that the revised version meets your expectations.